# On the Generative Utility of Cyclic Conditionals

**Chang Liu**[1]*, **Haoyue Tang**[2]†, **Tao Qin**[1], **Jintao Wang**[2], **Tie-Yan Liu**[1]
[1] Microsoft Research Asia, Beijing, 100080.   [2] Tsinghua University, Beijing, 100084.

## Abstract

We study whether and how can we model a joint distribution $p(x, z)$ using two conditional models $p(x|z)$ and $q(z|x)$ that form a cycle. This is motivated by the observation that deep generative models, in addition to a likelihood model $p(x|z)$, often also use an inference model $q(z|x)$ for extracting representation, but they rely on a usually uninformative prior distribution $p(z)$ to define a joint distribution, which may render problems like posterior collapse and manifold mismatch. To explore the possibility to model a joint distribution using only $p(x|z)$ and $q(z|x)$, we study their *compatibility* and *determinacy*, corresponding to the existence and uniqueness of a joint distribution whose conditional distributions coincide with them. We develop a general theory for operable equivalence criteria for compatibility, and sufficient conditions for determinacy. Based on the theory, we propose a novel generative modeling framework CyGen that only uses the two cyclic conditional models. We develop methods to achieve compatibility and determinacy, and to use the conditional models to fit and generate data. With the prior constraint removed, CyGen better fits data and captures more representative features, supported by both synthetic and real-world experiments.

## 1 Introduction

Deep generative models have achieved a remarkable success in the past decade for generating realistic complex data $x$ and extracting useful representations through their latent variable $z$. Variational auto-encoders (VAEs) [45, 67, 14, 15, 46, 80] follow the Bayesian framework and specify a prior distribution $p(z)$ and a likelihood model $p(x|z)$, so that a joint distribution $p(x, z) = p(z)p(x|z)$ is defined for generative modeling (the joint induces a distribution $p(x)$ on data). An inference model $q(z|x)$ is also used to approximate the posterior distribution $p(z|x)$ (derived from the joint $p(x, z)$), which serves for extracting representations. Other frameworks like generative adversarial nets [30, 25, 27], flow-based models [24, 60, 44, 31] and diffusion-based models [74, 38, 76, 49] follow the same structure, with different choices of the conditional models $p(x|z)$ and $q(z|x)$ and training objectives. While for the prior $p(z)$, there is often not much knowledge for complex data (like images, text, audio), and these models widely adopt an uninformative prior such as a standard Gaussian. This however, introduces some side effects:

- **Posterior collapse** [15, 34, 64]: The standard Gaussian prior tends to squeeze $q(z|x)$ towards the origin for all $x$, which degrades the representativeness of the inferred $z$ for $x$ and hurts downstream tasks in the latent space like classification and clustering.
- **Manifold mismatch** [22, 28, 41]: Typically the likelihood model is continuous (keeps topology), so the standard Gaussian prior would restrict the modeled data distribution to a simply-connected support, which limits the capacity for fitting data from a non-(simply) connected support.

While there are works trying to mitigate the two problems, they require either a strong domain knowledge [48, 41], or additional cost to learn a complicated prior model [52, 21, 79] sometimes even at the cost of inconvenient inference [59, 87].

---

*Correspondence to: Chang Liu <changliu@microsoft.com>.
†Work done during an internship at Microsoft Research Asia.

35th Conference on Neural Information Processing Systems (NeurIPS 2021).

One question then naturally emerges: *Can we model a joint distribution $p(x, z)$ only using the likelihood $p(x|z)$ and inference $q(z|x)$ models?* If we can, the limitations from specifying or learning a prior are then removed from the root. Also, the inference model $q(z|x)$ is then no longer a struggling approximation to a predefined posterior but participates in defining the joint distribution (avoid "inner approximation"). Modeling conditionals is also argued to be much easier than modeling marginal or joint distributions directly [2, 9, 10]. In many cases, one may even have better knowledge on the conditionals than on the prior, *e.g.* shift/rotation invariance of image representations (CNNs [51] / SphereNet [20]), and rules to extract frequency/energy features for audio [65]. It is then more natural and effective to incorporate this knowledge into the conditionals than using an uninformative prior.

In this paper, we explore such a possibility, and develop both a systematic theory and a novel generative modeling framework CyGen (**Cy**clic-conditional **Gen**erative model).

**(1)** Theoretical analysis on the question amounts to two sub-problems: can two given cyclic conditionals correspond to a common joint, and if yes, can they determine the joint uniquely. We term them *compatibility* and *determinacy* of two conditionals, corresponding to the existence and uniqueness of a common joint. For this, we develop novel compatibility criteria and sufficient conditions for determinacy. Beyond existing results, ours are operable (vs. existential [11]) and self-contained (vs. need a marginal [9, 10, 50, 32]), and are general enough to cover both continuous and discrete cases. Our compatibility criteria are also equivalence (vs. unnecessary [1, 4–6]) conditions. The seminal book [6] makes extensive analysis for various parametric families. Besides the equivalence criteria, we also extend their general analysis beyond the product support case, and also cover the Dirac case.

**(2)** In addition to its independent contribution, the theory also enables generative modeling using only the two cyclic conditional models, *i.e.* the CyGen framework. We develop methods for achieving compatibility and determinacy to make an eligible generative model, and for fitting and generating data to serve as a generative model. Efficient implementation techniques are designed. Note CyGen also determines a prior implicitly; it just does not need an explicit model for the prior (vs. [52, 21, 79, 59]). We show the practical utility of CyGen in both synthetic and real-world tasks. The improved performance in downstream classification and data generation demonstrates the advantage to mitigate the posterior collapse and manifold mismatch problems.

## 1.1 Related work

**Dependency networks** ([35]; similarly [39]) are perhaps the first to pursue the idea of modeling a joint by a set of conditionals. They use Gibbs sampling to determine the joint and are equivalent to undirected graphical models. They do not allow latent variables, so compatibility is not a key consideration as the data already specifies a joint as the common target of the conditionals. Beyond that, we introduce latent variables to better handle sensory data like images, for which we analyze the conditions for compatibility and determinacy and design novel methods to solve this different task.

**Denoising auto-encoders** (DAEs). AEs [71, 7] aim to extract data features by enforcing reconstruction through its encoder and decoder, which are deterministic hence insufficient determinacy (see Sec. 2.2.2). DAEs [82, 9, 10] use a probabilistic encoder and decoder for robust reconstruction against random data corruption. Their utility as a generative model is first noted through the equivalence to score matching (implies modeling $p(x)$) for a Gaussian RBM [81] or an infinitesimal Gaussian corruption [2]. In more general cases, the utility to modeling the joint $p(x, z)$ is studied via the Gibbs chain, *i.e.* the Markov chain with transition kernel $p(x'|z')q(z'|x)$. Under a global [9, 50, 32] or local [10] shared support condition, its stationary distribution $\pi(x, z)$ exists uniquely. But this is *not really determinacy*: even incompatible conditionals can have this unique existence, in which case $\pi(z|x) \neq q(z|x)$ [35, 9]. Moreover, the Gibbs chain does not give an explicit expression of $\pi(x, z)$ (thus intractable likelihood evaluation), and requires many iterations to converge for data generation and even for training (Walkback [9], GibbsNet [50]), making the algorithms costly and unstable.

As for *compatibility*, it is not really covered in DAEs. Existing results only consider the statistical consistency (unbiasedness under infinite data) of the $p(x|z)$ estimator by fitting $(x, z)$ data from $p^*(x)q(z|x)$ [9, 10, 50, 32], where $p^*(x)$ denotes the true data distribution. Particularly, they require a marginal $p^*(x)$ in advance, so that the joint is already defined by $p^*(x)q(z|x)$ regardless of $p(x|z)$, while compatibility (as well as determinacy) is a matter only of the two conditionals.

More crucially, the DAE loss is not proper for optimizing $q(z|x)$ as it promotes a mode-collapse behavior. This hinders both compatibility and determinacy (Sec. 3.2): one may not use $q(z|x)$ for inference, and data generation may depend on initialization. In contrast, CyGen explicitly enforces compatibility and guarantees determinacy, and enables likelihood evaluation and better generation.

**Dual learning** considers conversion between two modalities in both directions, *e.g.*, machine translation [33, 85, 84] and image style transfer [42, 90, 88, 53]. Although we also consider both directions, the fundamental distinction is that in generative modeling there is no data of the latent variable $z$ (not even unpaired). Technically, they did not consider determinacy: they require a marginal to determine a joint. We find their determinacy is actually insufficient (see Sec. 2.2.2). Their cycle-consistency loss [42, 90, 88] is a version of our compatibility criterion in the Dirac case (see Sec. 2.2.1), and we extend it to allow probabilistic conversion (see Sec. 3.1).

## 2 Compatibility and Determinacy Theory

To be a generative model, a system needs to determine a distribution on the data variable $x$. With latent variable $z$, this amounts to determining a joint distribution over $(x, z)$, which calls for compatibility and determinacy analysis for cyclic conditionals. In this section we build a general theory on the conditions for compatibility and determinacy. We begin with formalizing the problems.

**Setup.** Denote the measure spaces of the two random variables $x$ and $z$ as $(\mathbb{X}, \mathscr{X}, \xi)$ and $(\mathbb{Z}, \mathscr{Z}, \zeta)^3$, where $\mathscr{X}$, $\mathscr{Z}$ are the respective sigma-fields, and the base measures $\xi$, $\zeta$ (*e.g.*, Lebesgue measure on Euclidean spaces, counting measure on finite/discrete spaces) are sigma-finite. We use $\mathcal{X} \in \mathscr{X}$, $\mathcal{Z} \in \mathscr{Z}$ to denote measurable sets, and use "$\overset{\xi}{=}$", "$\subseteq^\xi$" as the extensions of "$=$", "$\subseteq$" up to a set of $\xi$-measure-zero (Def. A.1). Following the convention in machine learning, we call a "probability measure" as a "distribution". We do not require any further structures such as topology, metric, or linearity, for the interest of the most general conclusions that unify Euclidean/manifold and finite/discrete spaces and allow $\mathbb{X}$, $\mathbb{Z}$ to have different dimensions or types.

Joint and conditional distributions are defined on the product measure space $(\mathbb{X} \times \mathbb{Z}, \mathscr{X} \otimes \mathscr{Z}, \xi \otimes \zeta)$, where "$\times$" is the usual Cartesian product, $\mathscr{X} \otimes \mathscr{Z} := \sigma(\mathscr{X} \times \mathscr{Z})$ is the sigma-field generated by measurable rectangles from $\mathscr{X} \times \mathscr{Z}$, and $\xi \otimes \zeta$ is the product measure [13, Thm. 18.2]. Define the *slice* of $\mathcal{W} \in \mathscr{X} \otimes \mathscr{Z}$ at $z$ as $\mathcal{W}_z := \{x \mid (x, z) \in \mathcal{W}\} \in \mathscr{X}$ [13, Thm. 18.1(i)], and its *projection* onto $\mathbb{Z}$ as $\mathcal{W}^\mathbb{Z} := \{z \mid \exists x \in \mathbb{X} \text{ s.t. } (x, z) \in \mathcal{W}\} \in \mathscr{Z}$ (Appx. A.3). In a similar style, denote the *marginal* of a joint $\pi$ on $\mathbb{Z}$ as $\pi^\mathbb{Z}(\mathcal{Z}) := \pi(\mathbb{X} \times \mathcal{Z})$. To keep the same level of generality, we follow the general definition of conditionals ([13, p.457]; see also Appx. A.4): the conditional $\pi(\mathcal{X}|z)$ of a joint $\pi$ is the density function (R-N derivative) of $\pi(\mathcal{X} \times \cdot)$ w.r.t $\pi^\mathbb{Z}$. We highlight the key characteristic under this generality that $\pi(\cdot|z)$ can be arbitrary on a set of $\pi^\mathbb{Z}$-measure-zero, particularly, outside the support of $\pi^\mathbb{Z}$. Appx. A and B provide more background details and our technical preparations that are also of independent interest. The goal of analysis can then be formalized below.

**Definition 2.1** (compatibility and determinacy). We say two conditionals $\mu(\mathcal{X}|z)$, $\nu(\mathcal{Z}|x)$ are *compatible*, if there exists a joint distribution $\pi$ on $(\mathbb{X} \times \mathbb{Z}, \mathscr{X} \otimes \mathscr{Z})$ such that $\mu(\mathcal{X}|z)$ and $\nu(\mathcal{Z}|x)$ are its conditional distributions. We say two compatible conditionals have *determinacy* on a set $\mathcal{S} \in \mathscr{X} \otimes \mathscr{Z}$, if there is only one joint distribution concentrated on $\mathcal{S}$ that makes them compatible.

To put the concept into practical use, the analysis aims at operable conditions for compatibility and determinacy. We consider two cases separately (still unifying continuous and discrete cases), as they correspond to different types of generative models, and lead to qualitatively different conclusions.

### 2.1 Absolutely Continuous Case

We first consider the case where for any $z \in \mathbb{Z}$ and any $x \in \mathbb{X}$,[4] the conditionals $\mu(\cdot|z)$ and $\nu(\cdot|x)$ are either absolutely continuous (w.r.t $\xi$ and $\zeta$, resp.) [13, p.448], or zero in the sense of a measure. Equivalently, they have density functions $p(x|z)$ and $q(z|x)$ (non-negative by definition; may integrate to zero). This case include "smooth" distributions on Euclidean spaces or manifolds, and *all* distributions on finite/discrete spaces. Many generative modeling frameworks use density models thus count for this case, including VAEs [45, 67, 66, 46, 80] and diffusion-based models [74, 38, 76].

#### 2.1.1 Compatibility criterion in the absolutely continuous case

One may expect that when absolutely continuous conditionals $p(x|z)$ and $q(z|x)$ are compatible, their joint is also absolutely continuous (w.r.t $\xi \otimes \zeta$) with some density $p(x, z)$. This intuition is verified by our Lem. C.1 in Appx. C.1. One could then safely apply density function formulae and get $\frac{p(x|z)}{q(z|x)} = \frac{p(x,z)}{p(z)} / \frac{p(x,z)}{p(x)} = \frac{p(x)}{p(z)}$ factorizes into a function of $x$ and a function of $z$. Conversely, if the ratio factorizes as such $\frac{p(x|z)}{q(z|x)} = a(x)b(z)$, one could get $p(x|z)\frac{1}{Ab(z)} = q(z|x)\frac{a(x)}{A}$ where

---

[3]The symbol $\mathbb{Z}$ overwrites the symbol for the set of integers, which is not involved in this paper.

[4]There may be problems if absolute continuity holds only for $\zeta$-a.e. $z$ and $\xi$-a.e. $x$; see Appx. Example C.2.

$A := \int_{\mathbb{X}} a(x)\xi(\mathrm{d}x)$, which defines a joint density and compatibility is achieved. This intuition leads to the classical compatibility criterion [1; 4, Thm. 4.1; 5, Thm. 1; 6, Thm. 2.2.1].

However, the problem is more complicated than imagined. Berti et al. [11, Example 9] point out that the classical criterion is unfortunately *not necessary*. The subtlety is about on which region does this factorization have to hold. The classical criterion requires it to be the positive region of $p(x|z)$ which also needs to coincide with that of $q(z|x)$. But as mentioned, conditional $\mu(\cdot|z)$ can be arbitrary outside the support of the marginal $\pi^{\mathbb{Z}}$ (similarly for $\nu(\cdot|x)$), which may lead to additional positive regions that violate the requirement.[5] To address the problem, Berti et al. [11] give an equivalence criterion (Thm. 8), but it is *existential* thus less useful as the definition of compatibility itself is existential. Moreover, these criteria are restricted to either Euclidean or discrete spaces.

Next we give our *equivalence* criterion that is *operable*. In addressing the subtlety with regions, we first introduce a related concept that helps identify appropriate regions.

**Definition 2.2** ($\xi \otimes \zeta$-complete component). For a set $\mathcal{W} \in \mathscr{X} \otimes \mathscr{Z}$, we say that a set $\mathcal{S} \in \mathscr{X} \otimes \mathscr{Z}$ is a $\xi \otimes \zeta$-*complete component* of $\mathcal{W}$, if $\mathcal{S}^{\sharp} \cap \mathcal{W} \overset{\xi \otimes \zeta}{=} \mathcal{S}$, where $\mathcal{S}^{\sharp} := \mathcal{S}^{\mathbb{X}} \times \mathbb{Z} \cup \mathbb{X} \times \mathcal{S}^{\mathbb{Z}}$ is the *stretch* of $\mathcal{S}$.

Fig. 1 illustrates the concept. Roughly, the stretch $\mathcal{S}^{\sharp}$ of $\mathcal{S}$ represents the region where the conditionals are a.s. determined if $\mathcal{S}$ is the *support*[6] of the joint. If $\mathcal{S}$ is a complete component of $\mathcal{W}$, it is complete under stretching and intersecting with $\mathcal{W}$. Such a set $\mathcal{S}$ is an a.s. subset of $\mathcal{W}$ (Lem. B.12), while has a.s. the same slice as $\mathcal{W}$ does for almost all $z \in \mathcal{S}^{\mathbb{Z}}$ and $x \in \mathcal{S}^{\mathbb{X}}$ (Lem. B.16). This is critical for the normalizedness of distributions in our criterion. Appx. B.3 shows more facts. With this concept, our compatibility criterion is presented below.

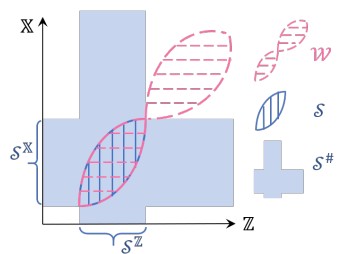

Figure 1: Illustration of a $\xi \otimes \zeta$-complete component $\mathcal{S}$ of $\mathcal{W}$.

**Theorem 2.3** (compatibility criterion, absolutely continuous). *Let* $p(x|z)$ *and* $q(z|x)$ *be the density functions of two everywhere absolutely continuous (or zero) conditional distributions, and define:*

$$\mathcal{P}_z := \{x \mid p(x|z) > 0\}, \mathcal{P}_x := \{z \mid p(x|z) > 0\},$$
$$\mathcal{Q}_z := \{x \mid q(z|x) > 0\}, \mathcal{Q}_x := \{z \mid q(z|x) > 0\}.$$

*Then they are compatible, if and only if they have a complete support* $\mathcal{S}$*, defined as a (i) $\xi \otimes \zeta$-complete component of both*

$$\mathcal{W}_{p,q} := \bigcup_{z : \mathcal{P}_z \subseteq^{\xi} \mathcal{Q}_z} \mathcal{P}_z \times \{z\}, \ \mathcal{W}_{q,p} := \bigcup_{x : \mathcal{Q}_x \subseteq^{\zeta} \mathcal{P}_x} \{x\} \times \mathcal{Q}_x,$$

*such that: (ii)* $\mathcal{S}^{\mathbb{X}} \subseteq^{\xi} \mathcal{W}_{q,p}^{\mathbb{X}}$, $\mathcal{S}^{\mathbb{Z}} \subseteq^{\zeta} \mathcal{W}_{p,q}^{\mathbb{Z}}$, *(iii)* $(\xi \otimes \zeta)(\mathcal{S}) > 0$, *and (iv)* $\frac{p(x|z)}{q(z|x)}$ *factorizes as* $a(x)b(z)$, $\xi \otimes \zeta$-*a.e. on* $\mathcal{S}$,[7] *where (v)* $a(x)$ *is* $\xi$-*integrable on* $\mathcal{S}^{\mathbb{X}}$*. For sufficiency,*

$$\pi(\mathcal{W}) := \frac{\int_{\mathcal{W} \cap \mathcal{S}} q(z|x)|a(x)|(\xi \otimes \zeta)(\mathrm{d}x\mathrm{d}z)}{\int_{\mathcal{S}^{\mathbb{X}}} |a(x)| \xi(\mathrm{d}x)}, \quad (1)$$

$\forall \mathcal{W} \in \mathscr{X} \otimes \mathscr{Z}$*, is a compatible joint of them.*

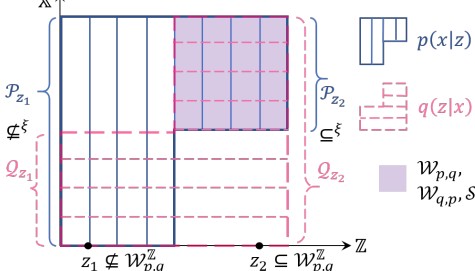

Figure 2: Illustration of our compatibility criterion in the absolutely continuous case (Thm. 2.3). The conditionals are uniform on the respective depicted slices. For condition (i), $\mathcal{P}_z \subseteq^{\xi} \mathcal{Q}_z$ is *not* satisfied on the left half, *e.g.* $z_1$, so $\mathcal{W}_{p,q}$ does not cover the left half; it is satisfied on the right half, *e.g.* $z_2$, so $\mathcal{W}_{p,q}$ is composed of slices $\mathcal{P}_z$ on the right half, making the top-right quadrant (shaded). Similarly, $\mathcal{W}_{q,p}$ is the same region, and it is a $\xi \otimes \zeta$-complete component of itself. It also satisfies other conditions thus is a complete support $\mathcal{S}$.

Fig. 2 shows an illustration of the conditions. To understand the criterion, conditions (iv) and (v) stem from the starting inspiration, which also shows a hint for Eq. (1). Other conditions handle the subtlety to find a region $\mathcal{S}$ where (iv) and (v) must hold. This is essentially the support of a compatible joint $\pi$ as there is no need and no way to control conditionals outside the support.

---

[5]The flexibility of $p(x|z)$ on a $\xi$-measure-zero set for a given $z$ (similarly for $q(z|x)$) is not a vital problem, as one can adjust the conditions to hold only a.e.

[6]While the typical definition of support requires a topological structure which is absent under our generality, Def. B.8 in Appx. B.1 defines such a concept for absolutely continuous distributions.

[7]Formally, there exist functions $a$ on $\mathcal{S}^{\mathbb{X}}$ and $b$ on $\mathcal{S}^{\mathbb{Z}}$ s.t. $(\xi \otimes \zeta)\{(x, z) \in \mathcal{S} \mid \frac{p(x|z)}{q(z|x)} \neq a(x)b(z)\} = 0$.

For necessity, informally, if $z$ is in the support of $\pi^{\mathbb{Z}}$, then $p(x|z)$ determines the distribution on $\mathbb{X} \times \{z\}$; particularly, the joint $\pi$ should be a.e. positive on $\mathcal{P}_z$, which in turn asks $q(z|x)$ to be so. This means $\mathcal{P}_z \subseteq^\xi \mathcal{Q}_z$ (unnecessary equal, since $q(z|x)$ is "out of control" outside the joint support), which leads to the definitions of $\mathcal{W}_{p,q}$ and $\mathcal{W}_{q,p}$. The joint support should be contained within the two sets in order to avoid *support conflict* (*e.g.*, although the bottom-left quadrant in Fig. 2 is part of the intersection of positive regions of the conditionals, a joint on it is required by $p(x|z)$ to also cover the top-left, on which $q(z|x)$ does not agree). Condition **(i)** indicates $\mathcal{S} \subseteq^{\xi \otimes \zeta} \mathcal{W}_{p,q}$ and $\mathcal{W}_{q,p}$ so $\mathcal{S}$ satisfies this requirement and also makes the ratio in **(iv)** a.e. well-defined. The complete-component condition in **(i)** also makes the conditionals *normalized* on $\mathcal{S}$: as mentioned, such an $\mathcal{S}$ has a.s. the same slice as $\mathcal{W}_{p,q}$ does for a given $z$ in support $\mathcal{S}^{\mathbb{Z}}$, so the integral of $p(x|z)$ on $\mathcal{S}_z$ is the same as that on $(\mathcal{W}_{p,q})_z = \mathcal{P}_z$ which is 1 by construction; similarly for $q(z|x)$. In contrast, Appx. Example C.3 shows $\mathcal{S} = \mathcal{W}_{p,q} \cap \mathcal{W}_{q,p}$ is inappropriate. Conditions **(ii)** and **(iii)** cannot be guaranteed by condition **(i)** (Appx. Example B.13), while are needed to rule out special cases (Appx. Lem. B.14, Example B.15). Appx. C.2 gives a formal proof. Finally, although the criterion relies on the *existence* of such a complete support, candidates are few (if any), so it is *operable*.

### 2.1.2 Determinacy in the absolutely continuous case

When compatible, absolutely continuous cyclic conditionals are very likely to have determinacy.

**Theorem 2.4** (determinacy, absolutely continuous)**.** *Let $p(x|z)$ and $q(z|x)$ be two compatible conditional densities, and $\mathcal{S}$ be a complete support that makes them compatible (necessarily exists due to Thm. 2.3). Suppose that $\mathcal{S}_z \stackrel{\xi}{=} \mathcal{S}^{\mathbb{X}}$, for $\zeta$-a.e. $z$ on $\mathcal{S}^{\mathbb{Z}}$, or $\mathcal{S}_x \stackrel{\zeta}{=} \mathcal{S}^{\mathbb{Z}}$, for $\xi$-a.e. $x$ on $\mathcal{S}^{\mathbb{X}}$. Then their compatible joint supported on $\mathcal{S}$ is unique, which is given by Eq. (1).*

Proof is given in Appx. C.4. The condition in the theorem roughly means that the complete support $\mathcal{S}$ is "rectangular". From the perspective of Markov chain, this corresponds to the *irreducibility* of the Gibbs chain for the unique existence of a stationary distribution. When the conditionals have multiple such complete supports, on each of which the compatible joint is unique, while globally on $\mathbb{X} \times \mathbb{Z}$, they may have multiple compatible joints. In general, determinacy in the absolutely continuous case is *sufficient*, particularly we have the following strong conclusion in a common case (*e.g.*, for VAEs).

**Corollary 2.5.** *We call two conditional densities have a.e.-full supports, if $p(x|z) > 0, q(z|x) > 0$ for $\xi \otimes \zeta$-a.e. $(x, z)$. If they are compatible, then their compatible joint is unique, since $\mathbb{X} \times \mathbb{Z}$ is the $\xi \otimes \zeta$-unique complete support (Prop. C.4 in Appx. C.3), which satisfies the condition in Thm. 2.4.*

## 2.2 Dirac Case

Many other prevailing generative models, including generative adversarial networks (GANs) [30] and flow-based models [24, 60, 44, 31], use a deterministic function $x = f(z)$ as the likelihood model. In such cases, the conditional $\mu(\mathcal{X}|z) = \delta_{f(z)}(\mathcal{X}) := \mathbb{I}[f(z) \in \mathcal{X}], \forall \mathcal{X} \in \mathscr{X}$ is a Dirac measure. Note it does not have a density function when $\xi$ assigns zero to all single-point sets, *e.g.* the Lebesgue measure on Euclidean spaces, so we keep the measure notion. This case is not exclusive to the absolutely continuous case: a Dirac conditional on a discrete space is also absolutely continuous.

### 2.2.1 Compatibility criterion in the Dirac case

Compatibility criterion is easier to imagine in this case. As illustrated in Fig. 3, it is roughly that the other-way conditional $\nu(\cdot|x)$ could find a way to put its mass only on the curve; otherwise support conflict is rendered.

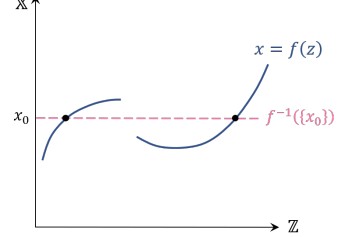

**Theorem 2.6** (compatibility criterion, Dirac)**.** *Suppose that $\mathscr{X}$ contains all the single-point sets: $\{x\} \in \mathscr{X}, \forall x \in \mathbb{X}$. Conditional distribution $\nu(\mathcal{Z}|x)$ is compatible with $\mu(\mathcal{X}|z) := \delta_{f(z)}(\mathcal{X})$ where function $f : \mathbb{Z} \to \mathbb{X}$ is $\mathscr{X}/\mathscr{Z}$-measurable[8], if and only if there exists $x_0 \in \mathbb{X}$ such that $\nu(f^{-1}(\{x_0\})|x_0) = 1$.*

Figure 3: Illustration of our compatibility criterion in the Dirac case (Thm. 2.6).

See Appx. C.6 for proof. Note such $x_0$ must be in the image set $f(\mathbb{Z})$, otherwise $\nu(f^{-1}(\{x_0\})|x_0) = \nu(\emptyset|x_0) = 0$. For a typical GAN generator, the preimage set $f^{-1}(\{x_0\})$ is discrete, so a compatible inference model must not be absolutely continuous. What may be counter-intuitive is that $\nu(\cdot|x)$ is not required to concentrate on the curve for *any* $x$; one $x_0$ is sufficient as $\delta_{(x_0, f(x_0))}$ is a compatible joint.

---

[8]For meaningful discussion, we require $f$ to be $\mathscr{X}/\mathscr{Z}$-measurable, which includes any function between discrete sets and continuous functions when $\mathscr{X}$ and $\mathscr{Z}$ are the Borel sigma-fields.

Nevertheless, in practice one often desires the compatibility to hold over a set $\mathcal{X}$ to make a useful model. When $\nu(\cdot|x)$ is also chosen in the Dirac form $\delta_{g(x)}$, this can be achieved by minimizing $\mathbb{E}_{p(x)}\ell\big(x, f(g(x))\big)$, where $p(x)$ is a distribution on $\mathcal{X}$ and $\ell$ is a metric on $\mathbb{X}$. This is the *cycle-consistency loss* used in dual learning [42, 90, 88, 53]. When $f$ is invertible, minimizing the loss (*i.e.*, $g = f^{-1}$ a.e. on $\mathcal{X}$) is also necessary, as $f^{-1}(x)$ only has one element. Particularly, flow-based models are naturally compatible (so are their injective variants [3, 86, 16] by Thm. 2.6).

### 2.2.2 Determinacy in the Dirac case

As mentioned, for any $x_0$ satisfying the condition, two compatible conditionals have the determinacy on this point $\{x_0\}$ with the unique joint $\delta_{(x_0, f(x_0))}$. But when such $x_0$ is not unique, the distribution over these $x_0$ values is not determined, so the two conditionals do not have the determinacy globally on $\mathbb{X} \times \mathbb{Z}$. This is similar to the absolutely continuous case with multiple complete supports; particularly, each $\{(x_0, f(x_0))\}$ is a complete support for discrete $\mathbb{X}$ and $\mathbb{Z}$. This meets one's intuition: compatible Dirac conditionals can only determine a curve in $\mathbb{X} \times \mathbb{Z}$, but cannot determine a distribution on the curve. One exception is when $f(z) \equiv x_0$ is constant, so this $x_0$ is the only candidate. The joint then degenerates to a distribution on $\mathbb{Z}$, which is fully determined by $\nu(\cdot|x_0)$.

In general, determinacy in the Dirac case is *insufficient*, and this type of generative models (GANs, flow-based models) have to specify a prior to define a joint.

## 3 Generative Modeling using Cyclic Conditionals

The theory suggests it is possible that cyclic conditionals achieve compatibility and a sufficient determinacy, so that they can determine a useful joint without specifying a prior. Note a certain prior is implicitly determined by the conditionals; we find we just do not need an explicit model for it. This inspires CyGen, a novel framework that only uses **Cy**clic conditionals for **Gen**erative modeling.

For the eligibility as a generative model, compatibility and a sufficient determinacy are required. For the latter, we just shown a deterministic likelihood or inference model is not suitable, so we use absolutely continuous conditionals as the theory suggests. The conditionals can then be modeled by parameterized densities $p_\theta(x|z)$, $q_\phi(z|x)$. We consider the common case where $\mathbb{X} = \mathbb{R}^{d_{\mathbb{X}}}$, $\mathbb{Z} = \mathbb{R}^{d_{\mathbb{Z}}}$, and $p_\theta(x|z)$, $q_\phi(z|x)$ have a.e.-full supports and are differentiable. Determinacy is then exactly guaranteed by Cor. 2.5. For compatibility, we develop an effective loss in Sec. 3.1 to enforce it.

For the usage as a generative model, we develop methods to fit the model-determined data distribution $p_{\theta,\phi}(x)$ to the true data distribution $p^*(x)$ in Sec. 3.2, and to generate data from $p_{\theta,\phi}(x)$ in Sec. 3.3.

### 3.1 Enforcing Compatibility

In this a.e.-full support case, the entire product space $\mathbb{X} \times \mathbb{Z}$ is the only possible complete support (Prop. C.4 in Appx. C.3), so for compatibility, condition **(iv)** in Thm. 2.3 is the most critical one. For this, we do not have to find functions $a(x)$, $b(z)$ in Thm. 2.3, but only need to enforce such a factorization. So we propose the following loss function to enforce compatibility:

$$(\min_{\theta,\phi})\quad C(\theta,\phi) := \mathbb{E}_{\rho(x,z)}\big\|\nabla_x \nabla_z^\top r_{\theta,\phi}(x,z)\big\|_F^2, \text{ where } r_{\theta,\phi}(x,z) := \log\big(p_\theta(x|z)/q_\phi(z|x)\big). \quad (2)$$

Here, $\rho$ is some absolutely continuous reference distribution on $\mathbb{X} \times \mathbb{Z}$, which can be taken as $p^*(x)q_\phi(z|x)$ in practice as it gives samples to estimate the expectation. When $C(\theta,\phi) = 0$, we have $\nabla_x \nabla_z^\top r_{\theta,\phi}(x,z) = 0$, $\xi \otimes \zeta$-a.e. [13, Thm. 15.2(ii)]. By integration, this means $\nabla_z r_{\theta,\phi}(x,z) = V(z)$ hence $r_{\theta,\phi}(x,z) = v(z) + u(x)$, $\xi \otimes \zeta$-a.e., for some functions $V(z)$, $v(z)$, $u(x)$ s.t. $V(z) = \nabla v(z)$. So the ratio $p_\theta(x|z)/q_\phi(z|x) = \exp\{r_{\theta,\phi}(x,z)\} = \exp\{u(x)\}\exp\{v(z)\}$ factorizes, $\xi \otimes \zeta$-a.s.

In the sense of enforcing compatibility, this loss generalizes the cycle-consistency loss to probabilistic conditionals. Also, the loss is different from the Jacobian-norm regularizers in contractive AE [68] and DAE [68, 2], and explains the "tied weights" trick for AEs [63, 82, 81, 68, 2] (see Appx. D.1).

**Implication on Gaussian VAE** which uses additive Gaussian conditional models, $p_\theta(x|z) := \mathcal{N}(x|f_\theta(z), \sigma_d^2 I_{d_{\mathbb{X}}})$ and $q_\phi(z|x) := \mathcal{N}(z|g_\phi(x), \sigma_e^2 I_{d_{\mathbb{Z}}})$. It is the vanilla and the most common form of VAE [45]. As its ELBO objective drives $q_\phi(z|x)$ to meet the joint $p(z)p_\theta(x|z)$, compatibility is enforced. Under our view, this amounts to minimizing the compatibility loss Eq. (2), which then enforces the match of Jacobians: $(\nabla_z f_\theta^\top(z))^\top = (\sigma_d^2/\sigma_e^2)\nabla_x g_\phi^\top(x)$. As the two sides indicate the equation is constant of both $x$ and $z$, it must be a constant, so $f_\theta(z)$ *and* $g_\phi(x)$ *must be affine*, and the joint is also a Gaussian [12; 6, Thm. 3.3.1]. This conclusion coincides with the theory on additive noise models in causality [89, 62], and explains the empirical observation that the latent space of such

VAEs is quite linear [73]. It is also the root of recent analyses that the latent space coordinates the data manifold [21], and the inference model learns an isometric embedding after a proper rescaling [58].

This finding reveals that the expectation to use deep neural networks for learning a flexible nonlinear representation will be disappointed in Gaussian VAE. So we use a non-additive-Gaussian model, *e.g.* a flow-based model [66, 46, 80, 31], for at least one of $p_\theta(x|z)$ and $q_\phi(z|x)$ (often the latter).

**Efficient implementation.** Direct Jacobian evaluation for Eq. (2) is of complexity $O(d_\mathbb{X} d_\mathbb{Z})$, which is often prohibitively large. We thus propose a stochastic but unbiased and much cheaper method based on Hutchinson's trace estimator [40]: $\mathrm{tr}(A) = \mathbb{E}_{p(\eta)}[\eta^\top A\eta]$, where $\eta$ is any random vector with zero mean and identity covariance (*e.g.*, a standard Gaussian). As the function within expectation is $\left\|\nabla_x \nabla_z^\top r\right\|_F^2 = \left\|\nabla_z \nabla_x^\top r\right\|_F^2 = \mathrm{tr}\left((\nabla_z \nabla_x^\top r)^\top \nabla_z \nabla_x^\top r\right)$, applying the estimator yields a formulation that reduces gradient evaluation complexity to $O(d_\mathbb{X} + d_\mathbb{Z})$:

$$(\min_{\theta,\phi})\ C(\theta,\phi) = \mathbb{E}_{\rho(x,z)}\mathbb{E}_{p(\eta_x)}\left\|\nabla_z\left(\eta_x^\top \nabla_x r_{\theta,\phi}(x,z)\right)\right\|_2^2, \text{where } \mathbb{E}[\eta_x] = 0, \mathrm{Var}[\eta_x] = I_{d_\mathbb{X}}. \quad (3)$$

As concluded from the above analysis on Gaussian VAE, we use a flow-based model for the inference model $q_\phi(z|x)$. But in common instances evaluating the inverse of the flow is intractable [66, 46, 80] or costly [31]. This however, disables the use of automatic differentiation tools for estimating the gradients in the compatibility loss. Appx. D.2 explains this problem in detail and shows our solution.

## 3.2 Fitting Data

After achieving compatibility, Cor. 2.5 guarantees the a.e.-fully supported conditional models uniquely determine a joint, hence a data distribution $p_{\theta,\phi}(x)$. To fit $p_{\theta,\phi}(x)$ to the true data distribution $p^*(x)$, an explicit expression is required. For this, Eq. (1) is not helpful as we do not have explicit expressions of $a(x), b(z)$. But when compatibility is given, we can safely use density function formulae:

$$p_{\theta,\phi}(x) = 1/\frac{1}{p_{\theta,\phi}(x)} = 1/\int_\mathbb{Z} \frac{p_{\theta,\phi}(z')}{p_{\theta,\phi}(x)}\zeta(\mathrm{d}z') = 1/\int_\mathbb{Z} \frac{q_\phi(z'|x)}{p_\theta(x|z')}\zeta(\mathrm{d}z') = 1/\mathbb{E}_{q_\phi(z'|x)}[1/p_\theta(x|z')],$$

which is an explicit expression in terms of the two conditionals. Although other expressions are possible, this one has a simple form, and the Monte-Carlo expectation estimation in $\mathbb{Z}$ has a lower variance than in $\mathbb{X}$ since usually $d_\mathbb{Z} \ll d_\mathbb{X}$. We can thus fit data by maximum likelihood estimation:

$$(\min_{\theta,\phi})\ \mathbb{E}_{p^*(x)}[-\log p_{\theta,\phi}(x)] = \mathbb{E}_{p^*(x)}[\log \mathbb{E}_{q_\phi(z'|x)}[1/p_\theta(x|z')]]. \quad (4)$$

The loss function can be estimated using the reparameterization trick [45] to reduce variance, and the `logsumexp` trick is adopted for numerical stability. This expression can also serve for data likelihood evaluation. The final training process of CyGen is the joint optimization with the compatibility loss.

**Comparison with DAE.** We note that the DAE loss [82, 9] $\mathbb{E}_{p^*(x)q_\phi(z'|x)}[-\log p_\theta(x|z')]$ is a *lower bound* of Eq. (4) due to Jensen's inequality, so it is not suitable for maximizing likelihood. In fact, the DAE loss minimizes $\mathbb{E}_{q_\phi(z)}\mathrm{KL}(q_\phi(x|z)\|p_\theta(x|z))$ for $p_\theta(x|z)$ to match $q_\phi(x|z)$, where $q_\phi(z)$ and $q_\phi(x|z)$ are induced from the joint $p^*(x)q_\phi(z|x)$, but it is not a proper loss for $q_\phi(z|x)$ as a *mode-collapse* behavior is promoted: the optimal $q_\phi(z|x)$ only concentrates on the point(s) of $\mathrm{argmin}_{z'}\, p_\theta(x|z')$, and an additional entropy term $-\mathbb{E}_{q_\phi(z)}\mathbb{H}[q_\phi(x|z)]$ is required to optimize the same KL loss. This behavior *hurts determinacy*, as $q_\phi(z|x)$ tends to be a (mixture of) Dirac measure (Sec. 2.2.2). The resulting Gibbs chain may also converge differently depending on initialization, as ergodicity is broken. This behavior also hurts compatibility, as $q_\phi(x|z)$ deviates from $p_\theta(x|z)$ (not Dirac), and does not match the Gibbs stationary distribution [35, 9]. In contrast, CyGen follows a more fundamental logic: enforce compatibility explicitly and follow the maximum likelihood principle faithfully. It leads to a proper loss for both conditionals that does not hinder determinacy.

## 3.3 Data Generation

Generating samples from the learned data distribution $p_{\theta,\phi}(x)$ is not as straightforward as typical models that specify a prior, since ancestral sampling is not available. But it is still tractable via Markov chain Monte Carlo methods (MCMCs). We propose using *dynamics-based MCMCs*, which are often more efficient than Gibbs sampling (used in DAE [9] and GibbsNet [50]). They only require an *unnormalized* density function of the target distribution, which is readily available in CyGen when compatible: $p_{\theta,\phi}(x) = \frac{p_{\theta,\phi}(x)}{p_{\theta,\phi}(z)}p_{\theta,\phi}(z) = \frac{p_\theta(x|z)}{q_\phi(z|x)}p_{\theta,\phi}(z) \propto \frac{p_\theta(x|z)}{q_\phi(z|x)}$ for any $z \in \mathbb{Z}$. In practice, this $z$ can be taken as a sample from $q_\phi(z|x)$ to lie in a high probability region for a confident estimate.

Stochastic gradient Langevin dynamics (SGLD) [83] is a representative instance, which has been shown to produce complicated realistic samples in energy-based [26], score-based [75] and diffusion-

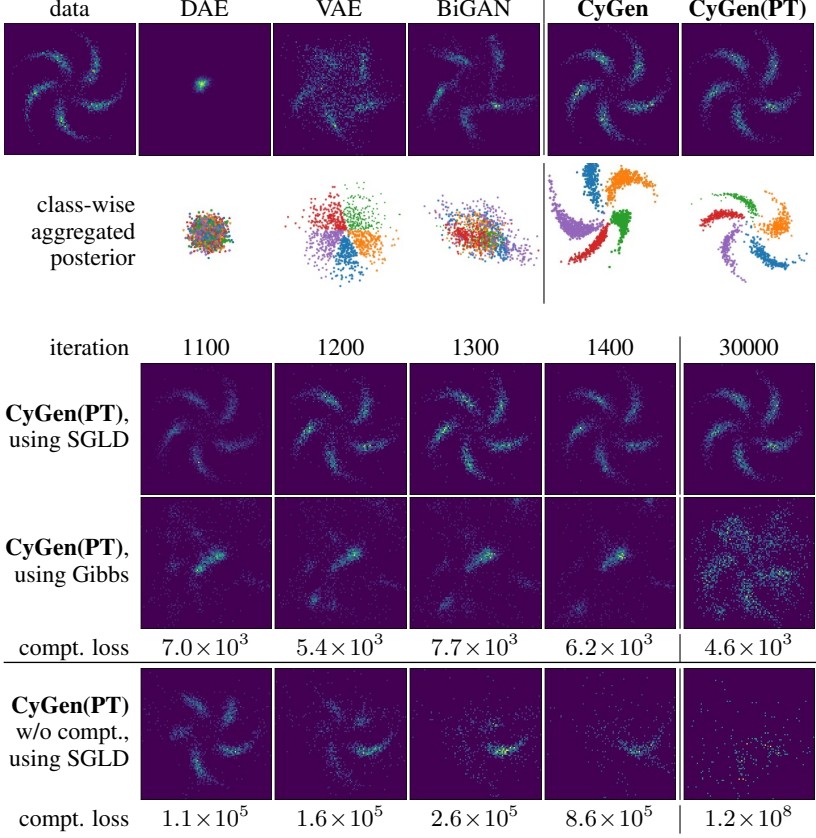

Figure 4: Generated data (DAE and CyGen use SGLD) and class-wise aggregated posteriors of DAE, VAE, BiGAN and CyGen. Also shows results of CyGen(PT) that is PreTrained as a VAE. (Best view in color.)

Figure 5: Generated data along the training process of CyGen after VAE pretraining (iteration 1000), using SGLD (rows 1,3) and Gibbs sampling (row 2) for generation, and with (rows 1,2) and without the compatibility loss (row 3) for training. See also Appx. Fig. 13.

| | data | DAE | VAE | BiGAN | **CyGen** | **CyGen(PT)** |
|---|---|---|---|---|---|---|

| | class-wise aggregated posterior |

| iteration | 1100 | 1200 | 1300 | 1400 | 30000 |
|---|---|---|---|---|---|
| **CyGen(PT)**, using SGLD | | | | | |
| **CyGen(PT)**, using Gibbs | | | | | |
| compt. loss | $7.0 \times 10^3$ | $5.4 \times 10^3$ | $7.7 \times 10^3$ | $6.2 \times 10^3$ | $4.6 \times 10^3$ |
| **CyGen(PT)** w/o compt., using SGLD | | | | | |
| compt. loss | $1.1 \times 10^5$ | $1.6 \times 10^5$ | $2.6 \times 10^5$ | $8.6 \times 10^5$ | $1.2 \times 10^8$ |

based [38, 76] models. It gives the following transition:

$$x^{(t+1)} = x^{(t)} + \varepsilon \nabla_{x^{(t)}} \log \frac{p_\theta(x^{(t)}|z^{(t)})}{q_\phi(z^{(t)}|x^{(t)})} + \sqrt{2\varepsilon}\,\eta_x^{(t)}, \text{where } z^{(t)} \sim q_\phi(z|x^{(t)}), \eta_x^{(t)} \sim \mathcal{N}(0, I_{d_\mathbb{X}}), \quad (5)$$

and $\varepsilon$ is a step size parameter. Method to draw $z \sim p_{\theta,\phi}(z)$ can be developed symmetrically (see Appx. Eq. (25)). Also applicable are other dynamics-based MCMCs [19, 23, 57], and particle-based variational inference methods [56, 17, 54, 55, 77] which are more sample-efficient.

## 4 Experiments

We demonstrate the power of CyGen for data generation and representation learning. Baselines include DAE, and generative models using Gaussian prior *e.g.* VAE and BiGAN (Appx. E.1). For a fair comparison, all methods use the same architecture, which is an additive Gaussian $p_\theta(x|z)$ and a Sylvester flow (Householder version) [80] for $q_\phi(z|x)$ (Appx. E.2), as required by CyGen (Sec. 3.1). It is necessarily probabilistic for determinacy, so we exclude flow-based generative models and common BiGAN/GibbsNet architectures, which are deterministic. We also considered GibbsNet [50] which also aims at the prior issue, but it does not produce reasonable results using the same architecture, due to its unstable training process (see Appx. E.1). Codes: `https://github.com/changliu00/cygen`.

### 4.1 Synthetic Experiments

For visual verification of the claims, we first consider a 2D toy dataset (Fig. 4 top-left). Appx. E.3 shows more details and results, including the investigation on another similar dataset.

**Data generation.** The learned data distributions (as the histogram of generated data samples) are shown in Fig. 4 (row 1). We see the five clusters are blurred to overlap in VAE's distribution and are still connected in BiGAN's, due to the specified prior. In contrast, our CyGen fits this distribution much better; particularly it clearly separates the five non-connected clusters. This verifies the advantage to overcome the *manifold mismatch* problem. As for DAE, it cannot capture the data distribution due to collapsed inference model and insufficient determinacy (Sec. 3.2).

**Representation.** Class-wise aggregated posteriors (as the scatter plot of $z$ samples from $q_\phi(z|x)p^*(x|y)$ for each class/cluster $y$) in Fig. 4 (row 2) show that CyGen mitigates the *poste-*

*rior collapse* problem, as the learned inference model $q_\phi(z|x)$ better separates the classes with a margin in the latent space. This more informative and representative feature would benefit downstream tasks like classification or clustering in the latent space. In contrast, the specified Gaussian prior squeezes the VAE latent clusters to touch, and the BiGAN latent clusters even to mix. The mode-collapsed inference model of DAE locates all latent clusters in the same place.

**Incorporating knowledge into conditionals.** CyGen alone (without pretraining) already performs well. When knowledge is available, we can further incorporate it into the conditional models. Fig. 4 shows pretraining CyGen's likelihood model as in a VAE (CyGen(PT)) embodies VAE's knowledge that the prior is centered and centrosymmetric, as the (all-class) aggregated posterior ($\approx$ prior) is such. Note its data generation quality is not sacrificed. Appx. Fig. 14 verifies this directly via the priors.

**Comparison of data generation methods.** We then make more analysis on CyGen. Fig. 5 (rows 1,2) shows generated data of CyGen using SGLD and Gibbs sampling. We see SGLD better recovers the true distribution, and is more robust to slight incompatibility.

**Impact of the compatibility loss.** Fig. 5 (rows 1,3) also shows the comparison with training CyGen without the compatibility loss. We see the compatibility is then indeed out of control, which invalidates the likelihood estimation Eq. (4) for fitting data and the gradient estimation in Eq. (5) for data generation, leading to the failure in row 3. Along the training process of the normal CyGen, we also find a smaller compatibility loss makes better generation (esp. using Gibbs sampling).

### 4.2 Real-World Experiments

We test the performance of CyGen on real-world image datasets MNIST and SVHN. We consider the VAE-pretrained version, CyGen(PT), for more stable training. Appx. E.4 shows more details. On these datasets, even BiGAN cannot produce reasonable results using the same architecture, similar to GibbsNet.

**Data generation.** From Fig. 6, We see that CyGen(PT) generates both sharp and diverse samples, as a sign to mitigate *manifold mismatch*. DAE samples are mostly imperceptible, due to the mode-collapsed $q_\phi(z|x)$ and the subsequent lack of determinacy (Sec. 3.2). VAE samples are a little blurry as a typical behavior due to the simply-connected prior. This observation is also quantitatively supported by the FID score [36, 72] on SVHN: CyGen achieves 102, while DAE 157 and VAE 128 (lower is better).

**Representation.** We then show in Table 7 that CyGen(PT)'s latent representation is

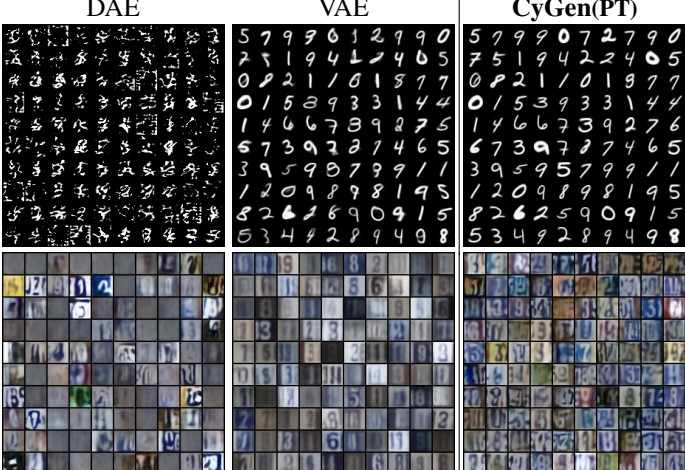

| DAE | VAE | CyGen(PT) |

Figure 6: Generated data on the MNIST and SVHN datasets.

Table 7: Downstream classification accuracy (%) using learned representation by various models.
[†]: Results from [50] using a different, deterministic architecture (not suitable for CyGen).

| | DAE | VAE | BiGAN[†] | GibbsNet[†] | **CyGen(PT)** |
|---|---|---|---|---|---|
| MNIST | $98.0_{\pm0.1}$ | $94.5_{\pm0.3}$ | 91.0 | 97.7 | $\mathbf{98.3_{\pm0.1}}$ |
| SVHN | $74.5_{\pm1.0}$ | $30.8_{\pm0.2}$ | 66.7 | **79.6** | $75.8_{\pm0.5}$ |

more informative for the downstream classification task, as an indicator to avoid *posterior collapse*. BiGAN and GibbsNet make random guess using the same probabilistic flow architecture, and their reported results in [50] using a different, deterministic architecture (not suitable for CyGen due to insufficient determinacy) are still not always better, due to the prior constraint. We conclude that CyGen achieves both superior generation and representation learning performance.

## 5 Conclusions and Discussions

In this work we investigate the possibility of defining a joint distribution using two conditional distributions, under the motivation for generative modeling without an explicit prior. We develop a systematic theory with novel and operable equivalence criteria for compatibility and sufficient conditions for determinacy, and propose a novel generative modeling framework CyGen that only

uses cyclic conditional models. Methods for achieving compatibility and determinacy, fitting data and data generation are developed. Experiments show the benefits of CyGen over DAE and prevailing generative models that specify a prior in overcoming manifold mismatch and posterior collapse.

The novel CyGen framework broadens the starting point to build a generative model, and the general theory could also foster a deeper understanding of other machine learning paradigms, *e.g.*, dual learning and self-supervised learning, and inspire more efficient algorithms.

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
