## Supplementary Materials

## A Background in Measure Theory

### A.1 The Integral

The *integral* of a nonnegative measurable function $f$ on a measure space $(\Omega, \mathscr{F}, \mu)$ is defined as:

$$\int f \, \mathrm{d}\mu := \sup \sum_i \mu(\mathcal{W}^{(i)}) \inf_{\omega \in \mathcal{W}^{(i)}} f(\omega),$$

where the supremum is taken over all finite decompositions $\{\mathcal{W}^{(i)}\}$ of $\Omega$ into $\mathscr{F}$-sets [13, p.211]. For a general measurable function, its integral is defined as the subtraction from the integral of its positive part $f^+(\omega) := \max\{0, f(\omega)\}$ with the integral of its negative part $f^-(\omega) := \max\{0, -f(\omega)\}$. A measurable function is said to be $\mu$-*integrable* [13, p.212] if both integrals of its positive and negative parts are finite.

**(i)** This is a general definition of integral. When $\Omega$ is an Euclidean space and $\mu$ is the Lebesgue measure on it, this integral reduces to the Lebesgue integral (which in turn coincides with the Riemann integral when the latter exists). When $\Omega$ is a discrete set (*i.e.*, a finite or countable set) and $\mu$ is the counting measure, this integral reduces to summation.

**(ii)** The integral satisfies common properties like linearity and monotonicity [13, Thm. 16.1], continuity under boundedness [13, Thm. 16.4, Thm. 16.5], *etc.* For a nonnegative function $f$, $\int f \, \mathrm{d}\mu = 0$ if and only if $f = 0$, $\mu$-a.e. [13, Thm. 15.2].

**(iii)** The integral over a set $\mathcal{W} \in \mathscr{F}$ is defined as $\int_{\mathcal{W}} f \, \mathrm{d}\mu := \int \mathbb{I}_{\mathcal{W}} f \, \mathrm{d}\mu$ [13, p.226], where $\mathbb{I}_{\mathcal{W}}$ is the indicator function.
**(1)** We thus sometimes also write $\int_{\Omega} f \, \mathrm{d}\mu$ for $\int f \, \mathrm{d}\mu$ to highlight the integral area. By this definition, $\int_{\mathcal{W}} f \, \mathrm{d}\mu = 0$ if $\mu(\mathcal{W}) = 0$ [13, p.226].
**(2)** For two measurable functions $f$ and $g$, if $f = g$, $\mu$-a.e., then $\int_{\mathcal{W}} f \, \mathrm{d}\mu = \int_{\mathcal{W}} g \, \mathrm{d}\mu$ for any $\mathcal{W} \in \mathscr{F}$ [13, Thm. 15.2]. The inverse also holds if $f$ and $g$ are nonnegative and $\mu$ is sigma-finite, or $f$ and $g$ are integrable [13, Thm. 16.10(i,ii)][9].
**(3)** If $f$ is a nonnegative measurable function, then $\nu(\mathcal{W}) := \int_{\mathcal{W}} f \, \mathrm{d}\mu, \forall \mathcal{W} \in \mathscr{F}$, is a measure on $(\Omega, \mathscr{F})$ [13, p.227][10]. Such a measure $\nu$ is finite, if and only if $f$ is $\mu$-integrable.

### A.2 Absolute Continuity and Radon-Nikodym Derivative

For two measures $\mu$ and $\nu$ on the same measurable space $(\Omega, \mathscr{F})$, $\nu$ is said to be *absolutely continuous* w.r.t $\mu$, denoted as $\nu \ll \mu$, if $\mu(\mathcal{W}) = 0$ indicates $\nu(\mathcal{W}) = 0$ for $\mathcal{W} \in \mathscr{F}$ [13, p.448]. If $\mu$ and $\nu$ are sigma-finite and $\nu \ll \mu$, the *Radon-Nikodym theorem* [13, Thm. 32.2] asserts that there exists a $\mu$-unique nonnegative function $f$ on $\Omega$, such that $\nu(\mathcal{W}) = \int_{\mathcal{W}} f(\omega) \mu(\mathrm{d}\omega)$ for any $\mathcal{W} \in \mathscr{F}$. Such a function $f$ is called the *Radon-Nikodym (R-N) derivative* of $\nu$ w.r.t $\mu$, and is also denoted as $\frac{\mathrm{d}\nu}{\mathrm{d}\mu}$. It represents the density function of $\nu$ w.r.t base measure $\mu$.

**(i)** Since the general definition of integral includes summation in the discrete case, this density function also includes the probability mass function in the discrete case.

**(ii)** The Dirac measure $\delta_{\omega_0}(\mathcal{W}) := \mathbb{I}_{\mathcal{W}}(\omega_0)$ ($\mathbb{I}$ is the indicator function) at a single point $\omega_0 \in \Omega$ is not absolutely continuous on Euclidean spaces w.r.t the Lebesgue measure, which assigns measure 0 to the set $\{\omega_0\}$. To be strict, the Dirac delta function is not a proper density function, since its integrals covering $\omega_0$ involve the indefinite $\infty \cdot 0$ on the component $\{\omega_0\}$ of the integral domain. Its characteristic that such integrals equal to one, is a standalone structure from being a function. So it is better treated as a measure of functional.

### A.3 Product Measure Space

Two measure spaces $(\mathbb{X}, \mathscr{X}, \xi)$ and $(\mathbb{Z}, \mathscr{Z}, \zeta)$ induce a *product measure space* $(\mathbb{X} \times \mathbb{Z}, \mathscr{X} \otimes \mathscr{Z}, \xi \otimes \zeta)$.

---

[9]13, Thm. 16.10(iii): $f = g$, $\mu$-a.e., if $\int_{\mathcal{W}} f \, \mathrm{d}\mu = \int_{\mathcal{W}} g \, \mathrm{d}\mu$ for any $\mathcal{W}$ from a pi-system $\Pi$ that generates $\mathscr{F}$, and $\Omega$ is a finite or countable union of $\Pi$-sets.

[10]Its countable additivity is guaranteed by Billingsley [13, Thm. 16.9].

**(i)** The *product sigma-field* $\mathscr{X} \otimes \mathscr{Z} := \sigma(\mathscr{X} \times \mathscr{Z})$ is the smallest sigma-field on $\mathbb{X} \times \mathbb{Z}$ containing $\mathscr{X} \times \mathscr{Z}$ (29, Thm. 22; 70, Def. 7.1; equivalently, 47, Remark 14.10; 47, Def. 14.4). Note that the Cartesian product $\mathscr{X} \times \mathscr{Z}$, representing the set of *measurable rectangles*, is only a semiring (thus also a pi-system). So we need to extend for a sigma-field. For any $\mathcal{W} \in \mathscr{X} \otimes \mathscr{Z}$, its *slice* (or section) at $z \in \mathbb{Z}$, defined by:

$$\mathcal{W}_z := \{x \mid (x, z) \in \mathcal{W}\},$$

lies in $\mathscr{X}$, and similarly $\mathcal{W}_x \in \mathscr{Z}$ [13, Thm. 18.1(i)]. We define the *projection* (or restriction) of $\mathcal{W}$ onto $\mathbb{Z}$, as $\mathcal{W}^{\mathbb{Z}} := \{z \mid \exists x \in \mathbb{X} \text{ s.t. } (x, z) \in \mathcal{W}\}$. By definition, for any $z \in \mathbb{Z} \setminus \mathcal{W}^{\mathbb{Z}}$, $\mathcal{W}_z = \emptyset$.

**(ii)** The *product measure* $\xi \otimes \zeta$ is characterized by $(\xi \otimes \zeta)(\mathcal{X} \times \mathcal{Z}) = \xi(\mathcal{X})\zeta(\mathcal{Z})$ for measurable rectangles $\mathcal{X} \times \mathcal{Z} \in \mathscr{X} \times \mathscr{Z}$. Some common conclusions require $\xi$ and $\zeta$ to be sigma-finite on $\mathscr{X}$ and $\mathscr{Z}$, respectively.

**(1)** In the characterization $(\xi \otimes \zeta)(\mathcal{X} \times \mathcal{Z}) = \xi(\mathcal{X})\zeta(\mathcal{Z})$, if the indefinite $0 \cdot \infty$ is met, it is zero. To see this, consider two sets $\mathcal{X}$ and $\mathcal{Z}$ that satisfy $\xi(\mathcal{X}) = 0$ and $\zeta(\mathcal{Z}) = 0$. Since $\zeta$ is sigma-finite, there are finite or countable disjoint $\mathscr{Z}$-sets $\mathcal{Z}^{(1)}, \mathcal{Z}^{(2)}, \cdots$ such that $\zeta(\mathcal{Z}^{(i)}) < \infty$ for any $i \geqslant 1$ and $\bigcup_{i=1}^{\infty} \mathcal{Z}^{(i)} = \mathbb{Z}$. Redefining $\mathcal{Z}^{(i)}$ as $\mathcal{Z}^{(i)} \cap \mathcal{Z}$, we have $\bigcup_{i=1}^{\infty} \mathcal{Z}^{(i)} = \mathcal{Z}$ while still $\zeta(\mathcal{Z}^{(i)}) < \infty$. So $(\xi \otimes \zeta)(\mathcal{X} \times \mathbb{Z}) = (\xi \otimes \zeta)(\mathcal{X} \times \bigcup_{i=1}^{\infty} \mathcal{Z}^{(i)}) = (\xi \otimes \zeta)(\bigcup_{i=1}^{\infty} \mathcal{X} \times \mathcal{Z}^{(i)})$. Recalling that a measure is countably additive by definition, this is $= \sum_{i=1}^{\infty} (\xi \otimes \zeta)(\mathcal{X} \times \mathcal{Z}^{(i)}) = \sum_{i=1}^{\infty} \xi(\mathcal{X})\zeta(\mathcal{Z}^{(i)}) = 0$.

**(2)** In this case, such a $\xi \otimes \zeta$ is sigma-finite on $\mathscr{X} \times \mathscr{Z}$, and the characterization on the pi-system $\mathscr{X} \times \mathscr{Z}$ determines a unique sigma-finite measure on $\sigma(\mathscr{X} \times \mathscr{Z}) = \mathscr{X} \otimes \mathscr{Z}$ [13, Thm. 10.3]. See also Galambos [29, Thm. 22]; Klenke [47, Thm. 14.14]; Rinaldo [70, Thm. 7.9]. Moreover, we have [13, Thm. 18.2]:

$$(\xi \otimes \zeta)(\mathcal{W}) = \int_{\mathbb{Z}} \xi(\mathcal{W}_z)\zeta(\mathrm{d}z) = \int_{\mathbb{X}} \zeta(\mathcal{W}_x)\xi(\mathrm{d}x), \ \forall \mathcal{W} \in \mathscr{X} \otimes \mathscr{Z}. \tag{6}$$

Since for any $z \in \mathbb{Z} \setminus \mathcal{W}^{\mathbb{Z}}$, $\mathcal{W}_z = \emptyset$ (see **(i)**) thus $\xi(\mathcal{W}_z) = 0$, we also have (by leveraging the additivity of integrals over a countable partition [13, Thm. 16.9] and that an a.e. zero function gives a zero integral [13, Thm. 15.2(i)]):

$$(\xi \otimes \zeta)(\mathcal{W}) = \int_{\mathcal{W}^{\mathbb{Z}}} \xi(\mathcal{W}_z)\zeta(\mathrm{d}z) = \int_{\mathcal{W}^{\mathbb{X}}} \zeta(\mathcal{W}_x)\xi(\mathrm{d}x), \ \forall \mathcal{W} \in \mathscr{X} \otimes \mathscr{Z}. \tag{7}$$

**(iii)** For a function $f$ on $\mathbb{X} \times \mathbb{Z}$, if it is $\mathscr{X} \otimes \mathscr{Z}$-measurable, then $f(x, \cdot)$ is $\mathscr{Z}$-measurable for any $x \in \mathbb{X}$, and $f(\cdot, z)$ is $\mathscr{X}$-measurable for any $z \in \mathbb{Z}$ [13, Thm. 18.1(ii)]. When $f$ is $\xi \otimes \zeta$-integrable, *Fubini's theorem* [13, Thm. 18.3] asserts its integral on $\mathbb{X} \times \mathbb{Z}$ can be computed iteratedly in either order:

$$\int_{\mathbb{X} \times \mathbb{Z}} f(x, z)(\xi \otimes \zeta)(\mathrm{d}x\mathrm{d}z) = \int_{\mathbb{Z}} \left( \int_{\mathbb{X}} f(x, z)\xi(\mathrm{d}x) \right) \zeta(\mathrm{d}z) = \int_{\mathbb{X}} \left( \int_{\mathbb{Z}} f(x, z)\zeta(\mathrm{d}z) \right) \xi(\mathrm{d}x). \tag{8}$$

For any $\mathcal{W} \in \mathscr{X} \otimes \mathscr{Z}$, the same equalities hold for function $\mathbb{I}_{\mathcal{W}} f$. For the first iterated integral, we have $\int_{\mathbb{X}} \mathbb{I}_{\mathcal{W}}(x, z) f(x, z)\xi(\mathrm{d}x) = \int_{\mathbb{X}} \mathbb{I}_{\mathcal{W}_z}(x) f(x, z)\xi(\mathrm{d}x) = \int_{\mathcal{W}_z} f(x, z)\xi(\mathrm{d}x)$, and on the region $\mathbb{Z} \setminus \mathcal{W}^{\mathbb{Z}}$, the integral $\int_{\mathcal{W}_z} f(x, z)\xi(\mathrm{d}x) = 0$ [13, p.226] since $\mathcal{W}_z = \emptyset$ on that region (see **(i)**). So we have a more general form of Fubini's theorem:

$$\int_{\mathcal{W}} f(x, z)(\xi \otimes \zeta)(\mathrm{d}x\mathrm{d}z) = \int_{\mathcal{W}^{\mathbb{Z}}} \left( \int_{\mathcal{W}_z} f(x, z)\xi(\mathrm{d}x) \right) \zeta(\mathrm{d}z) = \int_{\mathcal{W}^{\mathbb{X}}} \left( \int_{\mathcal{W}_x} f(x, z)\zeta(\mathrm{d}z) \right) \xi(\mathrm{d}x). \tag{9}$$

**(iv)** For a measure $\pi$ on the product measurable space $(\mathbb{X} \times \mathbb{Z}, \mathscr{X} \otimes \mathscr{Z})$, define its marginal distributions: $\pi^{\mathbb{X}}(\mathcal{X}) := \pi(\mathcal{X} \times \mathbb{Z}), \forall \mathcal{X} \in \mathscr{X}$, and $\pi^{\mathbb{Z}}(\mathcal{Z}) := \pi(\mathbb{X} \times \mathcal{Z}), \forall \mathcal{Z} \in \mathscr{Z}$.

### A.4 Conditional Distributions

In the most general case, a distribution (probability measure) $\pi$ on a measurable space $(\Omega, \mathscr{F})$ gives a *conditional distribution* (conditional probability) $\pi(\mathcal{W}|\omega)$ for $\mathcal{W} \in \mathscr{F}$ w.r.t a sub-sigma-field $\mathscr{G} \subseteq \mathscr{F}$.

**(i)** For any $\mathcal{W} \in \mathscr{F}$, the function $\mathscr{G} \to \mathbb{R}^{\geqslant 0}, \mathcal{G} \mapsto \pi(\mathcal{G} \cap \mathcal{W})$ gives a measure on $\mathscr{G}$. It is absolutely continuous w.r.t $\pi^{\mathscr{G}} : \mathscr{G} \to \mathbb{R}^{\geqslant 0}, \mathcal{G} \mapsto \pi(\mathcal{G})$, the projection of $\pi$ onto $\mathscr{G}$, due to the monotonicity (or (sub-)additivity) of measures. So the R-N derivative on $\mathscr{G}$ exists, which defines the conditional

distribution [13, p.457]:

$$\pi(\mathcal{W}|\omega) := \frac{\mathrm{d}\pi(\cdot \cap \mathcal{W})}{\mathrm{d}\pi^{\mathscr{G}}(\cdot)}(\omega),$$

where $\omega \in \Omega$. Note that as defined as an R-N derivative, the conditional distribution is only $\pi^{\mathscr{G}}$-unique.

**(ii)** As a function of $\omega$, $\pi(\mathcal{W}|\omega)$ is $\mathscr{G}$-measurable and $\pi$-integrable, and satisfies [13, p.457, Thm. 33.1]:

$$\int_{\mathcal{G}} \pi(\mathcal{W}|\omega)\pi^{\mathscr{G}}(\mathrm{d}\omega) = \pi(\mathcal{G} \cap \mathcal{W}), \forall \mathcal{G} \in \mathscr{G}. \tag{10}$$

This could serve as an alternative definition of conditional probability.

**(iii)** For $\pi^{\mathscr{G}}$-a.e. $\omega$, $\pi(\cdot|\omega)$ is a distribution (probability measure) on $(\Omega, \mathscr{F})$ [13, Thm. 33.2].

**(iv)** Conditional distributions on a product measurable space $(\mathbb{X} \times \mathbb{Z}, \mathscr{X} \otimes \mathscr{Z})$. Consider the sub-sigma-field $\mathscr{G} := \{\mathbb{X}\} \times \mathscr{Z}$. By construction, any $\mathcal{G} \in \mathscr{G}$ can be formed by $\mathcal{G} = \mathbb{X} \times \mathcal{Z}$ for some $\mathcal{Z} \in \mathscr{Z}$. So $\pi^{\mathscr{G}}(\mathcal{G}) := \pi(\mathcal{G}) = \pi(\mathbb{X} \times \mathcal{Z}) =: \pi^{\mathbb{Z}}(\mathcal{Z})$, and Eq. (10) becomes $\pi\big((\mathbb{X} \times \mathcal{Z}) \cap \mathcal{W}\big) = \int_{\mathbb{X} \times \mathcal{Z}} \pi(\mathcal{W}|x, z)\pi^{\mathscr{G}}(\mathrm{d}x\mathrm{d}z) = \int_{\mathbb{X} \times \mathcal{Z}} \pi(\mathcal{W}|x, z)\pi^{\mathbb{Z}}(\mathrm{d}z) = \int_{\mathcal{Z}} \pi(\mathcal{W}|x, z)\pi^{\mathbb{Z}}(\mathrm{d}z)$. This indicates that the conditional probability $\pi(\mathcal{W}|x, z)$ in this case is constant w.r.t $x$. We hence denote it as $\pi(\mathcal{W}|z)$.

Consider $\mathcal{W} \in \mathscr{X} \otimes \mathscr{Z}$ in the form $\mathcal{W} = \mathcal{X} \times \mathbb{Z}$ for some $\mathcal{X} \in \mathscr{X}$. For any $\mathcal{G} = \mathbb{X} \times \mathcal{Z} \in \mathscr{G}$, we have from Eq. (10) that $\int_{\mathcal{G}} \pi(\mathcal{W}|z)\pi^{\mathscr{G}}(\mathrm{d}x\mathrm{d}z) = \pi(\mathcal{G} \cap \mathcal{W}) = \pi(\mathcal{X} \times \mathcal{Z})$. From the above deduction, the l.h.s is $\int_{\mathbb{X} \times \mathcal{Z}} \pi(\mathcal{W}|z)\pi^{\mathscr{G}}(\mathrm{d}x\mathrm{d}z) = \int_{\mathcal{Z}} \pi(\mathcal{X} \times \mathbb{Z}|z)\pi^{\mathbb{Z}}(\mathrm{d}z)$. Defining $\pi(\mathcal{X}|z)$ as $\pi(\mathcal{X} \times \mathbb{Z}|z)$ for any $\mathcal{X} \in \mathscr{X}$, we have:

$$\pi(\mathcal{X} \times \mathcal{Z}) = \int_{\mathcal{Z}} \pi(\mathcal{X}|z)\pi^{\mathbb{Z}}(\mathrm{d}z) = \int_{\mathcal{X}} \pi(\mathcal{Z}|x)\pi^{\mathbb{X}}(\mathrm{d}x), \ \forall \mathcal{X} \times \mathcal{Z} \in \mathscr{X} \times \mathscr{Z}. \tag{11}$$

This is the conditional distribution in the usual sense. Note again that as defined as R-N derivatives, the conditional distributions $\pi(\mathcal{X}|z)$ and $\pi(\mathcal{Z}|x)$ are only $\pi^{\mathbb{Z}}$-unique and $\pi^{\mathbb{X}}$-unique, respectively.

For any $\mathcal{W} \in \mathscr{X} \otimes \mathscr{Z}$, define $\tilde{\pi}(\mathcal{W}) := \int_{\mathbb{Z}} \pi(\mathcal{W}_z|z)\pi^{\mathbb{Z}}(\mathrm{d}z)$. It is easy to verify that $\tilde{\pi}$ is a distribution (probability measure; thus finite and sigma-finite) on $(\mathbb{X} \times \mathbb{Z}, \mathscr{X} \otimes \mathscr{Z})$ [13, p.227], since $\pi(\mathcal{X}|z)$ is a distribution (and thus nonnegative) on $(\mathbb{X}, \mathscr{X})$ for $\pi^{\mathbb{Z}}$-a.e. $z$ [13, Thm. 33.2]. For any $\mathcal{W} = \mathcal{X} \times \mathcal{Z} \in \mathscr{X} \times \mathscr{Z}$, $\tilde{\pi}(\mathcal{W}) = \int_{\mathbb{Z}} \pi(\mathcal{X}|z)\pi^{\mathbb{Z}}(\mathrm{d}z) = \int_{\mathcal{Z}} \pi(\mathcal{X}|z)\pi^{\mathbb{Z}}(\mathrm{d}z) = \pi(\mathcal{X} \times \mathcal{Z})$ due to Eq. (11). So $\tilde{\pi}$ and $\pi$ agree on the pi-system $\mathscr{X} \times \mathscr{Z}$, which indicates that they agree on $\sigma(\mathscr{X} \times \mathscr{Z}) = \mathscr{X} \otimes \mathscr{Z}$ due to Billingsley [13, Thm. 10.3, Thm. 3.3]. This means that (see the argument in **(ii) (2)** in Supplement A.3 for the second line of the equation):

$$\pi(\mathcal{W}) = \int_{\mathbb{Z}} \pi(\mathcal{W}_z|z)\pi^{\mathbb{Z}}(\mathrm{d}z) = \int_{\mathbb{X}} \pi(\mathcal{W}_x|x)\pi^{\mathbb{X}}(\mathrm{d}x) \tag{12}$$

$$= \int_{\mathcal{W}^{\mathbb{Z}}} \pi(\mathcal{W}_z|z)\pi^{\mathbb{Z}}(\mathrm{d}z) = \int_{\mathcal{W}^{\mathbb{X}}} \pi(\mathcal{W}_x|x)\pi^{\mathbb{X}}(\mathrm{d}x), \ \forall \mathcal{W} \in \mathscr{X} \otimes \mathscr{Z}.$$

Finally, we formalize some definitions in the main text below.

**Definition A.1.** Consider a general measure space $(\Omega, \mathscr{F}, \mu)$. **(i)** We say that two measurable sets $\mathcal{S}, \tilde{\mathcal{S}} \in \mathscr{F}$ are $\mu$-*a.s. the same*, denoted as "$\mathcal{S} \overset{\mu}{=} \tilde{\mathcal{S}}$", if $\mu(\mathcal{S} \triangle \tilde{\mathcal{S}}) = 0$, where "$\triangle$" denotes the symmetric difference between two sets. **(ii)** We say that $\mathcal{S}$ is a $\mu$-*a.s. subset* of $\mathcal{W}$, denoted as "$\mathcal{S} \subseteq^{\mu} \mathcal{W}$", if $\mu(\mathcal{S} \setminus \mathcal{W}) = 0$.

# B Lemmas

## B.1 Lemmas for General Probability

**Lemma B.1.** *Let $\mathcal{O}$ be a measure-zero set, $\mu(\mathcal{O}) = 0$, on a measure space $(\Omega, \mathscr{F}, \mu)$. Then for any measurable set $\mathcal{W}$, we have $\mu(\mathcal{O} \setminus \mathcal{W}) = \mu(\mathcal{W} \cap \mathcal{O}) = 0$, and $\mu(\mathcal{W} \cup \mathcal{O}) = \mu(\mathcal{W} \setminus \mathcal{O}) = \mu(\mathcal{W})$.*

**Proof.** Due to the monotonicity of a measure [13, Thm. 16.1], we have $\mu(\mathcal{O} \setminus \mathcal{W}) \leqslant \mu(\mathcal{O}) = 0$ and $\mu(\mathcal{W} \cap \mathcal{O}) \leqslant \mu(\mathcal{O}) = 0$, so we get $\mu(\mathcal{O} \setminus \mathcal{W}) = \mu(\mathcal{W} \cap \mathcal{O}) = 0$. Since $\mu(\mathcal{W} \cup \mathcal{O}) = \mu(\mathcal{W} \cup (\mathcal{O} \setminus \mathcal{W}))$ and the two sets are disjoint, it equals to $\mu(\mathcal{W}) + \mu(\mathcal{O} \setminus \mathcal{W})$, which is $\mu(\mathcal{W})$ by the above conclusion. So we get $\mu(\mathcal{W} \cup \mathcal{O}) = \mu(\mathcal{W})$. When applying this conclusion to $\mathcal{W} \setminus \mathcal{O}$,

we have $\mu((\mathcal{W} \setminus \mathcal{O}) \cup \mathcal{O}) = \mu(\mathcal{W} \setminus \mathcal{O})$, while the l.h.s is $\mu(\mathcal{W} \cup \mathcal{O})$ which is $\mu(\mathcal{W})$ by the same conclusion. So we get $\mu(\mathcal{W} \setminus \mathcal{O}) = \mu(\mathcal{W})$. $\qquad\square$

**Lemma B.2.** *Let $\pi$ be an absolutely continuous distribution (probability measure) on a measure space $(\Omega, \mathscr{F}, \mu)$ with a density function $f$, and let $\mathcal{S} \in \mathscr{F}$ be a measurable set. Then $\pi(\mathcal{S}) = 1$ if and only if $\pi(\mathcal{W}) = \int_{\mathcal{W} \cap \mathcal{S}} f \, \mathrm{d}\mu, \forall \mathcal{W} \in \mathscr{F}$.*

*Proof.* **"Only if":** Since $\mathcal{S} \subseteq \Omega$, we have $\pi(\Omega \setminus \mathcal{S}) = \pi(\Omega) - \pi(\mathcal{S}) = 0$. For any $\mathcal{W} \in \mathscr{F}$, we have $\pi(\mathcal{W}) = \pi(\mathcal{W} \cap \mathcal{S}) + \pi(\mathcal{W} \cap (\Omega \setminus \mathcal{S}))$, while $0 \leqslant \pi(\mathcal{W} \cap (\Omega \setminus \mathcal{S})) \leqslant \pi(\Omega \setminus \mathcal{S}) = 0$. So we have $\pi(\mathcal{W}) = \pi(\mathcal{W} \cap \mathcal{S}) = \int_{\mathcal{W} \cap \mathcal{S}} f \, \mathrm{d}\mu$.

**"If":** $1 = \pi(\Omega) = \int_{\Omega \cap \mathcal{S}} f \, \mathrm{d}\mu = \int_{\mathcal{S}} f \, \mathrm{d}\mu = \int_{\mathcal{S} \cap \mathcal{S}} f \, \mathrm{d}\mu = \pi(\mathcal{S})$. $\qquad\square$

**Lemma B.3.** *Let $\mathcal{S}$ and $\tilde{\mathcal{S}}$ be two measurable sets on a measure space $(\Omega, \mathscr{F}, \mu)$ such that $\mathcal{S} \overset{\mu}{=} \tilde{\mathcal{S}}$. Then $\mu(\mathcal{S} \setminus \tilde{\mathcal{S}}) = \mu(\tilde{\mathcal{S}} \setminus \mathcal{S}) = 0$, and $\mu(\mathcal{S}) = \mu(\tilde{\mathcal{S}}) = \mu(\mathcal{S} \cup \tilde{\mathcal{S}}) = \mu(\mathcal{S} \cap \tilde{\mathcal{S}})$.*

*Proof.* Let $\mathcal{D}^+ := \tilde{\mathcal{S}} \setminus \mathcal{S}$ and $\mathcal{D}^- := \mathcal{S} \setminus \tilde{\mathcal{S}}$. By construction, we have $\mathcal{D}^+ \cap \mathcal{S} = \emptyset$ and $\mathcal{D}^- \subseteq \mathcal{S}$, so we also have $\mathcal{D}^+ \cap \mathcal{D}^- = \emptyset$, and $\tilde{\mathcal{S}} = (\mathcal{S} \setminus \mathcal{D}^-) \cup \mathcal{D}^+ = (\mathcal{S} \cup \mathcal{D}^+) \setminus \mathcal{D}^-$. By definition, $\mathcal{S} \overset{\mu}{=} \tilde{\mathcal{S}}$ indicates $0 = \mu(\mathcal{S} \triangle \tilde{\mathcal{S}}) = \mu(\mathcal{D}^+ \cup \mathcal{D}^-) = \mu(\mathcal{D}^+) + \mu(\mathcal{D}^-)$, so we have both $\mu(\mathcal{D}^+) = 0$ and $\mu(\mathcal{D}^-) = 0$. Subsequently, $\mu(\tilde{\mathcal{S}}) = \mu((\mathcal{S} \setminus \mathcal{D}^-) \cup \mathcal{D}^+) = \mu(\mathcal{S} \setminus \mathcal{D}^-) + \mu(\mathcal{D}^+) = \mu(\mathcal{S} \setminus \mathcal{D}^-) = \mu(\mathcal{S}) - \mu(\mathcal{D}^- \cap \mathcal{S}) = \mu(\mathcal{S}) - \mu(\mathcal{D}^-) = \mu(\mathcal{S})$, and $\mu(\mathcal{S} \cup \tilde{\mathcal{S}}) = \mu(\mathcal{S} \cup \mathcal{D}^+) = \mu(\mathcal{S}) + \mu(\mathcal{D}^+) = \mu(\mathcal{S})$. Noting also that $\mathcal{S} \cup \tilde{\mathcal{S}} = (\mathcal{S} \cap \tilde{\mathcal{S}}) \cup (\mathcal{S} \triangle \tilde{\mathcal{S}})$ and that this is a disjoint union, we have $\mu(\mathcal{S} \cup \tilde{\mathcal{S}}) = \mu(\mathcal{S} \cap \tilde{\mathcal{S}}) + \mu(\mathcal{S} \triangle \tilde{\mathcal{S}}) = \mu(\mathcal{S} \cap \tilde{\mathcal{S}})$. $\qquad\square$

**Lemma B.4.** *On a measure space $(\Omega, \mathscr{F}, \mu)$, "$\cdot \overset{\mu}{=} \cdot$" is an equivalence relation.*

*Proof.* Symmetry and reflexivity are obvious. For transitivity, let $\mathcal{A}$, $\mathcal{B}$ and $\mathcal{C}$ be three measurable sets such that $\mathcal{A} \overset{\mu}{=} \mathcal{B}$ and $\mathcal{B} \overset{\mu}{=} \mathcal{C}$. Since $\mathcal{A} \setminus \mathcal{C} = ((\mathcal{A} \setminus \mathcal{C}) \cap \mathcal{B}) \cup ((\mathcal{A} \setminus \mathcal{C}) \setminus \mathcal{B}) = (\mathcal{A} \cap (\mathcal{B} \setminus \mathcal{C})) \cup ((\mathcal{A} \setminus \mathcal{B}) \setminus \mathcal{C}) \subseteq (\mathcal{B} \setminus \mathcal{C}) \cup (\mathcal{A} \setminus \mathcal{B})$, we have $\mu(\mathcal{A} \setminus \mathcal{C}) \leqslant \mu(\mathcal{B} \setminus \mathcal{C}) + \mu(\mathcal{A} \setminus \mathcal{B}) = 0$ due to Lemma B.3. Similarly, $\mu(\mathcal{C} \setminus \mathcal{A}) = 0$. So $\mu(\mathcal{A} \triangle \mathcal{C}) = \mu(\mathcal{A} \setminus \mathcal{C}) + \mu(\mathcal{C} \setminus \mathcal{A}) = 0$. $\qquad\square$

**Lemma B.5.** *Let $\mathcal{S}$ and $\tilde{\mathcal{S}}$ be two measurable sets on a measure space $(\Omega, \mathscr{F}, \mu)$ such that $\mathcal{S} \overset{\mu}{=} \tilde{\mathcal{S}}$. Then for any measurable set $\mathcal{W}$, we have $\mathcal{S} \cup \mathcal{W} \overset{\mu}{=} \tilde{\mathcal{S}} \cup \mathcal{W}$, $\mathcal{S} \cap \mathcal{W} \overset{\mu}{=} \tilde{\mathcal{S}} \cap \mathcal{W}$, $\mathcal{S} \setminus \mathcal{W} \overset{\mu}{=} \tilde{\mathcal{S}} \setminus \mathcal{W}$ and $\mathcal{W} \setminus \mathcal{S} \overset{\mu}{=} \mathcal{W} \setminus \tilde{\mathcal{S}}$.*

*Proof.* Let $\mathcal{D}^+ := \tilde{\mathcal{S}} \setminus \mathcal{S}$ and $\mathcal{D}^- := \mathcal{S} \setminus \tilde{\mathcal{S}}$. By Lemma B.3, we have $\mu(\mathcal{D}^+) = 0$ and $\mu(\mathcal{D}^-) = 0$.

For any measurable set $\mathcal{W}$, we have $(\tilde{\mathcal{S}} \cup \mathcal{W}) \setminus (\mathcal{S} \cup \mathcal{W}) = \tilde{\mathcal{S}} \setminus \mathcal{S} \setminus \mathcal{W} = \mathcal{D}^+ \setminus \mathcal{W}$, and similarly $(\mathcal{S} \cup \mathcal{W}) \setminus (\tilde{\mathcal{S}} \cup \mathcal{W}) = \mathcal{D}^- \setminus \mathcal{W}$. So $\mu((\mathcal{S} \cup \mathcal{W}) \triangle (\tilde{\mathcal{S}} \cup \mathcal{W})) = \mu(((\mathcal{S} \cup \mathcal{W}) \setminus (\tilde{\mathcal{S}} \cup \mathcal{W})) \cup ((\tilde{\mathcal{S}} \cup \mathcal{W}) \setminus (\mathcal{S} \cup \mathcal{W}))) = \mu((\mathcal{D}^- \setminus \mathcal{W}) \cup (\mathcal{D}^+ \setminus \mathcal{W})) = \mu(\mathcal{D}^- \setminus \mathcal{W}) + \mu(\mathcal{D}^+ \setminus \mathcal{W}) \leqslant \mu(\mathcal{D}^-) + \mu(\mathcal{D}^+) = 0$, that is $\mathcal{S} \cup \mathcal{W} \overset{\mu}{=} \tilde{\mathcal{S}} \cup \mathcal{W}$.

Since $(\tilde{\mathcal{S}} \cap \mathcal{W}) \setminus (\mathcal{S} \cap \mathcal{W}) = (\tilde{\mathcal{S}} \setminus \mathcal{S}) \cap \mathcal{W} = \mathcal{D}^+ \cap \mathcal{W}$ and similarly $(\mathcal{S} \cap \mathcal{W}) \setminus (\tilde{\mathcal{S}} \cap \mathcal{W}) = \mathcal{D}^- \cap \mathcal{W}$, we have $\mu((\mathcal{S} \cap \mathcal{W}) \triangle (\tilde{\mathcal{S}} \cap \mathcal{W})) = \mu(((\mathcal{S} \cap \mathcal{W}) \setminus (\tilde{\mathcal{S}} \cap \mathcal{W})) \cup ((\tilde{\mathcal{S}} \cap \mathcal{W}) \setminus (\mathcal{S} \cap \mathcal{W}))) = \mu((\mathcal{D}^- \cap \mathcal{W}) \cup (\mathcal{D}^+ \cap \mathcal{W})) = \mu(\mathcal{D}^- \cap \mathcal{W}) + \mu(\mathcal{D}^+ \cap \mathcal{W}) \leqslant \mu(\mathcal{D}^-) + \mu(\mathcal{D}^+) = 0$, so $\mathcal{S} \cap \mathcal{W} \overset{\mu}{=} \tilde{\mathcal{S}} \cap \mathcal{W}$.

Since $(\tilde{\mathcal{S}} \setminus \mathcal{W}) \setminus (\mathcal{S} \setminus \mathcal{W}) = \tilde{\mathcal{S}} \setminus \mathcal{W} \setminus \mathcal{S} = \tilde{\mathcal{S}} \setminus \mathcal{S} \setminus \mathcal{W} = \mathcal{D}^+ \setminus \mathcal{W}$ and similarly $(\mathcal{S} \setminus \mathcal{W}) \setminus (\tilde{\mathcal{S}} \setminus \mathcal{W}) = \mathcal{D}^- \setminus \mathcal{W}$, we have $\mu((\mathcal{S} \setminus \mathcal{W}) \triangle (\tilde{\mathcal{S}} \setminus \mathcal{W})) = \mu(((\mathcal{S} \setminus \mathcal{W}) \setminus (\tilde{\mathcal{S}} \setminus \mathcal{W})) \cup ((\tilde{\mathcal{S}} \setminus \mathcal{W}) \setminus (\mathcal{S} \setminus \mathcal{W}))) = \mu((\mathcal{D}^- \setminus \mathcal{W}) \cup (\mathcal{D}^+ \setminus \mathcal{W})) = \mu(\mathcal{D}^- \setminus \mathcal{W}) + \mu(\mathcal{D}^+ \setminus \mathcal{W}) \leqslant \mu(\mathcal{D}^-) + \mu(\mathcal{D}^+) = 0$, so $\mathcal{S} \setminus \mathcal{W} \overset{\mu}{=} \tilde{\mathcal{S}} \setminus \mathcal{W}$.

Since $(\mathcal{W} \setminus \tilde{\mathcal{S}}) \setminus (\mathcal{W} \setminus \mathcal{S}) = \mathcal{W} \setminus (\mathcal{W} \setminus \mathcal{S}) \setminus \tilde{\mathcal{S}} = (\mathcal{W} \cap \mathcal{S}) \setminus \tilde{\mathcal{S}} = (\mathcal{S} \setminus \tilde{\mathcal{S}}) \cap \mathcal{W} = \mathcal{D}^- \cap \mathcal{W}$ and similarly $(\mathcal{W} \setminus \mathcal{S}) \setminus (\mathcal{W} \setminus \tilde{\mathcal{S}}) = \mathcal{D}^+ \cap \mathcal{W}$, we have $\mu((\mathcal{W} \setminus \mathcal{S}) \triangle (\mathcal{W} \setminus \tilde{\mathcal{S}})) = \mu(((\mathcal{W} \setminus \mathcal{S}) \setminus (\mathcal{W} \setminus \tilde{\mathcal{S}})) \cup ((\mathcal{W} \setminus \tilde{\mathcal{S}}) \setminus (\mathcal{W} \setminus \mathcal{S}))) = \mu((\mathcal{D}^+ \cap \mathcal{W}) \cup (\mathcal{D}^- \cap \mathcal{W})) = \mu(\mathcal{D}^+ \cap \mathcal{W}) + \mu(\mathcal{D}^- \cap \mathcal{W}) \leqslant \mu(\mathcal{D}^+) + \mu(\mathcal{D}^-) = 0$, so $\mathcal{W} \setminus \mathcal{S} \overset{\mu}{=} \mathcal{W} \setminus \tilde{\mathcal{S}}$. $\qquad\square$

**Definition B.6.** We say that a set satisfying a certain condition is $\mu$-*unique*, if for any two such sets $\mathcal{S}$ and $\tilde{\mathcal{S}}$, it holds that $\mathcal{S} \overset{\mu}{=} \tilde{\mathcal{S}}$.

**Lemma B.7.** *Let $\pi$ be an absolutely continuous distribution (probability measure) on a measure space $(\Omega, \mathscr{F}, \mu)$ with a density function $f$. If a set $\mathcal{S} \in \mathscr{F}$ satisfies $\pi(\mathcal{S}) = 1$ and that $f > 0$, $\mu$-a.e. on $\mathcal{S}$, then such an $\mathcal{S}$ is $\mu$-unique.*

**Proof.** Suppose we have two such sets $\mathcal{S}$ and $\tilde{\mathcal{S}}$. By Lemma B.2, we know that for any $\mathcal{W} \in \mathscr{F}$, $\pi(\mathcal{W}) = \int_{\mathcal{W} \cap \mathcal{S}} f \, \mathrm{d}\mu = \int_{\mathcal{W}} \mathbb{I}_{\mathcal{S}} f \, \mathrm{d}\mu = \int_{\mathcal{W}} \mathbb{I}_{\tilde{\mathcal{S}}} f \, \mathrm{d}\mu$. So by Billingsley [13, Thm. 16.10(ii)], we know that $\mathbb{I}_{\mathcal{S}} f = \mathbb{I}_{\tilde{\mathcal{S}}} f$, $\mu$-a.e.

Since $f > 0$, $\mu$-a.e. on $\mathcal{S}$, we know that $\mathbb{I}_{\mathcal{S}} = \mathbb{I}_{\tilde{\mathcal{S}}}$, $\mu$-a.e. on $\mathcal{S}$. This means that $\mu\{\omega \in \mathcal{S} \mid \mathbb{I}_{\mathcal{S}} \neq \mathbb{I}_{\tilde{\mathcal{S}}}\} = \mu\{\omega \in \mathcal{S} \mid \omega \notin \tilde{\mathcal{S}}\} = \mu(\mathcal{S} \setminus \tilde{\mathcal{S}}) = 0$. Symmetrically, since $f > 0$, $\mu$-a.e. also on $\tilde{\mathcal{S}}$, we know that $\mu(\tilde{\mathcal{S}} \setminus \mathcal{S}) = 0$. So we have $\mu(\mathcal{S} \triangle \tilde{\mathcal{S}}) = \mu((\mathcal{S} \setminus \tilde{\mathcal{S}}) \cup (\tilde{\mathcal{S}} \setminus \mathcal{S})) = \mu(\mathcal{S} \setminus \tilde{\mathcal{S}}) + \mu(\tilde{\mathcal{S}} \setminus \mathcal{S}) = 0$, which means that $\mathcal{S} \stackrel{\mu}{=} \tilde{\mathcal{S}}$. $\qquad \square$

The $\mu$-unique set $\mathcal{S}$ in the lemma serves as another form of the *support* of a distribution. The standard definition of the support requires a topological structure and $\mathscr{F}$ is the corresponding Borel sigma-field. If given absolute continuity $\pi \ll \mu$, this lemma enables the generality that does not require a topological structure. The condition $\pi(\mathcal{S}) = 1$ prevents $\mathcal{S}$ to be too small, while the condition that $f > 0$, $\mu$-a.e. on $\mathcal{S}$ prevents $\mathcal{S}$ to be too large.

**Definition B.8** (support of an absolutely continuous distribution (without topology))**.** Define the *support* of an absolutely continuous distribution (probability measure) $\pi$ on a measure space $(\Omega, \mathscr{F}, \mu)$, as the $\mu$-unique set $\mathcal{S} \in \mathscr{F}$ such that $\pi(\mathcal{S}) = 1$ and for any density function $f$ of $\pi$, it holds that $f > 0$, $\mu$-a.e. on $\mathcal{S}$.

## B.2 Lemmas for Product Probability

In this subsection and the following, let $(\mathbb{X} \times \mathbb{Z}, \mathscr{X} \otimes \mathscr{Z}, \xi \otimes \zeta)$ be the product measure space by the two individual ones $(\mathbb{X}, \mathscr{X}, \xi)$ and $(\mathbb{Z}, \mathscr{Z}, \zeta)$, where $\xi$ and $\zeta$ are sigma-finite.

**Lemma B.9.** *For a measure $\pi$ on the product measure space $(\mathbb{X} \times \mathbb{Z}, \mathscr{X} \otimes \mathscr{Z}, \xi \otimes \zeta)$, if $\pi \ll \xi \otimes \zeta$, then $\pi^{\mathbb{X}} \ll \xi$ and $\pi^{\mathbb{Z}} \ll \zeta$.*

**Proof.** For any $\mathcal{X} \in \mathscr{X}$ such that $\xi(\mathcal{X}) = 0$, we have $(\xi \otimes \zeta)(\mathcal{X} \times \mathbb{Z}) = \xi(\mathcal{X})\zeta(\mathbb{Z}) = 0$, where the last equality is verified in **(ii) (1)** in Supplement A.3 when $\zeta(\mathbb{Z}) = \infty$. Since $\pi \ll \xi \otimes \zeta$, this means that $\pi(\mathcal{X} \times \mathbb{Z}) = \pi^{\mathbb{X}}(\mathcal{X}) = 0$. So $\pi^{\mathbb{X}} \ll \xi$. Similarly, $\pi^{\mathbb{Z}} \ll \zeta$. $\qquad \square$

**Lemma B.10.** *For an assertion $t(x, z)$ on $\mathcal{W} \in \mathscr{X} \otimes \mathscr{Z}$, $t(x, z)$ holds $\xi \otimes \zeta$-a.e. on $\mathcal{W}$, if and only if $t(x, z)$ holds $\xi$-a.e. on $\mathcal{W}_z$, for $\zeta$-a.e. $z$ on $\mathcal{W}^{\mathbb{Z}}$.*

**Proof.** By the definition of "$t(x, z)$ holds $\xi \otimes \zeta$-a.e. on $\mathcal{W}$", we have:

$$(\xi \otimes \zeta)\{(x, z) \in \mathcal{W} \mid \neg t(x, z)\} = 0 \qquad \text{(Since } \xi \text{ and } \zeta \text{ are sigma-finite, from Eq. (7),)}$$

$$\Longleftrightarrow \int_{\mathcal{W}^{\mathbb{Z}}} \xi\{x \in \mathcal{W}_z \mid \neg t(x, z)\}\zeta(\mathrm{d}z) = 0$$

(Since $\xi(\cdot)$ is nonnegative, from Billingsley [13, Thm. 15.2],)

$$\Longleftrightarrow \xi\{x \in \mathcal{W}_z \mid \neg t(x, z)\} = 0, \text{for } \zeta\text{-a.e. } z \text{ on } \mathcal{W}^{\mathbb{Z}},$$

which is "$t(x, z)$ holds $\xi$-a.e. on $\mathcal{W}_z$, for $\zeta$-a.e. $z$ on $\mathcal{W}^{\mathbb{Z}}$". $\qquad \square$

**Lemma B.11.** *Let $\mathcal{X}, \tilde{\mathcal{X}} \in \mathscr{X}$ such that $\mathcal{X} \stackrel{\xi}{=} \tilde{\mathcal{X}}$. Then $\mathcal{X} \times \mathbb{Z} \stackrel{\xi \otimes \zeta}{=} \tilde{\mathcal{X}} \times \mathbb{Z}$.*

**Proof.** Since $(\mathcal{X} \times \mathbb{Z}) \triangle (\tilde{\mathcal{X}} \times \mathbb{Z}) = ((\mathcal{X} \times \mathbb{Z}) \setminus (\tilde{\mathcal{X}} \times \mathbb{Z})) \cup ((\tilde{\mathcal{X}} \times \mathbb{Z}) \setminus (\mathcal{X} \times \mathbb{Z})) = ((\mathcal{X} \setminus \tilde{\mathcal{X}}) \cup (\tilde{\mathcal{X}} \setminus \mathcal{X})) \times \mathbb{Z}$, we can verify that $(\xi \otimes \zeta)\big((\mathcal{X} \times \mathbb{Z}) \triangle (\tilde{\mathcal{X}} \times \mathbb{Z})\big) = (\xi \otimes \zeta)\big(((\mathcal{X} \setminus \tilde{\mathcal{X}}) \cup (\tilde{\mathcal{X}} \setminus \mathcal{X})) \times \mathbb{Z}\big) = \xi\big((\mathcal{X} \setminus \tilde{\mathcal{X}}) \cup (\tilde{\mathcal{X}} \setminus \mathcal{X})\big)\zeta(\mathbb{Z}) = \xi(\mathcal{X} \triangle \tilde{\mathcal{X}})\zeta(\mathbb{Z}) = 0$, where the last equality is verified in **(ii) (1)** in Supplement A.3 when $\zeta(\mathbb{Z}) = \infty$. $\qquad \square$

## B.3 Lemmas for $\xi \otimes \zeta$-Complete Component

Echoing Def. 2.2, a set $\mathcal{S} \in \mathscr{X} \otimes \mathscr{Z}$ is called a $\xi \otimes \zeta$-*complete component* of $\mathcal{W} \in \mathscr{X} \otimes \mathscr{Z}$, if

$$\mathcal{S}^{\sharp} \cap \mathcal{W} \stackrel{\xi \otimes \zeta}{=} \mathcal{S}, \text{ where } \mathcal{S}^{\sharp} := \mathcal{S}^{\mathbb{X}} \times \mathbb{Z} \cup \mathbb{X} \times \mathcal{S}^{\mathbb{Z}}. \tag{13}$$

This means that $\mathcal{S}$ is complete under *stretching* and intersecting with $\mathcal{W}$.

**Lemma B.12.** *Let $\mathcal{S}$ be a $\xi \otimes \zeta$-complete component of $\mathcal{W}$. Then $\mathcal{S} \subseteq^{\xi \otimes \zeta} \mathcal{W}$.*

**Proof.** By construction, we have $\mathcal{S} \subseteq \mathcal{S}^{\sharp}$ so $\mathcal{S} \setminus \mathcal{W} = \mathcal{S} \setminus (\mathcal{S} \cap \mathcal{W}) = \mathcal{S} \setminus (\mathcal{S}^{\sharp} \cap \mathcal{W})$. Hence, $(\xi \otimes \zeta)(\mathcal{S} \setminus \mathcal{W}) = (\xi \otimes \zeta)\big(\mathcal{S} \setminus (\mathcal{S}^{\sharp} \cap \mathcal{W})\big) = 0$ by definition Eq. (13) and Lemma B.3. $\qquad \square$

**Example B.13.** Note that when $\mathcal{S}$ is a $\xi\otimes\zeta$-complete component of $\mathcal{W}$, it may not hold that $\mathcal{S}^{\mathbb{X}} \subseteq^{\xi} \mathcal{W}^{\mathbb{X}}$ and $\mathcal{S}^{\mathbb{Z}} \subseteq^{\zeta} \mathcal{W}^{\mathbb{Z}}$. Fig. 8 shows an example, where $(\mathbb{X}, \mathscr{X}, \xi)$ and $(\mathbb{Z}, \mathscr{Z}, \zeta)$ are the one dimensional Euclidean spaces with line Borel sigma-field and line Lebesgue measure, $(\mathbb{R}, \mathscr{R}, \lambda)$, and $\mathcal{W} := [0,1]^2$ and $\mathcal{S} := [0,1]^2 \cup ([1,2]\times\{\frac{1}{2}\})$. We have $\mathcal{S}^{\mathbb{X}} = [0,2]$ so $\mathcal{S}^{\sharp} = ([0,2]\times\mathbb{R})\cup(\mathbb{R}\times[0,1])$ and $\mathcal{S}^{\sharp} \cap \mathcal{W} = \mathcal{W}$. Since $\mathcal{S}\triangle\mathcal{W} = [1,2]\times\{\frac{1}{2}\}$ is a line segment that has measure zero under the plane Lebesgue measure $\xi\otimes\zeta = \lambda^2$, we have $\mathcal{S} \stackrel{\xi\otimes\zeta}{=} \mathcal{W}$ so $\mathcal{S}$ is a $\xi\otimes\zeta$-complete component of $\mathcal{W}$. But $\xi(\mathcal{S}^{\mathbb{X}} \setminus \mathcal{W}^{\mathbb{X}}) = \lambda([0,2] \setminus [0,1]) = \lambda(1,2] = 1$ is not zero, so $\mathcal{S}^{\mathbb{X}} \subseteq^{\xi} \mathcal{W}^{\mathbb{X}}$ does not hold.

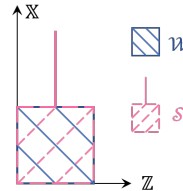

Figure 8: Example B.13 showing that a $\xi\otimes\zeta$-complete component of $\mathcal{W}$ may not have its projection be an a.s. subset of that of $\mathcal{W}$.

**Lemma B.14.** *Let $\mathcal{S}$ be a $\xi\otimes\zeta$-complete component of $\mathcal{W}$, and $\tilde{\mathcal{S}}$ be a measurable set such that $\tilde{\mathcal{S}} \stackrel{\xi\otimes\zeta}{=} \mathcal{S}$, $\tilde{\mathcal{S}}^{\mathbb{X}} \stackrel{\xi}{=} \mathcal{S}^{\mathbb{X}}$ and $\tilde{\mathcal{S}}^{\mathbb{Z}} \stackrel{\zeta}{=} \mathcal{S}^{\mathbb{Z}}$. Then this $\tilde{\mathcal{S}}$ is also a $\xi\otimes\zeta$-complete component of $\mathcal{W}$.*

**Proof.** By Lemma B.11, we know that $\tilde{\mathcal{S}}^{\mathbb{X}}\times\mathbb{Z} \stackrel{\xi\otimes\zeta}{=} \mathcal{S}^{\mathbb{X}}\times\mathbb{Z}$, $\mathbb{X}\times\tilde{\mathcal{S}}^{\mathbb{Z}} \stackrel{\xi\otimes\zeta}{=} \mathbb{X}\times\mathcal{S}^{\mathbb{Z}}$. Repeatedly applying Lemma B.5, we have $\tilde{\mathcal{S}}^{\sharp} := \tilde{\mathcal{S}}^{\mathbb{X}}\times\mathbb{Z} \cup \mathbb{X}\times\tilde{\mathcal{S}}^{\mathbb{Z}} \stackrel{\xi\otimes\zeta}{=} \mathcal{S}^{\mathbb{X}}\times\mathbb{Z} \cup \mathbb{X}\times\tilde{\mathcal{S}}^{\mathbb{Z}} \stackrel{\xi\otimes\zeta}{=} \mathcal{S}^{\mathbb{X}}\times\mathbb{Z} \cup \mathbb{X}\times\mathcal{S}^{\mathbb{Z}} =: \mathcal{S}^{\sharp}$, and $\tilde{\mathcal{S}}^{\sharp} \cap \mathcal{W} \stackrel{\xi\otimes\zeta}{=} \mathcal{S}^{\sharp} \cap \mathcal{W}$, which $\stackrel{\xi\otimes\zeta}{=} \mathcal{S} \stackrel{\xi\otimes\zeta}{=} \tilde{\mathcal{S}}$. From the transitivity (Lemma B.4), we have $\tilde{\mathcal{S}}^{\sharp} \cap \mathcal{W} \stackrel{\xi\otimes\zeta}{=} \tilde{\mathcal{S}}$. $\square$

**Example B.15.** Note that only the $\tilde{\mathcal{S}} \stackrel{\xi\otimes\zeta}{=} \mathcal{S}$ condition is not sufficient. Fig. 9 shows such an example, where $(\mathbb{X}, \mathscr{X}, \xi)$ and $(\mathbb{Z}, \mathscr{Z}, \zeta)$ are the one dimensional Euclidean spaces with line Borel sigma-field and line Lebesgue measure, $(\mathbb{R}, \mathscr{R}, \lambda)$, and $\mathcal{W} := [0,1]^2 \cup [1,2]^2$, $\mathcal{S} := [0,1]^2$, and $\tilde{\mathcal{S}} := [0,1]^2 \cup ([1,2]\times\{\frac{1}{2}\})$. We have $\mathcal{S}^{\sharp} = ([0,1]\times\mathbb{R}) \cup (\mathbb{R}\times[0,1])$ so $\mathcal{S}^{\sharp} \cap \mathcal{W} = \mathcal{S}$, justifying that $\mathcal{S}$ is a $\xi\otimes\zeta$-complete component of $\mathcal{W}$. On the other hand, since $\mathcal{S}\triangle\tilde{\mathcal{S}} = [1,2]\times\{\frac{1}{2}\}$ is a line segment that has measure zero under the plane Lebesgue measure $\xi\otimes\zeta = \lambda^2$, we have $\tilde{\mathcal{S}} \stackrel{\xi\otimes\zeta}{=} \mathcal{S}$. But $\tilde{\mathcal{S}}^{\mathbb{X}} = [0,2]$ so $\tilde{\mathcal{S}}^{\sharp} = ([0,2]\times\mathbb{R}) \cup (\mathbb{R}\times[0,1])$, which leads to $\tilde{\mathcal{S}}^{\sharp} \cap \mathcal{W} = \mathcal{W}$. Since $\tilde{\mathcal{S}}\triangle\mathcal{W} = ([1,2]\times\{\frac{1}{2}\}) \cup ([1,2]\times[1,2])$ has a nonzero measure under $\lambda^2$ (it equals to 1), we know that $\tilde{\mathcal{S}}$ is not a $\xi\otimes\zeta$-complete component of $\mathcal{W}$.

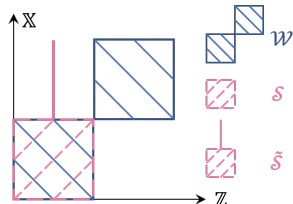

Figure 9: Example B.15 showing that in Lem. B.14, only being $\xi\otimes\zeta$-a.s. the same as a $\xi\otimes\zeta$-complete component $\tilde{\mathcal{S}}$ of $\mathcal{W}$ is not sufficient for $\tilde{\mathcal{S}}$ to be also a $\xi\otimes\zeta$-complete component of $\mathcal{W}$.

**Lemma B.16.** *Let $\mathcal{S}$ be a $\xi\otimes\zeta$-complete component of $\mathcal{W}$, and $f$ be an either nonnegative or $\xi\otimes\zeta$-integrable function on $\mathbb{X}\times\mathbb{Z}$. Then for any measurable sets $\mathcal{Z} \subseteq \mathcal{S}^{\mathbb{Z}}$ and $\mathcal{X} \subseteq \mathcal{S}^{\mathbb{X}}$, we have:*

$$\int_{\mathcal{Z}} \int_{\mathcal{W}_z} f(x,z)\xi(\mathrm{d}x)\zeta(\mathrm{d}z) = \int_{\mathcal{Z}} \int_{\mathcal{S}_z} f(x,z)\xi(\mathrm{d}x)\zeta(\mathrm{d}z),$$

$$\int_{\mathcal{X}} \int_{\mathcal{W}_x} f(x,z)\zeta(\mathrm{d}z)\xi(\mathrm{d}x) = \int_{\mathcal{X}} \int_{\mathcal{S}_x} f(x,z)\zeta(\mathrm{d}z)\xi(\mathrm{d}x).$$

*Particularly, $\int_{\mathcal{S}^{\mathbb{Z}}} \int_{\mathcal{W}_z} f(x,z)\xi(\mathrm{d}x)\zeta(\mathrm{d}z) = \int_{\mathcal{S}^{\mathbb{X}}} \int_{\mathcal{W}_x} f(x,z)\zeta(\mathrm{d}z)\xi(\mathrm{d}x) = \int_{\mathcal{S}} f(x,z)(\xi\otimes\zeta)(\mathrm{d}x\mathrm{d}z)$.*

**Proof.** Since $\mathcal{S}$ is a $\xi\otimes\zeta$-complete component of $\mathcal{W}$, Eq. (13) holds. By Lemma B.10, we know that for $\zeta$-a.e. $z$ on $\mathbb{Z}$, $\xi\big((\mathcal{S}^{\sharp} \cap \mathcal{W})\triangle\mathcal{S}\big)_z = \xi\big((\mathcal{S}^{\sharp}_z \cap \mathcal{W}_z)\triangle\mathcal{S}_z\big) = 0$. Noting that $\mathcal{S}^{\sharp}_z = \mathbb{X}$ for any $z \in \mathcal{S}^{\mathbb{Z}}$, this subsequently means that $\xi(\mathcal{W}_z\triangle\mathcal{S}_z) = 0$ for $\zeta$-a.e. $z$ on $\mathcal{S}^{\mathbb{Z}}$. By the additivity of integrals over a countable partition [13, Thm. 16.9] and that the integral over a measure-zero set is zero [13, p.226], we have $\int_{\mathcal{W}_z} f(x,z)\xi(\mathrm{d}x) = \int_{\mathcal{S}_z} f(x,z)\xi(\mathrm{d}x)$ for $\zeta$-a.e. $z$ on $\mathcal{S}^{\mathbb{Z}}$. Since a.e.-equal functions have the same integral [13, Thm. 15.2(v)], we have for any measurable $\mathcal{Z} \subseteq \mathcal{S}^{\mathbb{Z}}$, $\int_{\mathcal{Z}} \int_{\mathcal{W}_z} f(x,z)\xi(\mathrm{d}x)\zeta(\mathrm{d}z) = \int_{\mathcal{Z}} \int_{\mathcal{S}_z} f(x,z)\xi(\mathrm{d}x)\zeta(\mathrm{d}z)$. Similarly, for any measurable $\mathcal{X} \subseteq \mathcal{S}^{\mathbb{X}}$, $\int_{\mathcal{X}} \int_{\mathcal{W}_x} f(x,z)\zeta(\mathrm{d}z)\xi(\mathrm{d}x) = \int_{\mathcal{X}} \int_{\mathcal{S}_x} f(x,z)\zeta(\mathrm{d}z)\xi(\mathrm{d}x)$.

For $\mathcal{Z} = \mathcal{S}^{\mathbb{Z}}$, we have $\int_{\mathcal{S}^{\mathbb{Z}}} \int_{\mathcal{W}_z} f(x,z)\xi(\mathrm{d}x)\zeta(\mathrm{d}z) = \int_{\mathcal{S}^{\mathbb{Z}}} \int_{\mathcal{S}_z} f(x,z)\xi(\mathrm{d}x)\zeta(\mathrm{d}z)$, which is $\int_{\mathcal{S}} f(x,z)(\xi \otimes \zeta)(\mathrm{d}x\mathrm{d}z)$ by the generalized form Eq. (9) of Fubini's theorem. Similarly, $\int_{\mathcal{S}^{\mathbb{X}}} \int_{\mathcal{W}_x} f(x,z)\zeta(\mathrm{d}z)\xi(\mathrm{d}x) = \int_{\mathcal{S}} f(x,z)(\xi\otimes\zeta)(\mathrm{d}x\mathrm{d}z)$. $\square$

# C Proofs

Recall that $(\mathbb{X} \times \mathbb{Z}, \mathscr{X} \otimes \mathscr{Z}, \xi \otimes \zeta)$ is the product measure space by the two individual ones $(\mathbb{X}, \mathscr{X}, \xi)$ and $(\mathbb{Z}, \mathscr{Z}, \zeta)$, where $\xi$ and $\zeta$ are sigma-finite.

## C.1 The Joint-Conditional Absolute Continuity Lemma

Although this lemma is not formally presented in the main text, we highlight it here since it answers an important question and the answer is not straightforward.

The lemma reveals the relation between the absolute continuity of a joint $\pi$ and that of its conditionals $\pi(\cdot|z)$, $\pi(\cdot|x)$. Roughly, the former guarantees the latter on the supports of the marginals, and the reverse also holds, allowing one to safely use density function formulae for deduction. But given two conditionals, one does not have the knowledge on the marginals *a priori*. For a more useful sufficient condition, one may consider the absolute continuity of the conditionals for $\zeta$-a.e. $z$ and $\xi$-a.e. $x$. Unfortunately this is not sufficient, and an example (C.2) is given after the proof. The lemma shows it is sufficient if the absolute continuity of one of the conditionals, say $\pi(\cdot|z)$, holds for *any* $z \in \mathbb{Z}$. The condition in the compatibility criterion Thm. 2.3 is also inspired from this lemma.

**Lemma C.1** (joint-conditional absolute continuity). *(i) For a joint distribution $\pi$ on $(\mathbb{X} \times \mathbb{Z}, \mathscr{X} \otimes \mathscr{Z})$, it is absolutely continuous $\pi \ll \xi \otimes \zeta$ if and only if $\pi(\cdot|z) \ll \xi$ for $\pi^{\mathbb{Z}}$-a.e. $z$ and $\pi(\cdot|x) \ll \zeta$ for $\pi^{\mathbb{X}}$-a.e. $x$. (ii) As a sufficient condition, $\pi \ll \xi \otimes \zeta$ if $\pi(\cdot|z) \ll \xi$ for $\zeta$-a.e. $z$ and $\pi(\cdot|x) \ll \zeta$ for any $x \in \mathbb{X}$ (or for any $z \in \mathbb{Z}$ and $\xi$-a.e. $x$).*

**For conclusion (i):**

***Proof.*** **"Only if":** Consider any $\mathcal{X} \in \mathscr{X}$ such that $\xi(\mathcal{X}) = 0$. From the definition of conditional distribution Eq. (11), we have $\pi^{\mathbb{X}}(\mathcal{X}) = \pi(\mathcal{X} \times \mathbb{Z}) = \int_{\mathbb{Z}} \pi(\mathcal{X}|z) \pi^{\mathbb{Z}}(\mathrm{d}z) = 0$, so $\pi(\mathcal{X}|z) = 0$ for $\pi^{\mathbb{Z}}$-a.e. $z$ since $\pi(\mathcal{X}|z)$ is nonnegative [13, Thm. 15.2(ii)]. This means that $\pi(\cdot|z) \ll \xi$ for $\pi^{\mathbb{Z}}$-a.e. $z$. The same arguments apply symmetrically to $\pi(\cdot|x)$.

Note that since $\pi(\cdot|z)$ is defined as the R-N derivative, it is allowed to take any nonnegative value on a $\pi^{\mathbb{Z}}$-measure-zero set. So we cannot guarantee its behavior for *any* $z \in \mathbb{Z}$.

**"If":** Consider any $\mathcal{Z} \in \mathscr{Z}$ such that $\zeta(\mathcal{Z}) = 0$. Since $\pi(\cdot|x) \ll \zeta$ for $\pi^{\mathbb{X}}$-a.e. $x$, we have $\pi(\mathcal{Z}|x) = 0$ for $\pi^{\mathbb{X}}$-a.e. $x$. So from Eq. (11) we have $\pi^{\mathbb{Z}}(\mathcal{Z}) = \pi(\mathbb{X} \times \mathcal{Z}) = \int_{\mathbb{X}} \pi(\mathcal{Z}|x) \pi^{\mathbb{X}}(\mathrm{d}x) = 0$ [13, Thm. 15.2(i)]. This indicates that $\pi^{\mathbb{Z}} \ll \zeta$.

Now consider any $\mathcal{W} \in \mathscr{X} \otimes \mathscr{Z}$ such that $(\xi \otimes \zeta)(\mathcal{W}) = 0$. By the definition of product measure Eq. (6) [13, Thm. 18.2], we have $(\xi \otimes \zeta)(\mathcal{W}) = \int_{\mathbb{Z}} \xi(\mathcal{W}_z) \zeta(\mathrm{d}z) = 0$, so $\xi(\mathcal{W}_z) = 0$ for $\zeta$-a.e. $z$ since $\xi(\mathcal{W}_z)$ is nonnegative [13, Thm. 15.2(ii)]. Due to that $\pi^{\mathbb{Z}} \ll \zeta$, this means that $\xi(\mathcal{W}_z) = 0$ for $\pi^{\mathbb{Z}}$-a.e. $z$. Since $\pi(\cdot|z) \ll \xi$ for $\pi^{\mathbb{Z}}$-a.e. $z$, this in turn means that $\pi(\mathcal{W}_z|z) = 0$ for $\pi^{\mathbb{Z}}$-a.e. $z$. Subsequently, we have $\int_{\mathbb{Z}} \pi(\mathcal{W}_z|z) \pi^{\mathbb{Z}}(\mathrm{d}z) = 0$ [13, Thm. 15.2(i)], which is $\pi(\mathcal{W}) = 0$ by Eq. (12). So we get $\pi \ll \xi \otimes \zeta$. $\square$

**For conclusion (ii):**

***Proof.*** Consider any $\mathcal{Z} \in \mathscr{Z}$ such that $\zeta(\mathcal{Z}) = 0$. Since $\pi(\cdot|x) \ll \zeta$ for any $x \in \mathbb{X}$, we know that $\pi(\mathcal{Z}|x) = 0$ for any $x \in \mathbb{X}$. So from Eq. (11) we have $\pi^{\mathbb{Z}}(\mathcal{Z}) = \pi(\mathbb{X} \times \mathcal{Z}) = \int_{\mathbb{X}} \pi(\mathcal{Z}|x) \pi^{\mathbb{X}}(\mathrm{d}x) = 0$. This indicates that $\pi^{\mathbb{Z}} \ll \zeta$.

Now consider any $\mathcal{W} \in \mathscr{X} \otimes \mathscr{Z}$ such that $(\xi \otimes \zeta)(\mathcal{W}) = 0$. By the definition of product measure Eq. (6) [13, Thm. 18.2], we have $(\xi \otimes \zeta)(\mathcal{W}) = \int_{\mathbb{Z}} \xi(\mathcal{W}_z) \zeta(\mathrm{d}z) = 0$, so $\xi(\mathcal{W}_z) = 0$ for $\zeta$-a.e. $z$ since $\xi(\mathcal{W}_z)$ is nonnegative [13, Thm. 15.2(ii)]. Due to that $\pi(\cdot|z) \ll \xi$ for $\zeta$-a.e. $z$, this means that $\pi(\mathcal{W}_z|z) = 0$ for $\zeta$-a.e. $z$. Since $\pi^{\mathbb{Z}} \ll \zeta$, this in turn means that $\pi(\mathcal{W}_z|z) = 0$ for $\pi^{\mathbb{Z}}$-a.e. $z$. Subsequently, we have $\int_{\mathbb{Z}} \pi(\mathcal{W}_z|z) \pi^{\mathbb{Z}}(\mathrm{d}z) = 0$ [13, Thm. 15.2(i)], which is $\pi(\mathcal{W}) = 0$ by Eq. (12). So we get $\pi \ll \xi \otimes \zeta$. The same arguments apply symmetrically when $\pi(\cdot|z) \ll \xi$ for *any* $z \in \mathbb{Z}$ and $\pi(\cdot|x) \ll \zeta$ for $\xi$-a.e. $x$. $\square$

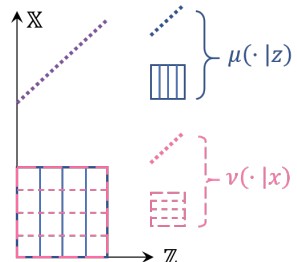

Figure 10: Illustration of the conditionals in Eq. (14) in Example C.2. Both conditionals are absolutely continuous for $\zeta$-a.e. $z$ or $\xi$-a.e. $x$, but they allow a compatible joint that is not absolutely continuous w.r.t $\xi \otimes \zeta$.

**Example C.2.** To see why it is not sufficient if the two conditionals are absolutely continuous only for $\zeta$-a.e. $z$ and $\xi$-a.e. $x$, we show an example below.

Consider the one-dimensional Euclidean space $\mathbb{X} = \mathbb{Z} = \mathbb{R}$ with line Borel sigma-field $\mathscr{X} = \mathscr{Z} = \mathscr{R}$ and line Lebesgue measure $\xi = \zeta = \lambda$. Let

$$\mu(\cdot|z) := \begin{cases} \delta_{z+2}, & z \in \mathbb{Q}[0,1], \\ \text{Unif}[0,1], & z \in \bar{\mathbb{Q}}[0,1], \\ 0, & \text{otherwise,} \end{cases} \quad \nu(\cdot|x) := \begin{cases} \text{Unif}[0,1], & x \in [0,1], \\ \delta_{x-2}, & x \in \mathbb{Q}[2,3], \\ 0, & \text{otherwise,} \end{cases} \quad (14)$$

where $\mathbb{Q}[0,1] := [0,1] \cap \mathbb{Q}$ and $\bar{\mathbb{Q}} := [0,1] \backslash \mathbb{Q}$ are the rational and irrational numbers on $[0,1]$. The conditionals are illustrated in Fig. 10. Since $\lambda(\mathbb{Q}) = 0$, the two conditionals are absolutely continuous for $\zeta$-a.e. $z$ and $\xi$-a.e. $x$. Consider the joint distribution on $\mathbb{X} \times \mathbb{Z} = \mathbb{R}^2$:

$$\pi := \frac{1}{2}\text{Unif}([0,1] \times [0,1]) + \frac{1}{2}\sum_{z \in \mathbb{Q}[0,1]} \varrho(z)\delta_{(z+2,z)},$$

where $\varrho$ is a distribution on the rationals $\mathbb{Q}[0,1]$ in $[0,1]$ with the sigma-field of all the subsets of $\mathbb{Q}[0,1]$. Such a distribution exists, for example, $\varrho(z) = 1/2^{n(z)}$ where $n : \mathbb{Q}[0,1] \to \mathbb{N}^*$ bijective is a numbering function of the countable set $\mathbb{Q}[0,1]$. In this way, each rational number $z \in \mathbb{Q}[0,1]$ has a positive probability, meanwhile we have $\varrho(\mathbb{Q}[0,1]) = \sum_{n=1}^{\infty} 1/2^n = 1$. Since $\pi(\{(z+2,z) \mid z \in \mathbb{Q}[0,1]\}) = \frac{1}{2}$ but $\lambda^2(\{(z+2,z) \mid z \in \mathbb{Q}[0,1]\}) = 0$ under the square Lebesgue measure $\lambda^2$, $\pi$ is not absolutely continuous.

To verify compatibility, note that $\mu$ and $\nu$ here satisfy the corresponding measurability and integrability. To verify Eq. (11) reduced from Eq. (10) for defining a conditional, we first derive the marginals:

$$\pi^{\mathbb{X}} = \frac{1}{2}\text{Unif}[0,1] + \frac{1}{2}\sum_{z \in \mathbb{Q}[0,1]} \varrho(z)\delta_{z+2}, \quad \pi^{\mathbb{Z}} = \frac{1}{2}\text{Unif}[0,1] + \frac{1}{2}\sum_{z \in \mathbb{Q}[0,1]} \varrho(z)\delta_z.$$

For any $\mathcal{X} \in \mathscr{X}$ and $\mathcal{Z} \in \mathscr{Z}$, we have:

$$\pi(\mathcal{X} \times \mathcal{Z}) = \frac{1}{2}\text{Unif}[0,1](\mathcal{X})\text{Unif}[0,1](\mathcal{Z}) + \frac{1}{2}\sum_{z \in \mathbb{Q}[0,1]} \varrho(z)\mathbb{I}[(z+2,z) \in \mathcal{X} \times \mathcal{Z}]$$

$$= \frac{1}{2}\lambda(\mathcal{X}[0,1])\lambda(\mathcal{Z}[0,1]) + \frac{1}{2}\sum_{z \in \mathbb{Q}[0,1]} \varrho(z)\mathbb{I}[z \in (\mathcal{X} - 2) \cap \mathcal{Z}],$$

where $\mathcal{X}[0,1] := [0,1] \cap \mathcal{X}$ and $\mathcal{Z}[0,1] := [0,1] \cap \mathcal{Z}$. To verify the conditional distribution $\mu(\mathcal{X}|z)$, we have:

$$\int_{\mathcal{Z}} \mu(\mathcal{X}|z)\pi^{\mathbb{Z}}(\mathrm{d}z) = \int_{\mathcal{Z} \cap \mathbb{Q}[0,1]} \delta_{z+2}(\mathcal{X})\pi^{\mathbb{Z}}(\mathrm{d}z) + \int_{\mathcal{Z} \cap \bar{\mathbb{Q}}[0,1]} \text{Unif}[0,1](\mathcal{X})\pi^{\mathbb{Z}}(\mathrm{d}z)$$

$$= \int_{\mathcal{Z} \cap \mathbb{Q}[0,1]} \mathbb{I}[z+2 \in \mathcal{X}]\pi^{\mathbb{Z}}(\mathrm{d}z) + \int_{\mathcal{Z} \cap \bar{\mathbb{Q}}[0,1]} \lambda(\mathcal{X}[0,1])\pi^{\mathbb{Z}}(\mathrm{d}z)$$

$$= \pi^{\mathbb{Z}}((\mathcal{X} - 2) \cap \mathcal{Z} \cap \mathbb{Q}[0,1]) + \lambda(\mathcal{X}[0,1])\pi^{\mathbb{Z}}(\mathcal{Z} \cap \bar{\mathbb{Q}}[0,1])$$

(Since a countable set has measure zero under $\text{Unif}[0,1]$, *i.e.* $\lambda(\cdot \cap [0,1])$,)

$$= \frac{1}{2}\sum_{z \in \mathbb{Q}[0,1]} \varrho(z)\mathbb{I}[z \in (\mathcal{X} - 2) \cap \mathcal{Z} \cap \mathbb{Q}[0,1]]$$

$$+ \frac{1}{2}\lambda(\mathcal{X}[0,1])\left(\text{Unif}[0,1](\mathcal{Z}[0,1]) + \sum_{z \in \mathbb{Q}[0,1]} \varrho(z)\mathbb{I}[z \in \mathcal{Z} \cap \bar{\mathbb{Q}}[0,1]]\right)$$

$$= \frac{1}{2}\sum_{z \in \mathbb{Q}[0,1]} \varrho(z)\mathbb{I}[z \in (\mathcal{X} - 2) \cap \mathcal{Z}] + \frac{1}{2}\lambda(\mathcal{X}[0,1])\lambda(\mathcal{Z}[0,1]) = \pi(\mathcal{X} \times \mathcal{Z}).$$

For the conditional distribution $\nu(\mathcal{Z}|x)$, we similarly have:

$$\int_{\mathcal{X}} \nu(\mathcal{Z}|x)\pi^{\mathbb{X}}(\mathrm{d}x) = \int_{\mathcal{X}[0,1]} \text{Unif}[0,1](\mathcal{Z})\pi^{\mathbb{X}}(\mathrm{d}x) + \int_{\mathcal{X} \cap \mathbb{Q}[2,3]} \delta_{x-2}(\mathcal{Z})\pi^{\mathbb{X}}(\mathrm{d}x)$$

$$= \lambda(\mathcal{Z}[0,1])\pi^{\mathbb{X}}(\mathcal{X}[0,1]) + \pi^{\mathbb{X}}(\mathcal{X} \cap \mathbb{Q}[2,3] \cap (\mathcal{Z} + 2))$$

$$= \frac{1}{2}\lambda(\mathcal{Z}[0,1])\lambda(\mathcal{X}[0,1]) + \frac{1}{2}\sum_{z\in\mathbb{Q}[0,1]}\varrho(z)\mathbb{I}[z+2\in\mathcal{X}\cap\mathbb{Q}[2,3]\cap(\mathcal{Z}+2)]$$

$$= \frac{1}{2}\lambda(\mathcal{Z}[0,1])\lambda(\mathcal{X}[0,1]) + \frac{1}{2}\sum_{z\in\mathbb{Q}[0,1]}\varrho(z)\mathbb{I}[z\in(\mathcal{X}-2)\cap\mathcal{Z}] = \pi(\mathcal{X}\times\mathcal{Z}).$$

So the two conditionals $\mu(\cdot|z)$ and $\nu(\cdot|x)$ are compatible and $\pi$ is their joint distribution. This example illustrates that the absolute continuity of $\pi(\cdot|z)$ w.r.t $\xi$ for $\zeta$-a.e. $z$ and that of $\pi(\cdot|x)$ w.r.t $\zeta$ for $\xi$-a.e. $x$, does not indicate the absolute continuity of $\pi$ w.r.t $\xi\otimes\zeta$.

This example does not contradict result **(i)** of the Lemma. For any $z_0 \in \mathbb{Q}[0,1]$, we have that $\mu(\cdot|z_0) = \delta_{z_0+2}$ is not absolutely continuous w.r.t $\xi = \lambda$. But $\pi^{\mathbb{Z}}(\{z_0\}) = \frac{1}{2}\varrho(z_0) > 0$. So it is not that $\mu(\cdot|z) \ll \xi$ for $\pi^{\mathbb{Z}}$-a.e. $z$, which aligns with that $\pi$ is not absolutely continuous w.r.t $\xi\otimes\zeta = \lambda^2$.

This example also shows that the absolute continuity of the compatible joint may depend on the joint itself, apart from the two conditionals. Consider another joint on $(\mathbb{R}^2, \mathscr{R}^2, \lambda^2)$:

$$\tilde{\pi} := \mathrm{Unif}([0,1]\times[0,1]).$$

It is easy to see that $\tilde{\pi}^{\mathbb{X}}(\mathcal{X}) = \mathrm{Unif}[0,1](\mathcal{X}) = \lambda(\mathcal{X}[0,1])$ and $\tilde{\pi}^{\mathbb{Z}}(\mathcal{Z}) = \lambda(\mathcal{Z}[0,1])$. For any $\mathcal{X}\in\mathscr{X}$ and $\mathcal{Z}\in\mathscr{Z}$, we have $\tilde{\pi}(\mathcal{X}\times\mathcal{Z}) = \lambda(\mathcal{X}[0,1])\lambda(\mathcal{Z}[0,1])$. To verify Eq. (11) for defining a conditional, we have:

$$\int_{\mathcal{Z}}\mu(\mathcal{X}|z)\tilde{\pi}^{\mathbb{Z}}(\mathrm{d}z) = \int_{\mathcal{Z}\cap\mathbb{Q}[0,1]}\delta_{z+2}(\mathcal{X})\tilde{\pi}^{\mathbb{Z}}(\mathrm{d}z) + \int_{\mathcal{Z}\cap\bar{\mathbb{Q}}[0,1]}\mathrm{Unif}[0,1](\mathcal{X})\tilde{\pi}^{\mathbb{Z}}(\mathrm{d}z)$$

$$= \int_{\mathcal{Z}\cap\mathbb{Q}[0,1]}\mathbb{I}[z+2\in\mathcal{X}]\tilde{\pi}^{\mathbb{Z}}(\mathrm{d}z) + \int_{\mathcal{Z}\cap\bar{\mathbb{Q}}[0,1]}\lambda(\mathcal{X}[0,1])\tilde{\pi}^{\mathbb{Z}}(\mathrm{d}z)$$

$$= \lambda((\mathcal{X}-2)\cap\mathcal{Z}\cap\mathbb{Q}[0,1]) + \lambda(\mathcal{X}[0,1])\lambda(\mathcal{Z}\cap\bar{\mathbb{Q}}[0,1])$$

(Since $\lambda((\mathcal{X}-2)\cap\mathcal{Z}\cap\mathbb{Q}[0,1]) \leqslant \lambda(\mathbb{Q}) = 0$ and $\lambda(\mathcal{Z}\cap\bar{\mathbb{Q}}[0,1]) = \lambda(\mathcal{Z}\cap\bar{\mathbb{Q}}[0,1]) + \lambda(\mathbb{Q}[0,1]) = \lambda(\mathcal{Z}[0,1])$,)

$$= \lambda(\mathcal{X}[0,1])\lambda(\mathcal{Z}[0,1]) = \tilde{\pi}(\mathcal{X}\times\mathcal{Z}).$$

For the conditional distribution $\nu(\mathcal{Z}|x)$, we similarly have:

$$\int_{\mathcal{X}}\nu(\mathcal{Z}|x)\tilde{\pi}^{\mathbb{X}}(\mathrm{d}x) = \int_{\mathcal{X}[0,1]}\mathrm{Unif}[0,1](\mathcal{Z})\tilde{\pi}^{\mathbb{X}}(\mathrm{d}x) + \int_{\mathcal{X}\cap\mathbb{Q}[2,3]}\delta_{x-2}(\mathcal{Z})\tilde{\pi}^{\mathbb{X}}(\mathrm{d}x)$$

$$= \lambda(\mathcal{Z}[0,1])\tilde{\pi}^{\mathbb{X}}(\mathcal{X}[0,1]) + \tilde{\pi}^{\mathbb{X}}(\mathcal{X}\cap\mathbb{Q}[2,3]\cap(\mathcal{Z}+2))$$

(Since $\tilde{\pi}^{\mathbb{X}}(\mathcal{X}\cap\mathbb{Q}[2,3]\cap(\mathcal{Z}+2)) = \lambda(\mathcal{X}\cap\mathbb{Q}[2,3]\cap(\mathcal{Z}+2)\cap[0,1]) \leqslant \lambda(\mathbb{Q}) = 0$,)

$$= \lambda(\mathcal{Z}[0,1])\lambda(\mathcal{X}[0,1]) = \tilde{\pi}(\mathcal{X}\times\mathcal{Z}).$$

So $\tilde{\pi}$ is also a compatible joint of $\mu(\cdot|z)$ and $\nu(\cdot|x)$. In this case, the violation set for $\mu(\cdot|z) \ll \xi$, *i.e.* $\mathbb{Q}[0,1]$, has measure zero under $\tilde{\pi}^{\mathbb{Z}}$, and the violation set for $\nu(\cdot|x) \ll \zeta$, *i.e.* $\mathbb{Q}[2,3]$, has measure zero under $\tilde{\pi}^{\mathbb{X}}$. So result **(i)** of the Lemma asserts that $\tilde{\pi} \ll \xi\otimes\zeta$, which aligns with the example. This example shows that although both $\pi$ and $\tilde{\pi}$ are the compatible joint of the same conditionals, they have different absolute continuity. So the condition in result **(i)** of the Lemma requires the knowledge of the marginals $\pi^{\mathbb{Z}}$ and $\pi^{\mathbb{X}}$.

## C.2   Proof of Theorem 2.3

**Example C.3.** Before presenting the proof, we give an example showing that only being the intersection of $\mathcal{W}_{p,q}$ and $\mathcal{W}_{q,p}$ is not sufficient to make a valid support of a compatible joint. The example is illustrated in Fig. 11. The conditionals are uniform on the respective depicted slices, so conditions **(iv)** and **(v)** in the theorem (2.3) are satisfied. The sets $\mathcal{W}_{p,q}$ and $\mathcal{W}_{q,p}$ are depicted in the figure, and their intersection $\mathcal{W}_{p,q}\cap\mathcal{W}_{q,p}$ is the right half. Although on $\mathcal{W}_{p,q}\cap\mathcal{W}_{q,p}$, the conditionals do not render support conflict, the conditional $q(z|x)$ is unnormalized for a given $x$ from the bottom half: it integrates to $1/2$ on $(\mathcal{W}_{p,q}\cap\mathcal{W}_{q,p})_x$. This means

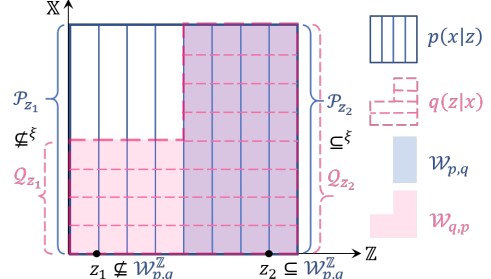

Figure 11: Illustration of Example C.3. The conditionals are uniform on the respective depicted slices. On $\mathcal{W}_{p,q}\cap\mathcal{W}_{q,p}$, the conditional $q(z|x)$ is not normalized.

that any compatible joint $\pi$ would also have its conditional $\pi(\cdot|x)$ unnormalized for such $x$, which is impossible.

One may consider trying $\mathcal{W}_{q,p}$ as the joint support. This renders a support conflict similar to the example in the main text: a joint on $\mathcal{W}_{q,p}$ is required by $p(x|z)$ to also cover the top-left quadrant since $z$ values in the left half are covered, but this contradicts with the absence of mass by $q(z|x)$. In fact, in this example, there is no $\xi \otimes \zeta$-complete component of both $\mathcal{W}_{p,q}$ and $\mathcal{W}_{q,p}$, so the two conditionals are not compatible.

***Proof.*** Let $\mu(\mathcal{X}|z)$ and $\nu(\mathcal{Z}|x)$ be the two everywhere absolutely continuous conditional distributions of whom $p(x|z)$ and $q(z|x)$ are the density functions. We begin with some useful conclusions.

**(1)** By construction, for any $(x,z) \in \mathcal{W}_{p,q}, p(x|z) > 0$. For any $z \in \mathcal{W}_{p,q}^{\mathbb{Z}}$, we have $\xi\{x \in (\mathcal{W}_{p,q})_z \mid q(z|x) = 0\} = \xi\{x \in \mathcal{P}_z \mid x \notin \mathcal{Q}_z\} = \xi(\mathcal{P}_z \setminus \mathcal{Q}_z) = 0$, which means that $q(z|x) > 0$, $\xi$-a.e. on $(\mathcal{W}_{p,q})_z$. By Lemma B.10, we have that $q(z|x) > 0$, $\xi\otimes\zeta$-a.e. on $\mathcal{W}_{p,q}$. Symmetrically, $q(z|x) > 0$ on $\mathcal{W}_{q,p}$, and $p(x|z) > 0$, $\xi\otimes\zeta$-a.e. on $\mathcal{W}_{q,p}$. Particularly, the ratio $\frac{p(x|z)}{q(z|x)}$ is well-defined and is positive and finite, both $\xi\otimes\zeta$-a.e. on $\mathcal{W}_{p,q}$ and $\xi\otimes\zeta$-a.e. on $\mathcal{W}_{q,p}$. The conclusions also hold ($\xi\otimes\zeta$-a.e.) on any ($\xi\otimes\zeta$-a.s.) subset of $\mathcal{W}_{p,q}$ or $\mathcal{W}_{q,p}$.

**"Only if" (necessity):**

Let $\pi$ be a compatible joint distribution of such conditional distributions $\mu(\cdot|z)$ and $\nu(\cdot|x)$.

**(2)** Since "for any" indicates "a.e." under any measure, the conditions in Lemma C.1 are satisfied, so we have $\pi \ll \xi\otimes\zeta$. By Lemma B.9, we also have $\pi^{\mathbb{X}} \ll \xi$ and $\pi^{\mathbb{Z}} \ll \zeta$. So there exist density functions (R-N derivatives; 13, Thm. 32.2) $u(x)$ and $v(z)$ such that for any $\mathcal{X} \in \mathscr{X}$ and $\mathcal{Z} \in \mathscr{Z}$, $\pi^{\mathbb{X}}(\mathcal{X}) = \int_{\mathcal{X}} u(x)\xi(\mathrm{d}x)$ and $\pi^{\mathbb{Z}}(\mathcal{Z}) = \int_{\mathcal{Z}} v(z)\zeta(\mathrm{d}z)$. This $u(x)$ is obviously $\xi$-integrable on any measurable subset of $\mathbb{X}$, since the integral is no larger (since $u$ is nonnegative) than $\int_{\mathbb{X}} u(x)\xi(\mathrm{d}x) = 1$ which is finite.

**(3)** By the definition of conditional distribution Eq. (11), for any $\mathcal{X} \times \mathcal{Z} \in \mathscr{X} \times \mathscr{Z}$, we have $\pi(\mathcal{X} \times \mathcal{Z}) = \int_{\mathcal{Z}} \mu(\mathcal{X}|z)\pi^{\mathbb{Z}}(\mathrm{d}z) = \int_{\mathcal{Z}} \mu(\mathcal{X}|z)v(z)\zeta(\mathrm{d}z) = \int_{\mathcal{Z}} \int_{\mathcal{X}} p(x|z)\xi(\mathrm{d}x)v(z)\zeta(\mathrm{d}z) = \int_{\mathcal{X}\times\mathcal{Z}} p(x|z)v(z)(\xi \otimes \zeta)(\mathrm{d}x\mathrm{d}z)$, where in the last equality, we have applied Fubini's theorem Eq. (8) [13, Thm. 18.3]. Similarly, $\pi(\mathcal{X} \times \mathcal{Z}) = \int_{\mathcal{X}\times\mathcal{Z}} q(z|x)u(x)(\xi\otimes\zeta)(\mathrm{d}x\mathrm{d}z)$. Noting that $\mathscr{X} \times \mathscr{Z}$ is the pi-system that generates $\mathscr{X} \otimes \mathscr{Z}$, this indicates that $p(x|z)v(z) = q(z|x)u(x)$, $\xi\otimes\zeta$-a.e. on $\mathbb{X}\times\mathbb{Z}$ [13, Thm. 16.10(iii)][11]. In other words, both $p(x|z)v(z)$ and $q(z|x)u(x)$ are density functions of $\pi$.

**(3.1)** Subsequently, by leveraging Lemma B.10, we have for $\zeta$-a.e. $z$ on $\mathbb{Z}$, $p(x|z)v(z) = q(z|x)u(x)$, $\xi$-a.e. on $\mathbb{X}$.

**(4)** Let $\mathcal{U} := \{x \mid u(x) > 0\}$ and $\mathcal{V} := \{z \mid v(z) > 0\}$, and define:
$$\mathcal{S} := (\mathcal{U}\times\mathcal{V}) \cap \mathcal{W}_{p,q}, \quad \tilde{\mathcal{S}} := (\mathcal{U}\times\mathcal{V}) \cap \mathcal{W}_{q,p}, \tag{15}$$
Since $u$ and $v$ are integrable thus measurable and $\mathbb{R}^{>0}$ is Lebesgue-measurable, we know that $\mathcal{U} \in \mathscr{X}$ and $\mathcal{V} \in \mathscr{Z}$ are also measurable. So $\mathcal{S}$ and $\tilde{\mathcal{S}}$ are measurable.

**(4.1)** We can verify that $\mathcal{S}$ is a $\xi\otimes\zeta$-complete component of $\mathcal{W}_{p,q}$. Since $\mathcal{S} \subseteq \mathcal{W}_{p,q}$, we only need to verify that:
$$(\xi\otimes\zeta)\big(\big[(\mathcal{S}^{\mathbb{X}} \times \mathbb{Z} \cup \mathbb{X} \times \mathcal{S}^{\mathbb{Z}}) \cap \mathcal{W}_{p,q}\big] \setminus \mathcal{S}\big)$$
(Since clearly $\mathcal{S}^{\mathbb{X}} \subseteq \mathcal{U}, \mathcal{S}^{\mathbb{Z}} \subseteq \mathcal{V}$, and measures are monotone,)
$$\leqslant (\xi\otimes\zeta)\big(\big[(\mathcal{U}\times\mathbb{Z} \cup \mathbb{X}\times\mathcal{V}) \cap \mathcal{W}_{p,q}\big] \setminus \mathcal{S}\big) \quad \text{(Since } (\mathcal{A}\cap\mathcal{C}) \setminus (\mathcal{B}\cap\mathcal{C}) = (\mathcal{A} \setminus \mathcal{B})\cap\mathcal{C}\text{,)}$$
$$= (\xi\otimes\zeta)\big(\big[(\mathcal{U}\times\mathbb{Z} \cup \mathbb{X}\times\mathcal{V}) \setminus (\mathcal{U}\times\mathcal{V})\big] \cap \mathcal{W}_{p,q}\big) \quad \text{(Since } (\mathcal{A}\cup\mathcal{B}) \setminus \mathcal{C} = (\mathcal{A} \setminus \mathcal{C})\cup(\mathcal{B} \setminus \mathcal{C})\text{,)}$$
$$= (\xi\otimes\zeta)\big(\big[((\mathcal{U}\times\mathbb{Z}) \setminus (\mathcal{U}\times\mathcal{V})) \cup ((\mathbb{X}\times\mathcal{V}) \setminus (\mathcal{U}\times\mathcal{V}))\big] \cap \mathcal{W}_{p,q}\big)$$
$$= (\xi\otimes\zeta)\big(\big[\{(x,z) \mid u(x) > 0, v(z) = 0\} \cup \mathcal{W}_{p,q}\big] \cup \big[\{(x,z) \mid u(x) = 0, v(z) > 0\} \cup \mathcal{W}_{p,q}\big]\big)$$

---

[11]The requirement that $\mathbb{X}\times\mathbb{Z}$ is a finite or countable union of $\mathscr{X} \times \mathscr{Z}$-sets is satisfied, since from the sigma-finiteness of $\xi$ and $\zeta$, $\mathbb{X}$ and $\mathbb{Z}$ are a finite or countable union of ($\xi$-finite) $\mathscr{X}$-sets and ($\zeta$-finite) $\mathscr{Z}$-sets, respectively.

(By Conclusion **(3)** and adjusting the set by a set of $\xi\otimes\zeta$-measure-zero according to Lemma B.1,)

$$= (\xi\otimes\zeta)\big(\big[\{(x,z) \mid u(x) > 0, v(z) = 0, q(z|x) = 0\} \cup \mathcal{W}_{p,q}\big]$$

$$\cup \big[\{(x,z) \mid u(x) = 0, v(z) > 0, p(x|z) = 0\} \cup \mathcal{W}_{p,q}\big]\big)$$

$$\leqslant (\xi\otimes\zeta)\big(\{(x,z) \mid u(x) > 0, v(z) = 0, q(z|x) = 0\} \cup \mathcal{W}_{p,q}\big)$$

$$+ (\xi\otimes\zeta)\big(\{(x,z) \mid u(x) = 0, v(z) > 0, p(x|z) = 0\} \cup \mathcal{W}_{p,q}\big) \qquad \text{(By Conclusion (1),)}$$

$= 0$. Symmetrically, we can also verify that $\tilde{\mathcal{S}}$ is a $\xi\otimes\zeta$-complete component of $\mathcal{W}_{q,p}$.

**(4.2)** We can also show that $\pi(\mathcal{S}) = \pi(\tilde{\mathcal{S}}) = 1$. From Conclusion **(3)**, we have:

$$1 = \pi(\mathbb{X}\times\mathbb{Z}) = \int_{\mathbb{Z}}\int_{\mathbb{X}} p(x|z)\xi(\mathrm{d}x)v(z)\zeta(\mathrm{d}z)$$

(Since the integral on a region with an a.e. zero value is zero [13, Thm. 15.2(i)],)

$$= \int_{\mathcal{V}}\int_{\mathcal{P}_z} p(x|z)\xi(\mathrm{d}x)v(z)\zeta(\mathrm{d}z) \qquad \text{(Since } \mathcal{S}^{\mathbb{Z}} \subseteq \mathcal{V} \text{ and due to Billingsley [13, Thm. 16.9],)}$$

$$= \Big(\int_{\mathcal{S}^{\mathbb{Z}}}\int_{\mathcal{P}_z} + \int_{\mathcal{V}\setminus\mathcal{S}^{\mathbb{Z}}}\int_{\mathcal{P}_z}\Big) p(x|z)\xi(\mathrm{d}x)v(z)\zeta(\mathrm{d}z) \quad \text{(Since } \mathcal{V}\setminus\mathcal{S}^{\mathbb{Z}} = \mathcal{V}\setminus(\mathcal{V}\cap\mathcal{W}_{p,q}^{\mathbb{Z}}) = \mathcal{V}\setminus\mathcal{W}_{p,q}^{\mathbb{Z}},)$$

$$= \Big(\int_{\mathcal{S}^{\mathbb{Z}}}\int_{\mathcal{P}_z} + \int_{\mathcal{V}\setminus\mathcal{W}_{p,q}^{\mathbb{Z}}}\int_{\mathcal{P}_z}\Big) p(x|z)\xi(\mathrm{d}x)v(z)\zeta(\mathrm{d}z).$$

For the second iterated integral, note that for any $z$ on $\mathcal{V}$, $v(z) > 0$, so from Conclusion **(3.1)**, for $\zeta$-a.e. $z$ on $\mathcal{V}$, we have $p(x|z) > 0 \Longrightarrow q(z|x) > 0$, $\xi$-a.e. on $\mathbb{X}$. This means that for $\zeta$-a.e. $z$ on $\mathcal{V}$, $\xi\{x \mid p(x|z) > 0, q(z|x) = 0\} = \xi(\mathcal{P}_z \setminus \mathcal{Q}_z) = 0$. So the set $\mathcal{V}\setminus\mathcal{W}_{p,q}^{\mathbb{Z}} = \{z \in \mathcal{V} \mid \xi(\mathcal{P}_z \setminus \mathcal{Q}_z) > 0 \text{ or } \mathcal{P}_z = \emptyset\}$ has the same measure under $\zeta$ as the set $\{z \in \mathcal{V} \mid \mathcal{P}_z = \emptyset\}$. This means that the second iterated integral $\int_{\mathcal{V}\setminus\mathcal{W}_{p,q}^{\mathbb{Z}}}\int_{\mathcal{P}_z} p(x|z)\xi(\mathrm{d}x)v(z)\zeta(\mathrm{d}z) = \int_{\{z\in\mathcal{V}\mid\mathcal{P}_z=\emptyset\}}\int_{\mathcal{P}_z} p(x|z)\xi(\mathrm{d}x)v(z)\zeta(\mathrm{d}z) = 0$ [13, p.226, Thm. 15.2(i)].

For the first iterated integral, note that by construction, $\mathcal{P}_z = (\mathcal{W}_{p,q})_z$ for any $z$ on $\mathcal{S}^{\mathbb{Z}}$, since $\mathcal{S}^{\mathbb{Z}} \subseteq \mathcal{W}_{p,q}^{\mathbb{Z}}$. Moreover, from Conclusion **(4.1)** that $\mathcal{S}$ is a $\xi\otimes\zeta$-complete component of $\mathcal{W}_{p,q}$, we can subsequently apply Lemma B.16, so $\int_{\mathcal{S}^{\mathbb{Z}}}\int_{(\mathcal{W}_{p,q})_z} p(x|z)\xi(\mathrm{d}x)v(z)\zeta(\mathrm{d}z) = \int_{\mathcal{S}} p(x|z)v(z)(\xi\otimes\zeta)(\mathrm{d}x\mathrm{d}z)$, which is $\pi(\mathcal{S})$ by Conclusion **(3)**. This means that $\pi(\mathcal{S}) = 1$. The same deduction applies symmetrically to $\tilde{\mathcal{S}}$, so we also have $\pi(\tilde{\mathcal{S}}) = 1$.

**Main procedure ("only if").** We will verify that the set $\mathcal{S}$ given in Eq. (15) satisfies all the necessary conditions.

Conclusion **(4.2)** shows that $\pi(\mathcal{S}) = 1 > 0$, and in Conclusion **(2)** we have verified that $\pi \ll \xi\otimes\zeta$. So we have $(\xi\otimes\zeta)(\mathcal{S}) > 0$, which verifies Condition **(iii)**.

To verify Conditions **(iv)** and **(v)**, by Conclusions **(3)** and **(1)**, we have $p(x|z)v(z) = q(z|x)u(x)$ and $q(z|x) > 0$, $\xi\otimes\zeta$-a.e. on $\mathcal{S}$. By the construction of $\mathcal{S}$, we also have $v(z) > 0$ and $p(x|z) > 0$ everywhere on $\mathcal{S}$. So the ratio $\frac{p(x|z)}{q(z|x)}$ is finite and positive, $\xi\otimes\zeta$-a.e. on $\mathcal{S}$, and it factorizes as $\frac{p(x|z)}{q(z|x)} = u(x)\frac{1}{v(z)}$, $\xi\otimes\zeta$-a.e. on $\mathcal{S}$. By Conclusion **(2)**, $a(x) := u(x)$ is $\xi$-integrable on $\mathcal{S}^{\mathbb{X}}$.

To verify Condition **(ii)**, note that by construction, $\mathcal{S}^{\mathbb{Z}} \subseteq \mathcal{V} \cap \mathcal{W}_{p,q}^{\mathbb{Z}} \subseteq \mathcal{W}_{p,q}^{\mathbb{Z}}$ so $\mathcal{S}^{\mathbb{Z}} \subseteq^{\zeta} \mathcal{W}_{p,q}^{\mathbb{Z}}$. Note that $\mathcal{S}^{\mathbb{X}} = \{x \in \mathcal{U} \mid \mathcal{V} \cap (\mathcal{W}_{p,q})_x \neq \emptyset\} = \{x \in \mathcal{U} \mid \exists z \in \mathcal{V} \text{ s.t. } x \in \mathcal{P}_z, \mathcal{P}_z \subseteq^{\xi} \mathcal{Q}_z\} \subseteq \{x \in \mathcal{U} \mid \exists z \in \mathcal{V} \text{ s.t. } x \in \mathcal{P}_z\} = \{x \mid u(x) > 0, \exists z \text{ s.t. } v(z) > 0, p(x|z) > 0\}$. By Conclusion **(3.1)**, for $\xi$-a.e. $x$ on $\mathcal{S}^{\mathbb{X}}$, we have $\exists z$ s.t. $q(z|x)u(x) = p(x|z)v(z) > 0$, so $q(z|x) > 0$ hence $\mathcal{Q}_x \neq \emptyset$. Moreover, for $\xi$-a.e. $x$ on $\mathcal{S}^{\mathbb{X}}$ and $\zeta$-a.e. $z$ on $\mathcal{Q}_x$, we have $p(x|z)v(z) = q(z|x)u(x) > 0$, so $p(x|z) > 0$ hence $\mathcal{Q}_x \subseteq^{\zeta} \mathcal{P}_x$. These two conclusions means that for $\xi$-a.e. $x$ on $\mathcal{S}^{\mathbb{X}}$, we have $x \in \{x \mid \mathcal{Q}_x \neq \emptyset, \mathcal{Q}_x \subseteq^{\zeta} \mathcal{P}_x\}$ which is exactly $\mathcal{W}_{q,p}^{\mathbb{X}}$. Hence $\mathcal{S}^{\mathbb{X}} \subseteq^{\xi} \mathcal{W}_{q,p}^{\mathbb{X}}$.

Now, all the left is to verify Condition **(i)**. Conclusion **(4.1)** has verified that $\mathcal{S}$ is a $\xi\otimes\zeta$-complete component of $\mathcal{W}_{p,q}$. To verify that $\mathcal{S}$ is a $\xi\otimes\zeta$-complete component also of $\mathcal{W}_{q,p}$, note that Conclusion **(4.1)** has also verified that $\tilde{\mathcal{S}}$ is a $\xi\otimes\zeta$-complete component of $\mathcal{W}_{q,p}$, so by Lemma B.14 it suffices to verify that $\mathcal{S} \overset{\xi\otimes\zeta}{=} \tilde{\mathcal{S}}$, $\mathcal{S}^{\mathbb{X}} \overset{\xi}{=} \tilde{\mathcal{S}}^{\mathbb{X}}$ and $\mathcal{S}^{\mathbb{Z}} \overset{\zeta}{=} \tilde{\mathcal{S}}^{\mathbb{Z}}$.

By construction, for any $(x, z) \in \mathcal{S}$, we have $v(z) > 0$ and $p(x|z) > 0$, so from Conclusion **(3)** the density function $p(x|z)v(z)$ of $\pi$ is positive. Similarly, the density function $q(z|x)u(x)$ of $\pi$ is positive everywhere on $\tilde{\mathcal{S}}$. Moreover, Conclusion **(4.2)** has shown that both $\pi(\mathcal{S}) = 1$ and $\pi(\tilde{\mathcal{S}}) = 1$. So by Lemma B.7, we know that $\mathcal{S} \stackrel{\xi \otimes \zeta}{=} \tilde{\mathcal{S}}$. Also by construction, the density function $u(x)$ of $\pi^{\mathbb{X}}$ is positive everywhere on $\mathcal{S}^{\mathbb{X}}$ and on $\tilde{\mathcal{S}}^{\mathbb{X}}$. Moreover, we have $\pi^{\mathbb{X}}(\mathcal{S}^{\mathbb{X}}) = \pi(\mathcal{S}^{\mathbb{X}} \times \mathbb{Z}) \geqslant \pi(\mathcal{S}) = 1$ so $\pi^{\mathbb{X}}(\mathcal{S}^{\mathbb{X}}) = 1$ and similarly $\pi^{\mathbb{X}}(\tilde{\mathcal{S}}^{\mathbb{X}}) = 1$. Again by Lemma B.7, we have $\mathcal{S}^{\mathbb{X}} \stackrel{\xi}{=} \tilde{\mathcal{S}}^{\mathbb{X}}$. It follows similarly that $\mathcal{S}^{\mathbb{Z}} \stackrel{\zeta}{=} \tilde{\mathcal{S}}^{\mathbb{Z}}$.

**"If" (sufficiency):**

**(5)** For Conditions **(iv)** and **(v)**, denote $\tilde{a}(x) := |a(x)|$ and $\tilde{b}(z) := |b(z)|$, and let $\tilde{A} := \int_{\mathcal{S}^{\mathbb{X}}} \tilde{a}(x)\xi(\mathrm{d}x)$.

**(5.1)** From the definition of integrability in Supplement A.1, Condition **(v)** is equivalent to that $\tilde{a}(x)$ is $\xi$-integrable on $\mathcal{S}^{\mathbb{X}}$. Particularly, $\tilde{A} < \infty$.

Due to Condition **(i)** and Lemma B.12, we have $\mathcal{S} \subseteq^{\xi \otimes \zeta} \mathcal{W}_{p,q}$. So by Condition **(iv)** and Conclusion **(1)**, we have $\frac{p(x|z)}{q(z|x)} = a(x)b(z) > 0$, $\xi \otimes \zeta$-a.e. on $\mathcal{S}$. This means both $(\xi \otimes \zeta)\{(x, z) \in \mathcal{S} \mid a(x)b(z) = 0\} = 0$ and $(\xi \otimes \zeta)\{(x, z) \in \mathcal{S} \mid a(x)b(z) < 0\} = 0$, since their summation is zero.

**(5.2)** Since $\{(x, z) \mid x \in \mathcal{S}^{\mathbb{X}}, a(x) = 0, z \in \mathcal{S}_x\} \subseteq \{(x, z) \in \mathcal{S} \mid a(x)b(z) = 0\}$, the second-last equation in Conclusion **(5.1)** above means that $(\xi \otimes \zeta)\{(x, z) \mid x \in \mathcal{S}^{\mathbb{X}}, a(x) = 0, z \in \mathcal{S}_x\} = 0$. So $(\xi \otimes \zeta)\{(x, z) \mid x \in \mathcal{S}^{\mathbb{X}}, a(x) \neq 0, z \in \mathcal{S}_x\} = (\xi \otimes \zeta)\{(x, z) \mid x \in \mathcal{S}^{\mathbb{X}}, z \in \mathcal{S}_x\} - (\xi \otimes \zeta)\{(x, z) \mid x \in \mathcal{S}^{\mathbb{X}}, a(x) = 0, z \in \mathcal{S}_x\} = (\xi \otimes \zeta)(\mathcal{S}) > 0$ by Condition **(iii)**. Moreover, by Eq. (7), we have $(\xi \otimes \zeta)\{(x, z) \mid x \in \mathcal{S}^{\mathbb{X}}, a(x) \neq 0, z \in \mathcal{S}_x\} = \int_{\{x \in \mathcal{S}^{\mathbb{X}}|a(x) \neq 0\}} \zeta(\mathcal{S}_x)\xi(\mathrm{d}x) > 0$, so $\xi\{x \in \mathcal{S}^{\mathbb{X}} \mid a(x) \neq 0\} > 0$ [13, p.226]. Particularly, $\tilde{A} > 0$ [13, Thm. 15.2(ii)].

**(5.3)** Since $\{(x, z) \in \mathcal{S} \mid a(x)b(z) \neq \tilde{a}(x)\tilde{b}(z)\} = \{(x, z) \in \mathcal{S} \mid a(x)b(z) < 0\}$, the last equation in Conclusion **(5.1)** above means that $a(x)b(z) = \tilde{a}(x)\tilde{b}(z)$, $\xi \otimes \zeta$-a.e. on $\mathcal{S}$. So we have $\frac{p(x|z)}{q(z|x)} = \tilde{a}(x)\tilde{b}(z)$, $\xi \otimes \zeta$-a.e. on $\mathcal{S}$, following Condition **(iv)**.

**(6)** Based on Conclusions **(5.1)** and **(5.2)**, we can define the following finite and nonnegative functions on $\mathbb{X}$ and $\mathbb{Z}$:

$$u(x) := \begin{cases} \frac{1}{\tilde{A}}\tilde{a}(x), & \text{if } x \in \mathcal{S}^{\mathbb{X}} \text{ and } \tilde{a}(x) > 0, \\ 0, & \text{otherwise,} \end{cases} \qquad v(z) := \begin{cases} \frac{1}{\tilde{A}\tilde{b}(z)}, & \text{if } z \in \mathcal{S}^{\mathbb{Z}} \text{ and } \tilde{b}(z) > 0, \\ 0, & \text{otherwise.} \end{cases}$$

**(6.1)** By construction, $\tilde{a}(x)\tilde{b}(z) = u(x)/v(z)$ on $\mathcal{S} \cap \{(x, z) \mid \tilde{b}(z) > 0\}$. So from Conclusion **(5.3)**, we have $p(x|z)v(z) = q(z|x)u(x)$, $\xi \otimes \zeta$-a.e. on $\mathcal{S} \cap \{(x, z) \mid \tilde{b}(z) > 0\}$. Moreover, following a similar deduction as in Conclusion **(5.2)**, we know that $(\xi \otimes \zeta)(\mathcal{S} \setminus \{(x, z) \mid \tilde{b}(z) > 0\}) = (\xi \otimes \zeta)\{(x, z) \mid z \in \mathcal{S}^{\mathbb{Z}}, b(z) = 0, x \in \mathcal{S}_z\} = 0$. So we have $p(x|z)v(z) = q(z|x)u(x)$, $\xi \otimes \zeta$-a.e. on $\mathcal{S}$.

**(6.2)** By construction, $\int_{\mathbb{X}} u(x)\xi(\mathrm{d}x) = \int_{\mathcal{S}^{\mathbb{X}}} u(x)\xi(\mathrm{d}x) = 1$.

**Main procedure ("if").** We will show that the following function on $\mathscr{X} \otimes \mathscr{Z}$, which is the same as Eq. (1) in the theorem, is a distribution on $\mathbb{X} \times \mathbb{Z}$ such that $\mu(\mathcal{X}|z)$ and $\nu(\mathcal{Z}|x)$ are its conditional distributions:

$$\pi(\mathcal{W}) := \int_{\mathcal{W} \cap \mathcal{S}} q(z|x)u(x)(\xi \otimes \zeta)(\mathrm{d}x\mathrm{d}z), \quad \forall \mathcal{W} \in \mathscr{X} \otimes \mathscr{Z}. \tag{16}$$

Consider any measurable rectangle $\mathcal{X} \times \mathcal{Z} \in \mathscr{X} \times \mathscr{Z}$. We have:

$$\pi(\mathcal{X} \times \mathcal{Z}) = \int_{\mathcal{X} \times \mathcal{Z} \cap \mathcal{S}} q(z|x)u(x)(\xi \otimes \zeta)(\mathrm{d}x\mathrm{d}z)$$

(By the generalized form Eq. (9) of Fubini's theorem,)

$$= \int_{\mathcal{X} \cap \mathcal{S}^{\mathbb{X}}} \int_{\mathcal{Z} \cap \mathcal{S}_x} q(z|x)\zeta(\mathrm{d}z)u(x)\xi(\mathrm{d}x) \qquad \text{(By Condition (i) and applying Lemma B.16,)}$$

$$= \int_{\mathcal{X} \cap \mathcal{S}^{\mathbb{X}}} \int_{\mathcal{Z} \cap (\mathcal{W}_{q,p})_x} q(z|x)\zeta(\mathrm{d}z)u(x)\xi(\mathrm{d}x)$$

(Since $\mathcal{S}^{\mathbb{X}} \subseteq^{\xi} \mathcal{W}_{q,p}^{\mathbb{X}}$ by Condition **(ii)** and $(\mathcal{W}_{q,p})_x = \mathcal{Q}_x$ on $\mathcal{W}_{q,p}^{\mathbb{X}}$,)

$$= \int_{\mathcal{X} \cap \mathcal{S}^{\mathbb{X}}} \int_{\mathcal{Z} \cap \mathcal{Q}_x} q(z|x) \zeta(\mathrm{d}z) u(x) \xi(\mathrm{d}x)$$

(Since by construction, $u(x) = 0$ outside $\mathcal{S}^{\mathbb{X}}$ and $q(z|x) = 0$ outside $\mathcal{Q}_x$,)

$$= \int_{\mathcal{X}} \int_{\mathcal{Z}} q(z|x) \zeta(\mathrm{d}z) u(x) \xi(\mathrm{d}x) \qquad \text{(Recalling that } q(z|x) \text{ is the density function of } \nu(\cdot|x),)$$

$$= \int_{\mathcal{X}} \nu(\mathcal{Z}|x) u(x) \xi(\mathrm{d}x). \tag{17}$$

Moreover, due to Conclusion **(6.1)**, we have $\pi(\mathcal{W}) = \int_{\mathcal{W} \cap \mathcal{S}} p(x|z) v(z)(\xi \otimes \zeta)(\mathrm{d}x \mathrm{d}z)$ on $\mathscr{X} \otimes \mathscr{Z}$ [13, Thm. 15.2(v)]. Using this form of $\pi$ and noting that the symmetrized conditions in the above deduction also hold, we have:

$$\pi(\mathcal{X} \times \mathcal{Z}) = \int_{\mathcal{Z}} \mu(\mathcal{X}|z) v(z) \zeta(\mathrm{d}z). \tag{18}$$

Since both $q(z|x)$ and $u(x)$ are finite and nonnegative on $\mathbb{X} \times \mathbb{Z}$, $\pi$ is a measure on $\mathscr{X} \otimes \mathscr{Z}$ by its definition Eq. (16) [13, p.227]. Moreover, from Eq. (17), we have $\pi(\mathbb{X} \times \mathbb{Z}) = \int_{\mathbb{X}} u(x) \xi(\mathrm{d}x) = 1$ by Conclusion **(6.2)**. So $\pi$ is a distribution (probability measure) on $\mathbb{X} \times \mathbb{Z}$.

From Eq. (17), we have $\pi^{\mathbb{X}}(\mathcal{X}) := \pi(\mathcal{X} \times \mathbb{Z}) = \int_{\mathcal{X}} u(x) \xi(\mathrm{d}x)$. So $u(x)$ is a density function of $\pi^{\mathbb{X}}$, and Eq. (17) in turn becomes $\pi(\mathcal{X} \times \mathcal{Z}) = \int_{\mathcal{X}} \nu(\mathcal{Z}|x) \pi^{\mathbb{X}}(\mathrm{d}x)$. This indicates that $\nu(\mathcal{Z}|x)$ is a conditional distribution of $\pi$ w.r.t sub-sigma-field $\mathscr{X} \times \{\mathcal{Z}\}$ for any $\mathcal{Z} \in \mathscr{Z}$, due to Eq. (11).

Similarly, from Eq. (18), we have $\pi^{\mathbb{Z}}(\mathcal{Z}) := \pi(\mathbb{X} \times \mathcal{Z}) = \int_{\mathcal{Z}} v(z) \zeta(\mathrm{d}z)$. So $v(z)$ is a density function of $\pi^{\mathbb{Z}}$, and Eq. (18) in turn becomes $\pi(\mathcal{X} \times \mathcal{Z}) = \int_{\mathcal{Z}} \mu(\mathcal{X}|z) \pi^{\mathbb{Z}}(\mathrm{d}z)$. This indicates that $\mu(\mathcal{X}|z)$ is a conditional distribution of $\pi$ w.r.t sub-sigma-field $\{\mathcal{X}\} \times \mathscr{Z}$ for any $\mathcal{X} \in \mathscr{X}$, due to Eq. (11). The proof is completed. $\qquad \square$

### C.3 Complete Support Proposition under the a.e.-Full Support Condition

**Proposition C.4.** *If $p(x|z)$ and $q(z|x)$ have a.e.-full supports, then $\mathcal{W}_{p,q} \overset{\xi \otimes \zeta}{=} \mathcal{W}_{q,p} \overset{\xi \otimes \zeta}{=} \mathbb{X} \times \mathbb{Z}$, and $\mathbb{X} \times \mathbb{Z}$ is the $\xi \otimes \zeta$-unique complete support of them when compatible.*

**Proof.** By definition, $p(x|z) > 0$ and $q(z|x) > 0$, $\xi \otimes \zeta$-a.e. By Lemma B.10, this means that for $\xi$-a.e. $x$, $p(x|z) > 0$ $\zeta$-a.e. thus $\mathcal{P}_x \overset{\zeta}{=} \mathbb{Z}$, and for $\zeta$-a.e. $z$, $\mathcal{Q}_z \overset{\xi}{=} \mathbb{X}$. Similarly, we also have for $\xi$-a.e. $x$, $\mathcal{Q}_x \overset{\zeta}{=} \mathbb{Z}$ thus $\mathcal{Q}_x \overset{\zeta}{=} \mathcal{P}_x$ by the transitivity (Lemma B.4) and subsequently $\mathcal{Q}_x \subseteq^{\zeta} \mathcal{P}_x$. Similarly, for $\zeta$-a.e. $z$, $\mathcal{P}_z \subseteq^{\xi} \mathcal{Q}_z$. This means that for $\xi$-a.e. $x$, $(\mathcal{W}_{q,p})_x = \mathcal{Q}_x \overset{\zeta}{=} \mathbb{Z}$ thus $z \in (\mathcal{W}_{q,p})_x$, $\zeta$-a.e. By Lemma B.10, this means that for $\xi \otimes \zeta$-a.e. $(x,z)$, we have $(x,z) \in \mathcal{W}_{q,p}$, so $\mathcal{W}_{q,p} \overset{\xi \otimes \zeta}{=} \mathbb{X} \times \mathbb{Z}$. Similarly, we have $\mathcal{W}_{p,q} \overset{\xi \otimes \zeta}{=} \mathbb{X} \times \mathbb{Z}$.

Let $\mathcal{S}$ be a complete support of $p(x|z)$ and $q(z|x)$ when they are compatible. Since $(\xi \otimes \zeta)(\mathcal{S}) = 1 > 0$ from Condition **(iii)** in Theorem 2.3, we know that $\xi(\mathcal{S}^{\mathbb{X}}) > 0$ by Eq. (7) and Billingsley [13, p.226]. So there is an $x \in \mathcal{S}^{\mathbb{X}}$ such that $(\mathcal{W}_{q,p})_x \overset{\zeta}{=} \mathbb{Z}$, otherwise there would be a non-measure-zero set of $x$ violating $(\mathcal{W}_{q,p})_x \overset{\zeta}{=} \mathbb{Z}$. As a $\xi \otimes \zeta$-complete component of $\mathcal{W}_{q,p}$ by Condition **(i)** in Theorem 2.3, we have $\mathcal{S} \overset{\xi \otimes \zeta}{=} (\mathcal{S}^{\mathbb{X}} \times \mathbb{Z} \cap \mathcal{W}_{q,p}) \cup (\mathbb{Z} \times \mathcal{S}^{\mathbb{Z}} \cap \mathcal{W}_{q,p}) \supseteq \mathcal{S}^{\mathbb{X}} \times \mathbb{Z} \cap \mathcal{W}_{q,p} \overset{\xi \otimes \zeta}{=} \mathcal{S}^{\mathbb{X}} \times \mathbb{Z}$. This means that $(\xi \otimes \zeta)(\mathcal{S}^{\mathbb{X}} \times \mathbb{Z} \setminus \mathcal{S}) = \int_{\mathcal{S}^{\mathbb{X}}} \zeta(\mathbb{Z} \setminus \mathcal{S}_x) \xi(\mathrm{d}x) = 0$ by Eq. (7), so for $\xi$-a.e. $x$ on $\mathcal{S}^{\mathbb{X}}$, $\zeta(\mathbb{Z} \setminus \mathcal{S}_x) = 0$ [13, Thm. 15.2(ii)] thus $\mathcal{S}_x \overset{\zeta}{=} \mathbb{Z}$. So $\mathcal{S}^{\mathbb{Z}} \overset{\zeta}{=} \mathbb{Z}$. Moreover, we also have $\mathcal{S} \supseteq^{\xi \otimes \zeta} \mathbb{X} \times \mathcal{S}^{\mathbb{Z}}$ which $\overset{\xi \otimes \zeta}{=} \mathbb{X} \times \mathbb{Z}$, so we have $\mathcal{S} \overset{\xi \otimes \zeta}{=} \mathbb{X} \times \mathbb{Z}$. Similarly, from that $\mathcal{S}$ is a $\xi \otimes \zeta$-complete component also of $\mathcal{W}_{p,q}$, we have the same conclusion. $\qquad \square$

### C.4 Proof of Theorem 2.4

**Proof.** Let $\pi$ and $\tilde{\pi}$ be two compatible joints of $p(x|z)$ and $q(z|x)$, and they are supported on the same complete support $\mathcal{S}$. By Conclusions **(2)** and **(3)** in the proof (Supplement C.2) of Theorem 2.3, there exist functions $u(x), v(z)$ and $\tilde{u}(x), \tilde{v}(z)$ such that $p(x|z)v(z)$ and $q(z|x)u(x)$ are the densities of $\pi$, and $p(x|z)\tilde{v}(z)$ and $q(z|x)\tilde{u}(x)$ of $\tilde{\pi}$, and $p(x|z)v(z) = q(z|x)u(x)$ and $p(x|z)\tilde{v}(z) = q(z|x)\tilde{u}(x)$,

$\xi \otimes \zeta$-a.e. By the definition of a support in Lemma B.7, we know that the densities of $\pi$ and $\tilde{\pi}$ are positive $\xi \otimes \zeta$-a.e. on $\mathcal{S}$, and $\int_{\mathcal{S}^{\mathbb{X}}} u \, d\xi = \int_{\mathcal{S}^{\mathbb{X}}} \tilde{u} \, d\xi = \int_{\mathcal{S}^{\mathbb{Z}}} v \, d\zeta = \int_{\mathcal{S}^{\mathbb{Z}}} \tilde{v} \, d\zeta = 1$.

Consequently, we have $\frac{p(x|z)}{q(z|x)} = \frac{u(x)}{v(z)} = \frac{\tilde{u}(x)}{\tilde{v}(z)}$, $\xi \otimes \zeta$-a.e. on $\mathcal{S}$. By Lemma B.10, for $\zeta$-a.e. $z$ on $\mathcal{S}^{\mathbb{Z}}$, we have $\frac{u(x)}{v(z)} = \frac{\tilde{u}(x)}{\tilde{v}(z)}$ for $\xi$-a.e. $x$ on $\mathcal{S}_z$, which means that $\int_{\mathcal{S}_z} \frac{u(x)}{v(z)} \xi(dx) = \int_{\mathcal{S}_z} \frac{\tilde{u}(x)}{\tilde{v}(z)} \xi(dx)$. Since for $\zeta$-a.e. $z$ on $\mathcal{S}^{\mathbb{Z}}$, $\mathcal{S}_z \overset{\xi}{=} \mathcal{S}^{\mathbb{X}}$, we have by Lemma B.3 that $\int_{\mathcal{S}^{\mathbb{X}}} \frac{u(x)}{v(z)} \xi(dx) = \int_{\mathcal{S}^{\mathbb{X}}} \frac{\tilde{u}(x)}{\tilde{v}(z)} \xi(dx)$, which in turn gives $\frac{1}{v(z)} \int_{\mathcal{S}^{\mathbb{X}}} u \, d\xi = \frac{1}{v(z)} = \frac{1}{\tilde{v}(z)} \int_{\mathcal{S}^{\mathbb{X}}} \tilde{u} \, d\xi = \frac{1}{\tilde{v}(z)}$. So $v(z) = \tilde{v}(z)$ for $\zeta$-a.e. $z$ on $\mathcal{S}^{\mathbb{Z}}$, and similarly $u(x) = \tilde{u}(x)$ for $\xi$-a.e. $x$ on $\mathcal{S}^{\mathbb{X}}$. Subsequently, the density $p(x|z)v(z)$ or $q(z|x)u(x)$ of $\pi$ is $\xi \otimes \zeta$-a.e. the same as the density $p(x|z)\tilde{v}(z)$ or $q(z|x)\tilde{u}(x)$ of $\tilde{\pi}$. Hence, $\pi$ and $\tilde{\pi}$ are the same distribution. $\qquad\square$

## C.5 The Dirac Compatibility Lemma

Before proving the main instructive compatibility theorem (2.6) for the Dirac case, we first present an existential equivalent criterion for compatibility, which provides insights to the problem.

**Lemma C.5** (Dirac compatibility, existential). *Conditional distribution $\nu(\mathcal{Z}|x)$ is compatible with $\mu(\mathcal{X}|z) := \delta_{f(z)}(\mathcal{X})$ where function $f : \mathbb{Z} \to \mathbb{X}$ is $\mathscr{X}/\mathscr{Z}$-measurable, if and only if there is a distribution $\beta$ on $(\mathbb{Z}, \mathscr{Z})$ such that $\nu(\mathcal{Z}|x) = \frac{d\beta(\mathcal{Z} \cap f^{-1}(\cdot))}{d\beta(f^{-1}(\cdot))}(x)$, and this $\beta$ is the marginal $\pi^{\mathbb{Z}}$ of a compatible joint $\pi$ of them.*

*Proof.* We first show the validity of the R-N derivative. Since $\beta$ is a distribution thus a finite measure, $\beta(\mathcal{Z} \cap f^{-1}(\cdot))$ and $\beta(f^{-1}(\cdot))$ are also finite thus sigma-finite. For any $\mathcal{X} \in \mathscr{X}$ such that $\beta(f^{-1}(\mathcal{X})) = 0$, we have $\beta(\mathcal{Z} \cap f^{-1}(\mathcal{X})) \leqslant \beta(f^{-1}(\mathcal{X})) = 0$ since $\mathcal{Z} \cap f^{-1}(\mathcal{X}) \subseteq f^{-1}(\mathcal{X})$ and measures are monotone. So $\beta(\mathcal{Z} \cap f^{-1}(\mathcal{X})) = 0$ and $\beta(\mathcal{Z} \cap f^{-1}(\cdot)) \ll \beta(f^{-1}(\cdot))$. By the R-N theorem [13, Thm. 32.2], the R-N derivative exists.

**"Only if" (necessity):** Let $\pi$ be a compatible joint. Since $\mu$ and $\nu$ are its conditional distributions, by Eq. (11), we have:

$$\pi(\mathcal{X} \times \mathcal{Z}) = \int_{\mathcal{Z}} \mu(\mathcal{X}|z) \pi^{\mathbb{Z}}(dz) = \int_{\mathcal{X}} \nu(\mathcal{Z}|x) \pi^{\mathbb{X}}(dx), \quad \forall \mathcal{X} \times \mathcal{Z} \in \mathscr{X} \times \mathscr{Z}.$$

The first integral is $\int_{\mathcal{Z}} \mathbb{I}[f(z) \in \mathcal{X}] \pi^{\mathbb{Z}}(dz) = \int_{\mathcal{Z}} \mathbb{I}[z \in f^{-1}(\mathcal{X})] \pi^{\mathbb{Z}}(dz) = \pi^{\mathbb{Z}}(\mathcal{Z} \cap f^{-1}(\mathcal{X}))$. Particularly, $\pi^{\mathbb{X}}(\mathcal{X}) = \pi(\mathcal{X} \times \mathbb{Z}) = \pi^{\mathbb{Z}}(f^{-1}(\mathcal{X}))$, *i.e.* $\pi^{\mathbb{X}}$ is the transformed (pushed-forward) distribution from $\pi^{\mathbb{Z}}$ by measurable function $f$ [13, p.196]. On the other hand, the equality to the second integral means that $\pi^{\mathbb{Z}}(\mathcal{Z} \cap f^{-1}(\mathcal{X})) = \int_{\mathcal{X}} \nu(\mathcal{Z}|x) \pi^{\mathbb{X}}(dx) = \int_{\mathcal{X}} \nu(\mathcal{Z}|x) \pi^{\mathbb{Z}}(f^{-1}(dx))$. This means that $\nu(\mathcal{Z}|x)$ is the R-N derivative of $\mathcal{X} \mapsto \pi^{\mathbb{Z}}(\mathcal{Z} \cap f^{-1}(\mathcal{X}))$ w.r.t $\mathcal{X} \mapsto \pi^{\mathbb{Z}}(f^{-1}(\mathcal{X}))$. Taking $\beta$ as $\pi^{\mathbb{Z}}$, which is a distribution on $(\mathbb{Z}, \mathscr{Z})$, yields the necessary condition.

**"If" (sufficiency):** For any measurable rectangle $\mathcal{X} \times \mathcal{Z} \in \mathscr{X} \times \mathscr{Z}$, define $\pi(\mathcal{X} \times \mathcal{Z}) := \int_{\mathcal{Z}} \mu(\mathcal{X}|z)\beta(dz)$ and $\tilde{\pi}(\mathcal{X} \times \mathcal{Z}) := \beta(\mathcal{Z} \cap f^{-1}(\mathcal{X}))$. Since for any $z \in \mathbb{Z}$, $f(z) \in \mathbb{X}$, so $\pi(\mathbb{X} \times \mathbb{Z}) = \int_{\mathbb{Z}} \mu(\mathbb{X}|z)\beta(dz) = \int_{\mathbb{Z}} \beta(dz) = 1$. Since $f^{-1}(\mathbb{X}) = \mathbb{Z}$, we have $\tilde{\pi}(\mathbb{X} \times \mathbb{Z}) = \beta(\mathbb{Z}) = 1$. So both $\pi$ and $\tilde{\pi}$ are finite thus sigma-finite. Moreover, for any $\mathcal{X} \times \mathcal{Z} \in \mathscr{X} \times \mathscr{Z}$, we have $\pi(\mathcal{X} \times \mathcal{Z}) = \int_{\mathcal{Z}} \mathbb{I}[f(z) \in \mathcal{X}]\beta(dz) = \int_{\mathcal{Z}} \mathbb{I}[z \in f^{-1}(\mathcal{X})]\beta(dz) = \int_{\mathcal{Z} \cap f^{-1}(\mathcal{X})} \beta(dz) = \beta(\mathcal{Z} \cap f^{-1}(\mathcal{X})) = \tilde{\pi}(\mathcal{X} \times \mathcal{Z})$, *i.e.* $\pi$ and $\tilde{\pi}$ agree on the pi-system $\mathscr{X} \times \mathscr{Z}$. So by Billingsley [13, Thm. 10.3], $\pi$ and $\tilde{\pi}$ extend to the same distribution (probability measure) on $(\mathbb{X} \times \mathbb{Z}, \mathscr{X} \otimes \mathscr{Z})$.

On the other hand, we have $\pi^{\mathbb{Z}}(\mathcal{Z}) = \pi(\mathbb{X} \times \mathcal{Z}) = \int_{\mathcal{Z}} \mu(\mathbb{X}|z)\beta(dz) = \int_{\mathcal{Z}} \beta(dz) = \beta(\mathcal{Z})$, and furthermore from this, $\mu$ is a conditional distribution of $\pi$ due to its construction and Eq. (11). Moreover, $\tilde{\pi}^{\mathbb{X}}(\mathcal{X}) = \tilde{\pi}(\mathcal{X} \times \mathbb{Z}) = \beta(\mathbb{Z} \cap f^{-1}(\mathcal{X})) = \beta(f^{-1}(\mathcal{X}))$, and by the definition of $\nu(\mathcal{Z}|x)$ as an R-N derivative, we have $\beta(\mathcal{Z} \cap f^{-1}(\mathcal{X})) = \int_{\mathcal{X}} \nu(\mathcal{Z}|x)\beta(f^{-1}(dx))$, which is $\tilde{\pi}(\mathcal{X} \times \mathcal{Z}) = \int_{\mathcal{X}} \nu(\mathcal{Z}|x)\tilde{\pi}^{\mathbb{X}}(dx)$. So again due to Eq. (11), $\nu$ is a conditional distribution of $\tilde{\pi}$. Since $\pi$ and $\tilde{\pi}$ are the same distribution on $(\mathbb{X} \times \mathbb{Z}, \mathscr{X} \otimes \mathscr{Z})$, we know that $\mu$ and $\nu$ are compatible. $\qquad\square$

**Key insights.** Let $\pi$ be a compatible joint of $\mu(\mathcal{X}|z) := \delta_{f(z)}(\mathcal{X})$ and $\nu(\mathcal{Z}|x)$. For any $\mathcal{X} \times \mathcal{Z} \in \mathscr{X} \times \mathscr{Z}$, we have:

$$\pi(\mathcal{X} \times \mathcal{Z}) = \pi^{\mathbb{Z}}(\mathcal{Z} \cap f^{-1}(\mathcal{X})) = \int_{\mathcal{X}} \nu(\mathcal{Z}|x)\pi^{\mathbb{Z}}(f^{-1}(dx)) = \int_{f^{-1}(\mathcal{X})} \nu(\mathcal{Z}|f(z))\pi^{\mathbb{Z}}(dz),$$

where the last equality holds due to the rule of change of variables [13, Thm. 16.13]. Let $f^{-1}(\mathscr{X}) := \sigma(\{f^{-1}(\mathcal{X}) \mid \mathcal{X} \in \mathscr{X}\})$ be the pulled-back sigma-field from $\mathscr{X}$ by $f$. It is a sub-sigma-field of $\mathscr{Z}$ as every $f^{-1}(\mathcal{X}) \in \mathscr{Z}$ since $f$ is measurable. So the last equality means that:

$$\nu(\mathcal{Z}|f(z)) = \left.\frac{\mathrm{d}\pi^{\mathbb{Z}}(\mathcal{Z} \cap \cdot)}{\mathrm{d}\pi^{\mathbb{Z}}(\cdot)}\right|_{f^{-1}(\mathscr{X})}(z).$$

The expression on the left makes sense since for all values of $z$ that yield the same value of $f(z)$, the R-N derivative is the same. The second equality also gives:

$$\nu(\mathcal{Z}|x) = \left.\frac{\mathrm{d}\pi^{\mathbb{Z}}(\mathcal{Z} \cap f^{-1}(\cdot))}{\mathrm{d}\pi^{\mathbb{Z}}(f^{-1}(\cdot))}\right|_{\mathscr{X}}(x).$$

### C.6 Proof of Theorem 2.6

***Proof.*** **"Only if" (necessity):** Suppose that $\nu(\mathcal{Z}|x)$ and $\mu(\mathcal{X}|z) := \delta_{f(z)}(\mathcal{X})$ are compatible but for *any* $x \in \mathbb{X}$, $\nu(f^{-1}(\{x\})|x) < 1$. Consider the set $\mathcal{S} := \{(f(z), z) \mid z \in \mathbb{Z}\}$. Since $f$ is $\mathscr{X}/\mathscr{Z}$-measurable, this set $\mathcal{S}$ is $\mathscr{X} \otimes \mathscr{Z}$-measurable. It is also easy to verify that $\mathcal{S}_z = \{f(z)\}$ and $\mathcal{S}_x = f^{-1}(\{x\})$. Now let $\pi$ be *any* of their compatible joint distribution. From Eq. (12), we know that $\pi(\mathcal{S}) = \int_{\mathbb{Z}} \mu(\mathcal{S}_z|z)\pi^{\mathbb{Z}}(\mathrm{d}z) = \int_{\mathbb{Z}} \delta_{f(z)}(\{f(z)\})\pi^{\mathbb{Z}}(\mathrm{d}z) = \int_{\mathbb{Z}} \pi^{\mathbb{Z}}(\mathrm{d}z) = \pi^{\mathbb{Z}}(\mathbb{Z}) = 1$. On the other hand, also from Eq. (12) and due to the compatibility, we have $\pi(\mathcal{S}) = \int_{\mathbb{X}} \nu(\mathcal{S}_x|x)\pi^{\mathbb{X}}(\mathrm{d}x) = \int_{\mathbb{X}} \nu(f^{-1}(\{x\})|x)\pi^{\mathbb{X}}(\mathrm{d}x) < \int_{\mathbb{X}} \pi^{\mathbb{X}}(\mathrm{d}x) = \pi^{\mathbb{X}}(\mathbb{X}) = 1$, which leads to a contradiction. So if $\nu(\mathcal{Z}|x)$ and $\mu(\mathcal{X}|z)$ are compatible, then there is $x_0 \in \mathbb{X}$ such that $\nu(f^{-1}(\{x_0\})|x_0) = 1$.

**"If" (sufficiency):** Let $\beta(\mathcal{Z}) := \nu(f^{-1}(\{x_0\}) \cap \mathcal{Z}|x_0)$ be a set function on $\mathscr{Z}$. We can verify that this $\beta$ is a distribution (probability measure) on $(\mathbb{Z}, \mathscr{Z})$ since $\nu(\cdot|x_0)$ is. Particularly, since $f$ is $\mathscr{X}/\mathscr{Z}$-measurable and $\{x_0\} \in \mathscr{X}$ due to the assumption, we know that $f^{-1}(\{x_0\})$ thus $f^{-1}(\{x_0\}) \cap \mathcal{Z}$ for any $\mathcal{Z} \in \mathscr{Z}$ are in $\mathscr{Z}$; $\beta(\emptyset) = \nu(\emptyset|x_0) = 0$; $\beta(\mathbb{Z}) = \nu(f^{-1}(\{x_0\})|x_0) = 1$ according to the assumption; for any countable disjoint $\mathscr{Z}$-sets $\mathcal{Z}^{(1)}, \mathcal{Z}^{(2)}, \cdots$, it holds that $\mathcal{Z}^{(1)} \cap f^{-1}(\{x_0\}), \mathcal{Z}^{(2)} \cap f^{-1}(\{x_0\}), \cdots$ are also disjoint $\mathscr{Z}$-sets, so $\beta(\bigcup_{i=1}^{\infty} \mathcal{Z}^{(i)}) = \nu(f^{-1}(\{x_0\}) \cap \bigcup_{i=1}^{\infty} \mathcal{Z}^{(i)}|x_0) = \nu(\bigcup_{i=1}^{\infty} f^{-1}(\{x_0\}) \cap \mathcal{Z}^{(i)}|x_0) = \sum_{i=1}^{\infty} \nu(f^{-1}(\{x_0\}) \cap \mathcal{Z}^{(i)}|x_0) = \sum_{i=1}^{\infty} \beta(\mathcal{Z}^{(i)})$.

Now we prove that $\beta(\mathcal{Z} \cap f^{-1}(\mathcal{X})) = \int_{\mathcal{X}} \nu(\mathcal{Z}|x)\beta(f^{-1}(\mathrm{d}x)), \forall \mathcal{X} \times \mathcal{Z} \in \mathscr{X} \times \mathscr{Z}$ which is sufficient due to Lemma C.5. For any $\mathcal{X} \times \mathcal{Z} \in \mathscr{X} \times \mathscr{Z}$, the l.h.s is $\beta(\mathcal{Z} \cap f^{-1}(\mathcal{X})) = \nu(f^{-1}(\{x_0\}) \cap f^{-1}(\mathcal{X}) \cap \mathcal{Z}|x_0) = \nu(f^{-1}(\{x_0\}) \cap \mathcal{Z}|x_0)\mathbb{I}[x_0 \in \mathcal{X}]$, where the last equality holds since $z \in f^{-1}(\{x_0\}) \cap f^{-1}(\mathcal{X})$ if and only if $f(z) = x_0 \in \mathcal{X}$. The integral on the r.h.s is $\int_{\mathcal{X}} \nu(\mathcal{Z}|x)\nu(f^{-1}(\{x_0\}) \cap f^{-1}(\mathrm{d}x)|x_0)$. Since the measure $\mathcal{X} \mapsto \nu(f^{-1}(\{x_0\}) \cap f^{-1}(\mathcal{X})|x_0)$ is zero on the set $\mathcal{X} \setminus \{x_0\}$ (if there is any $z \in f^{-1}(\{x_0\}) \cap f^{-1}(\mathcal{X} \setminus \{x_0\})$, then we have $f(z) = x_0$ and $f(z) \in \mathcal{X} \setminus \{x_0\}$, which is a contradiction), the integral can be reduced on $\{x_0\} \cap \mathcal{X}$ [13, Thm. 16.9]: $\int_{\{x_0\} \cap \mathcal{X}} \nu(\mathcal{Z}|x)\nu(f^{-1}(\{x_0\}) \cap f^{-1}(\mathrm{d}x)|x_0) = \mathbb{I}[x_0 \in \mathcal{X}]\nu(\mathcal{Z}|x_0)\nu(f^{-1}(\{x_0\})|x_0) = \nu(\mathcal{Z}|x_0)\mathbb{I}[x_0 \in \mathcal{X}]$. Moreover, $\nu(\mathcal{Z}|x_0) = \nu(\mathcal{Z} \cap f^{-1}(\{x_0\})|x_0) + \nu(\mathcal{Z} \setminus f^{-1}(\{x_0\})|x_0)$ where $\nu(\mathcal{Z} \setminus f^{-1}(\{x_0\})|x_0) \leqslant \nu(\mathbb{Z} \setminus f^{-1}(\{x_0\})|x_0) = 1 - \nu(f^{-1}(\{x_0\})|x_0) = 0$, we have $\nu(\mathcal{Z} \setminus f^{-1}(\{x_0\})|x_0) = 0$ and $\nu(\mathcal{Z}|x_0) = \nu(\mathcal{Z} \cap f^{-1}(\{x_0\})|x_0)$. So the integral on the r.h.s is $\nu(\mathcal{Z} \cap f^{-1}(\{x_0\})|x_0)\mathbb{I}[x_0 \in \mathcal{X}]$, which is the same as the l.h.s. So the equality is verified. $\square$

## D Topics on the Methods of CyGen

### D.1 Relation to other auto-encoder regularizations

There are methods that consider regularizing the standard auto-encoder (AE) [71, 7] with deterministic encoder $g(x)$ and decoder $f(z)$ for certain robustness. These regularizations are introduced in addition to the standard AE loss, *i.e.* the reconstruction loss: $\mathbb{E}_{p^*(x)}\ell(x, f(g(x)))$, where $\ell(x, x')$ is a measure of similarity between $x$ and $x'$. If $\ell(x, f(z))$ can be treated as a (scaled) negative log-likelihood $-\log p(x|z)$ on $\mathbb{X}$ (*e.g.*, squared 2-norm $\ell$ for a Gaussian $p(x|z)$, cross entropy $\ell$ for a Bernoulli/categorical $p(x|z)$), then we can adopt a distributional view of the decoder as $p(x|z)$ and the encoder as $\delta_{g(x)}(z)$ [12], and reformulate the reconstruction loss also under the distributional view: $\mathbb{E}_{p^*(x)}[-\log p(x|g(x))] = \mathbb{E}_{p^*(x)\delta_{g(x)}(z)}[-\log p(x|z)]$.

---

[12]This is the notation of a Dirac's delta function, which is not a function in the usual sense. We adopt this form for the similarity to the DAE loss.

**Comparison with Jacobian norm regularizations.** Contractive AE (CAE) [68, 69] regularizes the Jacobian norm of the encoder, $\lambda \mathbb{E}_{p^*(x)} \big\| \nabla_x g^\top(x) \big\|_F^2$ ($\lambda$ controls the scale), in hope to encourage the robustness of the encoded representation against local changes around training data. When it is combined with the reconstruction loss which preserves data variation in the representation for reconstruction, the robustness is confined to the orthogonal direction to the data manifold, which often does not reflect semantic meanings of interest. In other words, the variation in this orthogonal direction is contracted in the representation, hence the name. When applied to a linear encoder, this becomes the well-known weight-decay regularizer. Note that CAE does not have a generative modeling utility, as it uses a deterministic encoder which leads to insufficient determinacy (Sec. 2.2.2).

Denoising AE (DAE) [82, 9, 10] considers the robustness to random corruption/perturbation on data, so its encoding process is $z = g(x + \epsilon_e)$ where $\epsilon_e \sim \mathcal{N}(0, \sigma_e^2 I_{d_{\mathbb{X}}})$ (or any other distribution with $\mathbb{E}[\epsilon_e] = 0$ and $\mathrm{Var}[\epsilon_e] = \sigma_e^2 I_{d_{\mathbb{X}}}$), which defines a probabilistic encoder $q(z|x)$ (note that this is different from an additive Gaussian encoder). The goal for training a DAE is thus to try to reconstruct the input under the random corruption, by minimizing the DAE loss:

$$\mathbb{E}_{p^*(x)q(z|x)}[-\log p(x|z)], \tag{19}$$

which resembles the distributional form of the standard reconstruction loss. For infinitesimal corruption variance $\sigma_e^2$ and squared 2-norm $\ell$, the DAE loss Eq. (19) is roughly equivalent to regularizing the standard reconstruction loss with $\sigma_e^2 \mathbb{E}_{p^*(x)} \big\| \nabla_x (f \circ g)^\top(x) \big\|_F^2$, *i.e.* the Jacobian norm of the reconstruction function [68, 2]. So DAE can be viewed to promote the robustness of reconstruction while CAE of the representation [68].

In contrast, for additive Gaussian decoder (*i.e.*, squared 2-norm $\ell$) and encoder, our compatibility regularization Eq. (2) is $\mathbb{E}_{\rho(x,z)} \big\| \frac{1}{\sigma_d^2} \big( \nabla_z f^\top(z) \big)^\top - \frac{1}{\sigma_e^2} \nabla_x g^\top(x) \big\|_F^2$, which is different from CAE and DAE regularizations. Ideologically, the compatibility loss is an intrinsic constraint to make use of the distributional nature of the encoder and decoder, and is not motivated from the additional requirement of robustness in some sense.

**Comparison with a more accurate DAE reformulation.** In fact, the analysis in [68, 2] for DAE as a regularization of the reconstruction loss is inaccurate. Key ingredients for the analysis are the Taylor expansions: $\|x + \varepsilon\|_2^2 = \|x\|_2^2 + 2x^\top \varepsilon + \varepsilon^\top \varepsilon$, $\exp\{x + \varepsilon\} = \exp\{x\}(1 + \varepsilon + \frac{1}{2}\varepsilon^2) + o(\varepsilon^2)$, and $\log(1 + \varepsilon) = \varepsilon - \frac{1}{2}\varepsilon^2 + o(\varepsilon^2)$. In the following, we consider $\ell(x, x') = \|x - x'\|_2^2$, corresponding to an additive Gaussian decoder $p(x|z)$.

First consider the additive Gaussian encoder, $q(z|x) = \mathcal{N}(z|g(x), \sigma_e^2 I_{d_{\mathbb{Z}}})$, or $z = g(x) + \epsilon_e, \epsilon_e \sim \mathcal{N}(0, \sigma_e^2 I_{d_{\mathbb{Z}}})$. Consider the case for infinitesimal $\sigma_e$. For the DAE loss Eq. (19), we have (omitting the expectation over $p^*(x)$):

$$\mathbb{E}_{q(z|x)}[-\log p(x|z)] = \mathbb{E}_{q(z|x)}[\ell(x, f(z))] = \mathbb{E}_{q(z|x)} \|x - f(z)\|_2^2 = \mathbb{E}_{p(\epsilon_e)} \|x - f(g(x) + \epsilon_e)\|_2^2$$

$$= \mathbb{E}_{p(\epsilon_e)} \left[ \left\| x - f(g(x)) - (\nabla f^\top)^\top \epsilon_e - \frac{1}{2}\epsilon_e^\top (\nabla^2 f)\epsilon_e + o(\epsilon_e^2) \right\|_2^2 \right]$$

$$= \mathbb{E}_{p(\epsilon_e)} \left[ \|x - f(g(x))\|_2^2 - 2\big(x - f(g(x))\big)^\top \left( (\nabla f^\top)^\top \epsilon_e + \frac{1}{2}\epsilon_e^\top (\nabla^2 f)\epsilon_e \right) \right.$$

$$\left. + \epsilon_e^\top (\nabla f^\top)(\nabla f^\top)^\top \epsilon_e + o(\epsilon_e^2) \right]$$

$$= \ell\big(x, f(g(x))\big) - \sigma_e^2 \big(x - f(g(x))\big)^\top \Delta f + \sigma_e^2 \big\| \nabla f^\top \big\|_F^2 + o(\sigma_e^2), \tag{20}$$

where $(\nabla f^\top)^\top$ is the Jacobian of $f$, $(\epsilon_e^\top (\nabla^2 f)\epsilon_e)_i := \sum_{j,k=1..d_{\mathbb{Z}}} (\epsilon_e)_i (\epsilon_e)_j \partial_{z_i} \partial_{z_j} f_i$, and $(\Delta f)_i := \sum_{j=1..d_{\mathbb{Z}}} \partial_{z_j} \partial_{z_j} f_i$, and they are evaluated at $z = g(x)$. Note that in addition to the Jacobian norm regularization term discovered in [68, 2], there is a second regularization term $-\sigma_e^2 \big(x - f(g(x))\big)^\top \Delta f$ that DAE imposes.

For the data-fitting loss of CyGen Eq. (4), a similar approximation can be derived (again, omitting the expectation over $p^*(x)$):

$$\log \mathbb{E}_{q(z|x)}[1/p(x|z)] = \log \mathbb{E}_{q(z|x)} \exp\{-\log p(x|z)\} = \log \mathbb{E}_{q(z|x)} \exp\{\ell(x, f(z))\}$$

$$= \log \mathbb{E}_{q(z|x)} \exp\{\|x - f(z)\|_2^2\} = \log \mathbb{E}_{p(\epsilon_e)} \exp\{\|x - f(g(x) + \epsilon_e)\|_2^2\}$$

$$= \log \mathbb{E}_{p(\epsilon_e)} \exp\left\{ \left\| x - f(g(x)) - (\nabla f^\top)^\top \epsilon_e - \frac{1}{2}\epsilon_e^\top (\nabla^2 f)\epsilon_e + o(\epsilon_e^2) \right\|_2^2 \right\}$$

$$= \log \mathbb{E}_{p(\epsilon_e)} \exp\left\{ \|x - f(g(x))\|_2^2 - 2(x - f(g(x)))^\top \left( (\nabla f^\top)^\top \epsilon_e + \frac{1}{2}\epsilon_e^\top (\nabla^2 f)\epsilon_e \right) \right.$$
$$\left. + \epsilon_e^\top (\nabla f^\top)(\nabla f^\top)^\top \epsilon_e + o(\epsilon_e^2) \right\}$$

$$= \log \mathbb{E}_{p(\epsilon_e)} \left[ \exp\{\|x - f(g(x))\|_2^2\} \left( 1 - 2(x - f(g(x)))^\top \left( (\nabla f^\top)^\top \epsilon_e + \frac{1}{2}\epsilon_e^\top (\nabla^2 f)\epsilon_e \right) \right. \right.$$
$$\left. \left. + \epsilon_e^\top (\nabla f^\top)(\nabla f^\top)^\top \epsilon_e + 2\left( (x - f(g(x)))^\top (\nabla f^\top)^\top \epsilon_e \right)^2 + o(\epsilon_e^2) \right) \right]$$

$$= \log \left[ \exp\{\|x - f(g(x))\|_2^2\} \left( 1 - \sigma_e^2 (x - f(g(x)))^\top \Delta f \right. \right.$$
$$\left. \left. + \sigma_e^2 \|\nabla f^\top\|_F^2 + 2\sigma_e^2 \|(\nabla f^\top)(x - f(g(x)))\|_2^2 + o(\sigma_e^2) \right) \right]$$

$$= \ell(x - f(g(x))) - \sigma_e^2 (x - f(g(x)))^\top \Delta f + \sigma_e^2 \|\nabla f^\top\|_F^2 + 2\sigma_e^2 \|(\nabla f^\top)(x - f(g(x)))\|_2^2 + o(\sigma_e^2).$$

This is different from the regularization interpretation of DAE Eq. (20) as a third regularization term $2\sigma_e^2 \|(\nabla f^\top)(x - f(g(x)))\|_2^2$ is presented.

The compatibility loss Eq. (2) in CyGen becomes: $\mathbb{E}_{\rho(x,z)} \left\| \frac{1}{\sigma_d^2}(\nabla_z f^\top(z))^\top - \frac{1}{\sigma_e^2}\nabla_x g^\top(x) \right\|_F^2$, where $\sigma_d^2$ is the Gaussian variance of the decoder $p(x|z)$ (inverse scale for $\ell(\cdot,\cdot)$). When $\rho(x,z) = p^*(x)q(z|x)$, this can be further reduced to (omitting the expectation over $p^*(x)$):

$$\mathbb{E}_{q(z|x)} \left\| \frac{1}{\sigma_d^2}(\nabla_z f^\top(z))^\top - \frac{1}{\sigma_e^2}\nabla_x g^\top(x) \right\|_F^2 = \mathbb{E}_{p(\epsilon_e)} \left\| \frac{1}{\sigma_d^2}(\nabla f^\top(g(x) + \epsilon_e))^\top - \frac{1}{\sigma_e^2}\nabla g^\top \right\|_F^2$$

$$= \mathbb{E}_{p(\epsilon_e)} \left\| \frac{1}{\sigma_d^2}\left( (\nabla f^\top)^\top + (\nabla^2 f^\top)^\top \epsilon_e + \frac{1}{2}\epsilon_e^\top (\nabla^3 f^\top)^\top \epsilon_e + o(\epsilon_e^2) \right) - \frac{1}{\sigma_e^2}\nabla g^\top \right\|_F^2$$

$$= \left\| \frac{1}{\sigma_d^2}(\nabla f^\top)^\top - \frac{1}{\sigma_e^2}\nabla g^\top \right\|_F^2 + \frac{\sigma_e^2}{\sigma_d^4}(\nabla f^\top) : (\nabla \Delta f^\top) + \frac{\sigma_e^2}{\sigma_d^4}\|\nabla^2 f^\top\|_F^2 + o(\sigma_e^2),$$

where $((\nabla^2 f^\top)^\top \epsilon_e)_{ij} := \sum_{k=1..d_\mathbb{Z}}(\epsilon_e)_k \partial_{z_k}\partial_{z_j} f_i$, $(\epsilon_e^\top (\nabla^3 f^\top)^\top \epsilon_e)_{ij} := \sum_{k,k'=1..d_\mathbb{Z}}(\epsilon_e)_k(\epsilon_e)_{k'}\partial_{z_k}\partial_{z_{k'}}\partial_{z_j} f_i$, and $(\nabla f^\top) : (\nabla \Delta f^\top) := \sum_{\substack{i=1..d_\mathbb{X} \\ j,k=1..d_\mathbb{Z}}}(\partial_{z_j} f_i)(\partial_{z_j}\partial_{z_k}\partial_{z_k} f_i)$, $\|\nabla^2 f^\top\|_F^2 := \sum_{\substack{i=1..d_\mathbb{X} \\ j,k=1..d_\mathbb{Z}}}(\partial_{z_k}\partial_{z_j} f_i)^2$, and all terms are evaluated at $z = g(x)$. This is different from the regularization of CAE and the regularization explanation of DAE.

For the corruption encoder, $z = g(x + \epsilon_e), \epsilon_e \sim \mathcal{N}(0, \sigma_e^2 I_{d_\mathbb{X}})$, approximations of the DAE loss Eq. (19) and the data-fitting loss of CyGen Eq. (4) (i.e., negative data likelihood loss) are similar to the above expansions except that derivatives of $f$ are replaced with those of $f \circ g$. Particularly, from Eq. (20), we find that the conclusion in [68, 2] missed the term $-\sigma_e^2(x - f(g(x)))^\top \Delta(f \circ g)$ that is also of order $\sigma_e^2$. For the compatibility loss, as there is no explicit expression of $\log q(z|x)$ (unless $g(x)$ is invertible), the above expression does not hold. But anyway, it is different from CAE and DAE regularizations.

**Relation to the tied weights trick.** The compatibility loss also explains the "tied weights" trick in AE, which is widely adopted and is vital for the success of AE [63, 82, 81, 68, 2]. The trick is considered when components of $x$ and $z$ are binary, and a one-layer, product-of-Bernoulli encoder $q(z|x) = \prod_{j=1}^{d_\mathbb{Z}} \text{Bern}(z_j|s((W_e)_{j,:}x + (b_e)_j))$ and decoder $p(x|z) = \prod_{i=1}^{d_\mathbb{X}} \text{Bern}(x_i|s((W_d)_{i,:}z + (b_d)_i))$ are used, where $s(l) := 1/(1 + \exp\{-l\})$ denotes the sigmoid activation function. For the encoder, we have $q(z|x) = \frac{\exp\{z^\top(W_e x + b_e)\}}{\prod_{j=1}^{d_\mathbb{Z}}(1 + \exp\{(W_e)_{j,:}x + (b_e)_j\})}$ thus $\log q(z|x) = z^\top(W_e x + b_e) - \sum_{j=1}^{d_\mathbb{Z}} \log(1 + \exp\{(W_e)_{j,:}x + (b_e)_j\})$, so $\nabla_x \nabla_z^\top \log q(z|x) = W_e^\top$. Similarly for the decoder, we have $\nabla_x \nabla_z^\top \log p(x|z) = W_d$. So the compatibility loss Eq. (2) in this case is $\|W_d - W_e^\top\|_F^2$, which leads to $W_d = W_e^\top$ when it is zero. This recovers the tied weight trick.

In this Bernoulli case, the CAE regularizer is $\mathbb{E}_{p^*(x)} \sum_{j=1}^{d_{\mathbb{Z}}} \frac{\exp\{-2((W_e)_{j,:}x+(b_e)_j)\} \sum_{i=1}^{d_{\mathbb{X}}} (W_e)_{ji}^2}{\left(1+\exp\{-((W_e)_{j,:}x+(b_e)_j)\}\right)^4} =$ $\mathbb{E}_{p^*(x)} \sum_{j=1}^{d_{\mathbb{Z}}} s((W_e)_{j,:}x+(b_e)_j)^2 \left(1 - s((W_e)_{j,:}x+(b_e)_j)\right)^2 \sum_{i=1}^{d_{\mathbb{X}}} (W_e)_{ji}^2$ and DAE does not have the Jacobian-norm regularization explanation, so they are different from the compatibility loss.

## D.2 Gradient estimation for flow-based models without tractable inverse

**Flow-based density models.** As the insight we draw from the analysis on Gaussian VAE in Sec. 3.1, it is inappropriate to implement both conditionals $p_\theta(x|z)$, $q_\phi(z|x)$ using additive Gaussian models. So we need more flexible and expressive probabilistic models that also allow explicit density evaluation (so implicit models like GANs are not suitable). Flow-based models [24, 60, 44, 8, 18] are a good choice. They also allow direct sampling with reparameterization for efficiently estimating and optimizing the data-fitting loss Eq. (4) (for which energy-based models are costly), and have been used as the inference model $q_\phi(z|x)$ of VAEs [66, 46, 80, 31]. For a connection to these prior works, we use a flow-based model also for the inference model $q_\phi(z|x)$. An additive-Gaussian likelihood model $p_\theta(x|z)$ is then allowed for learning nonlinear representations.

To define the distribution $q_\phi(z|x)$, a flow-based model uses a parameterized *invertible* differentiable transformation $z = T_\phi(e|x)$ to map a random seed $e$ (of the same dimension $d_{\mathbb{Z}}$) following a simple base distribution $p(e)$ [13](e.g., a standard Gaussian) to $\mathbb{Z} = \mathbb{R}^{d_{\mathbb{Z}}}$. By deliberately designed architectures, the transformation $T_\phi(\cdot|x)$ is guaranteed to be invertible, yet still being expressive, with some examples that are even universal approximators [78]. Benefited from the invertibility, the defined density can be explicitly given by the rule of change of variables [13, Thm. 17.2]:

$$q_\phi(z|x) = p(e = T_\phi^{-1}(z|x))\left|\nabla_z T_\phi^{-\top}(z|x)\right|,$$

where $\left|\nabla_z T_\phi^{-\top}(z|x)\right|$ is the absolute value of the determinant of the Jacobian of $T_\phi^{-1}(z|x)$ (w.r.t $z$).

**Problem for evaluating the compatibility loss.** Although $T_\phi(z|x)$ is guaranteed to be invertible, in common instances computing its inverse is intractable [66, 46, 80] or costly [31, 8, 18] (however, they all guarantee an easy calculation of the Jacobian determinant $\left|\nabla_z T_\phi^{-\top}(z|x)\right|$ for efficient density evaluation). This means that density estimation of $q_\phi(z|x)$ is intractable for an arbitrary $z$ value, but is only possible for a generated $z$ value, whose inverse $e$ is known in advance (the generated $z$ is computed from this $e$). This however, introduces problems when computing the gradients $\nabla_x \log q_\phi(z|x)$, $\nabla_z \log q_\phi(z|x)$ for the compatibility loss (Eq. (2) or Eq. (3)).

To see this, it is important to distinguish the "formal arguments" and "actual arguments" of a function. It makes a difference when taking derivatives if the actual arguments are fed to formal arguments in an involved way. What we need is the derivatives w.r.t the formal arguments, but automatic differentiation tools (*e.g.*, the `autograd` utility in PyTorch [61]) could only compute the derivatives w.r.t the actual arguments. We use capital subscripts for formal arguments and lowercase letters for actual arguments. Following this rule, we denote $\log q_{Z|X}^\phi(z|x)$ for $\log q_\phi(z|x)$ above, so $\nabla_Z \log q_{Z|X}^\phi$ denotes the gradient function that differentiates the first formal argument $Z$ of $\log q_{Z|X}^\phi$, and similarly for $\nabla_X \log q_{Z|X}^\phi$. Then at a generated value of $z = T_\phi(e|x)$ from a random seed $e$, the required gradients in the compatibility loss are w.r.t to the formal arguments $\nabla_Z \log q_{Z|X}^\phi(T_\phi(e|x)|x)$ and $\nabla_X \log q_{Z|X}^\phi(T_\phi(e|x)|x)$, while automatic differentiation tools could only give the gradients w.r.t the actual arguments $\nabla_e \log q_{Z|X}^\phi(T_\phi(e|x)|x)$ and $\nabla_x \log q_{Z|X}^\phi(T_\phi(e|x)|x)$, which are not the desired gradients. Note that we do not know the exact calculation rule of $\log q_{Z|X}^\phi(z|x)$ for arbitrary $z$ and $x$, but can only evaluate $h^\phi(e, x) := \log q_{Z|X}^\phi(T_\phi(e|x)|x)$ from a given $e$ and $x$. Automatic differentiation could only evaluate the gradients of this $h^\phi(e, x)$ but not of $\log q_{Z|X}^\phi(z|x)$.

---

[13]Although some flow-based models (*e.g.*, the Sylvester flow [80]) also incorporate the dependency on $x$ in the base distribution $\tilde{p}_\phi(\tilde{e}|x)$ (*e.g.*, $\mathcal{N}(\tilde{e}|\mu_\phi(x), \Sigma_\phi(x))$), we can reparameterize this distribution as transformed from a "more basic" parameter-free base distribution $p(e)$ (*e.g.*, $\mathcal{N}(0, I_{d_{\mathbb{Z}}})$) by an $x$-dependent transformation (*e.g.*, $\tilde{e} = \mu_\phi(x) + \Sigma_\phi(x)^{1/2}e$) and concatenate this transformation to the original one as $z = T_\phi(e|x)$.

**Solution.** An explicit deduction is thus required for an expression of the correct gradients in terms of what automatic differentiation could evaluate. From the chain rule, we have:

$$\nabla_e h^\phi(e,x) = \nabla_e \log q^\phi_{Z|X}(T_\phi(e|x)|x) = \left(\nabla_e T_\phi^\top(e|x)\right)\left(\nabla_Z \log q^\phi_{Z|X}(T_\phi(e|x)|x)\right),$$

$$\nabla_x h^\phi(e,x) = \nabla_x \log q^\phi_{Z|X}(T_\phi(e|x)|x) = \left(\nabla_x T_\phi^\top(e|x)\right)\left(\nabla_Z \log q^\phi_{Z|X}(T_\phi(e|x)|x)\right)$$
$$+ \nabla_X \log q^\phi_{Z|X}(T_\phi(e|x)|x).$$

The first equation gives one of the desired gradients: $\nabla_Z \log q^\phi_{Z|X}(T_\phi(e|x)|x) = \left(\nabla_e T_\phi^\top(e|x)\right)^{-1}\left(\nabla_e h^\phi(e,x)\right)$. The term $\nabla_e h^\phi(e,x)$ can be evaluated using automatic differentiation, as mentioned. The other term, *i.e.* the Jacobian $\nabla_e T_\phi^\top(e|x)$, can also use automatic differentiation by tracking the forward flow computation $z = T_\phi(e|x)$, but it is often available in closed-form for flow-based models, as flow-based models need to evaluate its determinant anyway so the architecture is designed to give its closed-form expression.

The second equation gives an expression of the other desired gradient: $\nabla_X \log q^\phi_{Z|X}(T_\phi(e|x)|x) = \nabla_x h^\phi(e,x) - \left(\nabla_x T_\phi^\top(e|x)\right)\left(\nabla_Z \log q^\phi_{Z|X}(T_\phi(e|x)|x)\right)$. Again, the first term $\nabla_x h^\phi(e,x)$ can be evaluated using automatic differentiation. The term $\left(\nabla_Z \log q^\phi_{Z|X}(T_\phi(e|x)|x)\right)$ can be evaluated using the expression we just derived above. For the rest term, *i.e.* the Jacobian $\nabla_x T_\phi^\top(e|x)$, it can also be evaluated using automatic differentiation by tracking the forward flow computation $z = T_\phi(e|x)$. For computation efficiency, this can be implemented by taking the gradient of $z = T_\phi(e|x)$ w.r.t $x$ with the `grad_outputs` argument of `torch.autograd.grad` fed by $\nabla_Z \log q^\phi_{Z|X}(T_\phi(e|x)|x)$ (gradients w.r.t $x$ will not be back-propagated through this $\nabla_Z \log q^\phi_{Z|X}(T_\phi(e|x)|x)$). This reduces computation complexity from $O(d_\mathbb{X} d_\mathbb{Z})$ down to $O(d_\mathbb{X} + d_\mathbb{Z})$. In summary, the desired gradients can be computed via the following expressions:

$$\nabla_Z \log q^\phi_{Z|X}(T_\phi(e|x)|x) = \left(\nabla_e T_\phi^\top(e|x)\right)^{-1}\left(\nabla_e h^\phi(e,x)\right), \tag{21}$$

$$\nabla_X \log q^\phi_{Z|X}(T_\phi(e|x)|x) = \nabla_x h^\phi(e,x) - \left(\nabla_x T_\phi^\top(e|x)\right)\left(\nabla_Z \log q^\phi_{Z|X}(T_\phi(e|x)|x)\right). \tag{22}$$

Second-order differentiations for the compatibility loss can be done in a similar way.

**A simplified compatibility loss.** For the compatibility loss Eq. (3) in the form of Hutchinson's trace estimator, a further simplification is possible. The loss is given by:

$$C(\theta,\phi) = \mathbb{E}_{\rho(x,z)}\mathbb{E}_{p(\eta_x)}\left\|\nabla_Z\left(\eta_x^\top \nabla_X \log p^\theta_{X|Z}(x|z) - \eta_x^\top \nabla_X \log q^\phi_{Z|X}(z|x)\right)\right\|_2^2,$$

with any random vector $\eta_x$ satisfying $\mathbb{E}[\eta_x] = 0, \mathrm{Var}[\eta_x] = I_{d_\mathbb{x}}$. The reference distribution $\rho(x,z)$ can be taken as $p^*(x)q_\phi(z|x)$ for practical sampling for estimating the expectation. For a flow-based $q_\phi(z|x)$, sampling from $(x,z) \sim p^*(x)q_\phi(z|x)$ is equivalent to $(x, T_\phi(e|x)), e \sim p(e)$. So the loss can be reformulated as:

$$C(\theta,\phi) = \mathbb{E}_{p^*(x)p(e)}\mathbb{E}_{p(\eta_x)}\left\|\nabla_Z\left(\eta_x^\top \nabla_X \log p^\theta_{X|Z}(x|T_\phi(e|x)) - \eta_x^\top \nabla_X \log q^\phi_{Z|X}(T_\phi(e|x)|x)\right)\right\|_2^2.$$

Note from Eq. (21), the gradient w.r.t $Z$ is an invertible linear transformation of the gradient w.r.t $e$, so its norm equals zero if and only if the gradient w.r.t $e$ has a zero norm. So to avoid this matrix inversion, we consider a simpler loss that achieves the same optimal solution:

$$\tilde{C}(\theta,\phi) := \mathbb{E}_{p^*(x)p(e)}\mathbb{E}_{p(\eta_x)}\left\|\nabla_e\left(\eta_x^\top \nabla_X \log p^\theta_{X|Z}(x|T_\phi(e|x)) - \eta_x^\top \nabla_X \log q^\phi_{Z|X}(T_\phi(e|x)|x)\right)\right\|_2^2. \tag{23}$$

For additive Gaussian $p^\theta_{X|Z}$, its gradient $\nabla_X \log p^\theta_{X|Z}$ is available in closed-form. For $\nabla_X \log q^\phi_{Z|X}(T_\phi(e|x)|x)$ in the second term, it can be estimated using Eq. (22) we just developed. The subsequent gradient w.r.t $e$ can be evaluated by automatic differentiation. So this loss is tractable to estimate and optimize.

# E   Experiment Details

## E.1   Baseline Methods

We compare our proposed CyGen with bi-directional models (composed of both a likelihood model and an inference model) including Denoising Auto-Encoder (DAE), Variational Auto-Encoder (VAE) and BiGAN. Sketches of these models are introduced below.

**DAE** [82]   first corrupts a real input data $x$ with a local noise and then pass it through an encoder to define $q_\phi(z|x)$. The latent code $z$ is then decoded to the data space by a decoder $p_\theta(x|z)$. The objective is to minimize the reconstruction error (Eq. (19)): $\mathbb{E}_{p^*(x)}\mathbb{E}_{q_\phi(z|x)}[-\log p_\theta(x|z)]$. Compared with VAE, it can be seen as the version with $\beta = 0$, *i.e.* it does not involve a prescribed prior $p(z)$. Nevertheless, optimizing the objective w.r.t $\phi$ may lead to undesired results. Particularly, for any given $x$, it may drive $q_\phi(z|x)$ to only concentrate on $z$ values that maximizes $\log p_\theta(x|z)$. This renders incompatibility and an insufficient determinacy (see Sec. 3.2).

**VAE** [45]   defines a joint distribution $p_\theta(x, z) = p(z)p_\theta(x|z)$ using a specified prior $p(z)$. It learns $p_\theta(x|z)$ to match data distribution $p^*(x)$ with the help of an inference model $q_\phi(z|x)$, using the Evidence Lower BOund (ELBO) objective:

$$\min_{\theta,\phi} \mathbb{E}_{p^*(x)}\mathbb{E}_{q_\phi(z|x)}\left[-\log p_\theta(x|z)\right] + \beta\mathbb{E}_{p^*(x)}[\mathrm{KL}(q_\phi(z|x)\|p(z))]. \tag{24}$$

When $\beta = 1$, the negative objective is a lower bound of the data likelihood (evidence) $\mathbb{E}_{p^*(x)}[\log p_\theta(x)]$ where $p_\theta(x) := \int_{\mathbb{Z}} p(z)p_\theta(x|z)\mathrm{d}z$, hence the name. Optimizing it w.r.t $\phi$ also drives $q_\phi(z|x)$ to the true posterior $p_\theta(z|x)$ and also makes the bound tighter. A $\beta$ other than 1 is considered when there is some desideratum on the latent variable, *e.g.*, disentanglement [37].

**BiGAN** [25, 27].   In addition to learning the data distribution $p^*(x)$ using GAN [30], BiGAN also aims to learn a representation extractor, so it introduces an inference model $q_\phi(z|x)$ which is often deterministic (*i.e.*, a Dirac distribution). The likelihood model (generator) $p_\theta(x|z)$ defines a joint $p(z)p_\theta(x|z)$ with the help of a prescribed prior $p(z)$, while the inference model also defines a joint $p^*(x)q_\phi(z|x)$. Samples from both distributions can be easily drawn, so BiGAN seeks to match them using the GAN loss (Jensen-Shannon divergence) with the help of a discriminator $D(x, z)$. In each training step, the discriminator $D(x, z)$ is first updated on a mini-batch of $p^*(x)q_\phi(z|x)$ data $x^+ \sim p^*(x), z^+ \sim q_\phi(\cdot|x^+)$ with positive labels $y^+ = 1$ and a mini-batch of $p(z)p_\theta(x|z)$ data $z^- \sim p(z), x^- \sim p_\theta(\cdot|z^-)$ with negative labels $y^- = 0$. The goal of the training the discriminator is to minimize the binary cross entropy loss $\mathrm{BCE}(D(x^+, z^+), y^+) + \mathrm{BCE}(D(x^-, z^-), y^-)$. The conditional models $q_\phi(z|x)$ and $p_\theta(x|z)$ are then updated to maximize the same loss $\mathrm{BCE}(D(x^+, z^+), y^+) + \mathrm{BCE}(D(x^-, z^-), y^-)$.

**GibbsNet** [50]   is also considered, which is similar to BiGAN, except that BiGAN's prior-driven joint sample generation $z^- \sim p(z), x^- \sim p(x|z^-)$ is modified to run through multiple cycles: $z_0 \sim p(z), x_0 \sim p(x|z_0), z_1 \sim q(z|x_0), x_1 \sim p(x|z_1), \cdots, z^- \sim q(z|x_{K-1}), x^- \sim p(x|z^-)$. This resembles a Gibbs chain, and is considered in GibbsNet for reducing the influence of a specified prior, as the stationary distribution of the Markov chain does not rely on the initial distribution but is determined by the two conditional models (see Sec. 1.1 (paragraph 2) for its limitation). But this iterated application of the likelihood and inference models makes gradient back-propagation involved. The gradient accumulates with the cycling iteration, resulting in a scale much larger than usual. This makes gradient-based optimization unstable and even leads to numerical instability. We did not find a reasonable result in any experiment using the same architecture so we omit the comparison.

## E.2   Model Architecture

Our code is developed based on the repositories of the Sylvester flow[14] [80] and FFJORD[15] [31] for the task environment and flow architectures. VAE, DAE and CyGen share the same architecture of $p_\theta(x|z)$ and of $q_\phi(z|x)$, which are detailed in Table 16. The inference model $q_\phi(z|x)$ adopts the architecture of Sylvester flow [80], illustrated in Fig. 12. It consists of a neural network (denoted as C-QNN) that outputs $q_{\mathrm{nn}}$, a reparameterization module, and a set of consecutive $N$ flows. The outputs $q_\mu$ and $q_\sigma$ are used to parameterize the diagonal Gaussian distribution for initializing $z_0$, the input to the flows. For implementation simplicity, we choose the Householder version of the Sylvester flow.

---

[14]https://github.com/riannevdberg/sylvester-flows
[15]https://github.com/rtqichen/ffjord

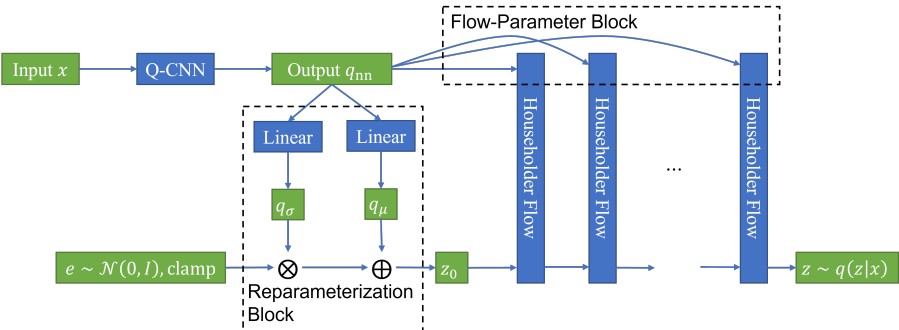

Figure 12: Flow architecture of the inference model $q_\phi(z|x)$. See Table 16 for detailed specification.

For each flow layer, the output $\mathbf{z}_t$ of the flow given input $\mathbf{z}_{t-1}$ is:

$$\mathbf{z}_t = \mathbf{z}_{t-1} + \mathbf{A}_t h(\mathbf{B}_t \mathbf{z}_{t-1} + \mathbf{b}_t),$$

where $\mathbf{A}_t = \mathbf{Q}_t \mathbf{R}_t$, $\mathbf{B}_t = \tilde{\mathbf{R}}_t \mathbf{Q}_t$ and $\mathbf{b}_t$ are parameters of the $t$-th flow and $h$ is the hyperbolic-tangent activation function. Let $\mathbf{A} = \mathbf{QR}$, $\mathbf{B} = \tilde{\mathbf{R}}\mathbf{Q}^\top$, where $\mathbf{R}$ and $\tilde{\mathbf{R}}$ are upper triangular matrices, and $\mathbf{Q} = \prod_{i=1}^H (\mathbf{I} - \frac{2\mathbf{v}_i \mathbf{v}_i^\top}{\mathbf{v}_i^\top \mathbf{v}_i})$ is a sequence of $H = 8$ Householder transformations. All the flow parameters, *i.e.* $\mathbf{v}_{1:H}$, $\mathbf{b}$, $\mathbf{R}$ and $\tilde{\mathbf{R}}$, depend on $q_{\mathrm{nn}}$ via a flow-parameter block.

### E.3 Synthetic Experiments

The synthetic datasets ("pinwheel" in the main text and "8gaussians" in this appendix) are adopted from the above mentioned repositories of the Sylvester flow and FFJORD. The dimension of the data space is $d_{\mathbb{X}} = 2$, and we take the latent space to be of the same dimension $d_{\mathbb{Z}} = 2$.

For the inference model $q_\phi(z|x)$, we use a three-layer MLP for the C-QNN component, with 8 hidden nodes in each layer. After the reparameterization block, consecutive $N = 32$ Householder flow layers are concatenated. Each flow layer has $H = 2$ Householder transformations[16]. For the likelihood model $p_\theta(x|z)$, it is implemented as an additive Gaussian model. Its mean function is a three-layer MLP with 16 hidden nodes in each layer, and its variance is taken isotropic with fixed scale 0.01.

For training, we use the Adam optimizer [43] with batch-size 1000 and weight decay parameter $1 \times 10^{-5}$ for all methods. All methods use a learning rate of $1 \times 10^{-3}$ except for DAE which uses $1 \times 10^{-4}$. For BiGAN, the generator is updated once per 128 updates of the discriminator using step size $1 \times 10^{-4}$. For CyGen, conditional models $p_\theta(x|z)$ and $q_\phi(z|x)$ are trained by minimizing: $1 \times 10^{-5} \times$ compatibility loss Eq. (23) + data-fitting loss Eq. (4), where the expectation in the data-fitting loss is estimated using 16 samples from $q_\phi(z|x)$ with reparameterization [45]. For the version CyGen(PT) with PreTraining, the conditional models are first pretrained as in a VAE by minimizing the ELBO objective Eq. (24) (with $\beta = 1$) using the standard Gaussian prior for 1000 epochs, and are then trained as in CyGen by minimizing the above objective with a 10-times smaller learning rate for the likelihood model $p_\theta(x|z)$ (same learning rate for the inference model $q_\phi(z|x)$). DAE is also pretrained in this way.

For data generation, VAE and BiGAN use ancestral sampling: first draw a sample of $z$ from the standard Gaussian prior $p(z)$, and then draw a data sample $x$ from the likelihood model $p_\theta(x|z)$. For DAE and CyGen, they do not have a prior model for ancestral sampling. They use MCMC methods, including Gibbs sampling and SGLD (see Sec. 3.3). One difference for the synthetic experiment is that the SGLD version is done by passing through the likelihood model $p_\theta(x|z)$ with prior samples drawn via SGLD in the latent space $\mathbb{Z}$ similar to Eq. (5):

$$z^{(t+1)} = z^{(t)} + \varepsilon \nabla_{z^{(t)}} \log \frac{q_\phi(z^{(t)}|x^{(t)})}{p_\theta(x^{(t)}|z^{(t)})} + \sqrt{2\varepsilon}\, \eta_z^{(t)}, \text{where } x^{(t)} \sim p_\theta(x|z^{(t)}), \eta_z^{(t)} \sim \mathcal{N}(0, I_{d_{\mathbb{Z}}}), \quad (25)$$

and $\varepsilon$ is a step size parameter, taken as $3 \times 10^{-4}$. Both Gibbs sampling and SGLD are run for 100 iterations (transition steps). Their comparison for CyGen is shown in Fig. 5, where SGLD is much better. For DAE, using Gibbs sampling and using SGLD produce similar data generation results.

---

[16]To make an invertible transformation, the number of Householder transformations $H$ needs to be no larger than the dimension of $z$, which is 2 in this synthetic experiment

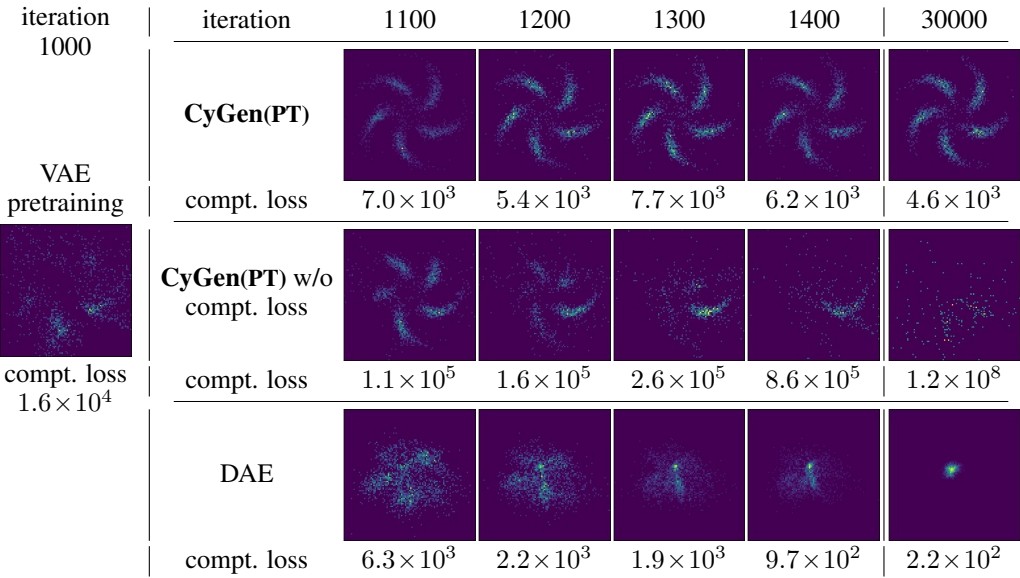

Figure 13: Generated data using $\mathbb{Z}$-space SGLD (Eq. (25)) along the training process after VAE pretraining (iter. 1000) using CyGen(PT), CyGen(PT) without compatibility loss, and DAE. The last DAE result is at iteration 9200, after which numerical overflow occurs.

We show more results next.

**Impact of the compatibility loss.** Fig. 13 shows the training process of CyGen(PT), CyGen(PT) without compatibility loss, and DAE, all after pretrained as a VAE (iter. 1000; leftmost panel), in terms of generated data distribution. We see that the normal CyGen(PT) behaves stably to the end and well approximates the data distribution along the training process. Its compatibility loss is indeed decreasing. On the other hand, CyGen(PT) without the compatibility loss diverges eventually, with an exploding compatibility loss. Although it well optimizes the data-fitting loss Eq. (4), if compatibility is not enforced, the loss is not the data likelihood that we want to optimize.

Note that CyGen(PT) without compatibility loss improves generation quality upon the VAE-pretrained model in the first few training iterations (*e.g.*, at iter. 1100). This is because the ELBO objective (Eq. (24)) of VAE also drives $q_\phi(z|x)$ towards the true posterior $p_\theta(z|x) \propto p(z)p_\theta(x|z)$ (defined with a specified prior $p(z)$) so compatibility approximately holds in the first few iterations, which makes the data-fitting loss (Eq. (4)) effective.

**The collapse process of DAE.** Fig. 13 (rows 1,3) also shows the comparison with DAE. We see that after pretraining, DAE quickly shrinks its data distribution, ending up with a collapsed data distribution, and even finally comes to a numerical problem. This is due to the mode-collapse behavior of the inference model $q_\phi(z|x)$ from minimizing the DAE loss Eq. (19) and the subsequent insufficient determinacy, as explained in Sec. 3.2. Although its compatibility loss is also decreasing, this comes at the cost of the insufficient determinacy which hinders capturing the data distribution and also makes the training process unstable.

**Incorporating knowledge into the conditionals.** We plot the prior distributions in Fig. 14 of VAE, Cy-Gen, and CyGen(PT) with VAE pretraining, in the form of the histogram of the drawn $z$ samples. For Cy-Gen/CyGen(PT), samples are drawn by $\mathbb{Z}$-space SGLD (Eq. (25)) using the same step size $\varepsilon = 3 \times 10^{-4}$ and number of iterations 100. Compared with VAE, the priors learned by CyGen and CyGen(PT) are more expressive. For CyGen which is not subjected to any further constraints, there may be multiple $p_\theta(x|z)$ and $q_\phi(z|x)$ that are compatible and well match the given data distribu-



Figure 14: Prior distributions $p(z)$ of VAE, CyGen (without pretraining), and CyGen(PT) (with VAE pretraining). Prior samples of CyGen/CyGen(PT) are drawn using $\mathbb{Z}$-space SGLD (Eq. (25)).

tion. Using a standard Gaussian prior for pretraining successfully incorporates the knowledge of a centered and centrosymmetric prior into the conditional model $p_\theta(x|z)$. The arbitrariness of possible $p_\theta(x|z)$ and $q_\phi(z|x)$ is largely mitigated in this way. This observation meets the discussion in Sec. 4.1 (paragraph 4) on the aggregated posteriors in Fig. 4.

Figure 15: Generated data (DAE and CyGen use $\mathbb{Z}$-space SGLD Eq. (25)) and class-wise aggregated posteriors of DAE, VAE, BiGAN and CyGen, on the "**8gaussians**" dataset. Also shows results of CyGen(PT) that is PreTrained as a VAE. (Best view in color.)

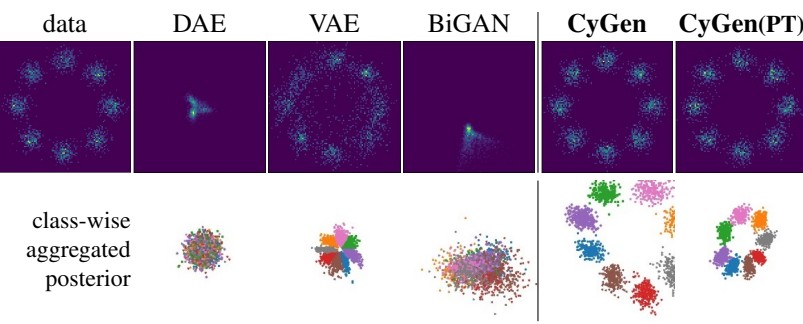

| data | DAE | VAE | BiGAN | **CyGen** | **CyGen(PT)** |

class-wise aggregated posterior

**Results on another similar dataset (8gaussians).** We repeat all the settings above (for the "pinwheel" dataset) on another similar synthetic dataset "8gaussians" (Fig. 15 top-left). Its data distribution also has a (nearly) non-connected support, with 8 connected components. The only difference in settings is that BiGAN uses a learning rate $1 \times 10^{-4}$ for updating its generator and $3 \times 10^{-5}$ for its discriminator. We see similar observations from the results in Fig. 15.

For data generation, DAE again produces a collapsed data distribution. VAE's data distribution is blurred and the 8 clusters are touching, making the support connected due to using the standard Gaussian prior. BiGAN is unstable on this dataset and does not produce a reasonable data distribution, despite some parameter tuning. In contrast, CyGen and CyGen(PT) still recover the data distribution faithfully, with clear 8 clusters. Again, the advantage to overcome manifold mismatch is demonstrated.

For representation learning, DAE again collapses the class-wise aggregated posteriors of all classes and puts them in the same place. VAE identifies the latent clusters but which are compressed to the origin and squeezed together to border each other. BiGAN's aggregated posteriors of different classes largely overlap each other. In contrast, CyGen and CyGen(PT) separate the latent clusters clearly. CyGen(PT) additionally embodies the knowledge from the VAE-pretrained likelihood model that the prior hence the (all-class) aggregated posterior is centered and centrosymmetric, without suffering fitting the data distribution.

In all, these observations again verify that CyGen achieves both superior generation and representation learning performances.

### E.4 Real-World Experiments

**Model architecture.** All methods use the same architecture of the inference model (encoder) and the likelihood model (decoder), illustrated in Fig. 12 and detailed in Table 16 (except for the reported results from [50] of BiGAN and GibbsNet listed in Table 7, which are around random guess using the same flow architecture). The Gaussian variance of the likelihood model $p_\theta(x|z)$ is selected to be 0.01 for all dimensions. All methods use the Adam optimizer [43] with learning rate $1 \times 10^{-4}$ and batch-size 100 for 100 epochs.

**Data generation.** VAE uses ancestral sampling, and DAE uses its standard Gibbs sampling procedure initialized from $z_0 \sim \mathcal{N}(0, I_{d_\mathbb{Z}})$. CyGen(PT) generates data by running $\mathbb{X}$-space SGLD (Eq. (5)) with step size $\varepsilon = 1 \times 10^{-3}$ initialized from $x_0 \sim p_\theta(\cdot|z_0)$ where $z_0 \sim \mathcal{N}(0, I_{d_\mathbb{Z}})$.

**Downstream classification.** For the downstream classification task on the latent space $\mathbb{Z}$, we sample one latent representation $z$ for each data point $x$ directly from the learned inference model $q_\phi(z|x)$, and then train a 2-layer MLP classifier with 10 hidden nodes on top of the latent representation. Results of DAE, VAE and CyGen(PT) in Table 7 are averaged over 10 random trials. All downstream classifiers are trained for 100 epochs.

**Training strategy of our method.** CyGen(PT) first pretrains its likelihood and inference models as in a VAE using the ELBO objective Eq. (24) (with $\beta = 1$ on MNIST and $\beta = 0.01$ on SVHN) for 100 epochs, and then trains them using CyGen ($1 \times 10^{-3} \times$ compatibility loss Eq. (23) + data-fitting loss Eq. (4)) with a 10-times smaller learning rate for the likelihood model (same learning rate for the inference model). On SVHN, we clamp the standard Gaussian random seed $e$ in the reparameterization step that initializes $z_0$ to be within the interval $[-0.1, 0.1]$ (element-wise).

**Remark on the VAE pretraining.** For real-world images, the optimization process of CyGen from a cold start is unstable, possibly because the data distribution roughly concentrates on a low-dimensional space thus is nearly not absolutely continuous, while the CyGen-defined distribution is absolutely continuous (see Lem. C.1 in Appx. C.1). Improved techniques to handle this issue are important future work. Moreover, the VAE pretraining downweighs the effect of the prior (*i.e.*, using a small $\beta$ on SVHN), so it is approximately a DAE, which does not show a reasonable result (Fig. 6). So CyGen is the key to the high-quality results.

Table 16: Inference and likelihood model architectures for MNIST and SVHN

| Layers | In-Out Size | Stride |
|---|---|---|
| **Inference Model $q_\phi(z\|x)$ Architecture for MNIST-C-QNN** | | |
| Input $x$ | $1\times28\times28$ | |
| $5\times5$ GatedConv2d (32), Sigmoid | $32\times28\times28$ | 1 |
| $5\times5$ GatedConv2d (32), Sigmoid | $32\times14\times14$ | 2 |
| $5\times5$ GatedConv2d (64), Sigmoid | $64\times14\times14$ | 1 |
| $5\times5$ GatedConv2d (64), Sigmoid | $64\times7\times7$ | 2 |
| $5\times5$ GatedConv2d (64), Sigmoid | $64\times7\times7$ | 1 |
| $7\times7$ GatedConv2d (256), Sigmoid | $256\times1\times1$ | 1 |
| Output $q_{nn}$, squeeze | 256 | |
| **Inference Model $q_\phi(z\|x)$ Architecture for SVHN-C-QNN** | | |
| Input $x$ | $3\times32\times32$ | |
| $5\times5$ Conv2d (32), LReLU | $32\times28\times28$ | 1 |
| $4\times4$ Conv2d (64), LReLU | $64\times13\times13$ | 2 |
| $4\times4$ Conv2d (128), LReLU | $128\times10\times10$ | 1 |
| $4\times4$ Conv2d (256), LReLU | $256\times4\times4$ | 2 |
| $4\times4$ Conv2d (512), LReLU | $512\times1\times1$ | 1 |
| $4\times4$ Conv2d (256), Sigmoid | $256\times1\times1$ | 1 |
| Output $q_{nn}$, squeeze | 256 | |
| **Reparameterization Block for $q_\phi(z\|x)$ for MNIST and SVHN** | | |
| Input $q_{nn}$ | 256 | |
| Output-1 $q_\mu$: Linear $256\times64$ | 64 | |
| Draw $e \sim \mathcal{N}(0, I_{d_{\mathbb{Z}}})$ and output $z_0 = q_\mu + e \odot q_\sigma$ | 64 | |
| **Flow-Parameter Block for $q_\phi(z\|x)$ for MNIST and SVHN** | | |
| Input $q_{nn}$ | 256 | |
| Output-1 $\mathbf{v}_{1:8}$: Linear $256\times512$ | 512 | |
| Output-2 $\mathbf{b}$: Linear $256\times8$ | 512 | |
| Output-3 $\mathbf{R}$: Linear $256\times(64\times64)$ | $(64\times64)$ | |
| Output-4 $\tilde{\mathbf{R}}$: Linear $256\times(64\times64)$ | $(64\times64)$ | |
| **Likelihood Model $p_\theta(z\|x)$ Architecture for MNIST** | | |
| Input $z$ | $64\times1\times1$ | |
| $7\times7$ GatedConvT2d (64), Sigmoid | $64\times7\times7$ | 1 |
| $5\times5$ GatedConvT2d (64), Sigmoid | $64\times7\times7$ | 1 |
| $5\times5$ GatedConvT2d (64), Sigmoid | $64\times14\times14$ | 2 |
| $5\times5$ GatedConvT2d (32), Sigmoid | $32\times14\times14$ | 1 |
| $5\times5$ GatedConvT2d (32), Sigmoid | $32\times28\times28$ | 2 |
| $5\times5$ GatedConvT2d (32), Sigmoid | $32\times28\times28$ | 1 |
| $1\times1$ GatedConv2d (1), Sigmoid | $1\times28\times28$ | 2 |
| Output $x$ | $1\times28\times28$ | |
| **Likelihood Model $p_\theta(z\|x)$ Architecture for SVHN** | | |
| Input $z$ | $64\times1\times1$ | |
| $4\times4$ ConvT2d (256), LReLU | $256\times4\times4$ | 1 |
| $4\times4$ ConvT2d (128), LReLU | $128\times10\times10$ | 1 |
| $4\times4$ ConvT2d (64), LReLU | $64\times13\times13$ | 1 |
| $4\times4$ ConvT2d (32), LReLU | $32\times28\times28$ | 2 |
| $5\times5$ ConvT2d (32), LReLU | $32\times32\times32$ | 1 |
| $1\times1$ ConvT2d (32), LReLU | $32\times32\times32$ | 1 |
| $1\times1$ Conv2d (32), Sigmoid | $32\times32\times32$ | 1 |
| Output $x$ | $3\times32\times32$ | |