# OpenReview forum: "On the Generative Utility of Cyclic Conditionals"
_NeurIPS.cc/2021/Conference — NeurIPS 2021 Poster_

### Official Review · Reviewer_F92e · 2021-07-16

**Rating:** 6
**Confidence:** 3

**Summary:**

The work answers an interesting question: Given two cyclic conditional distributions, under which condition there exists a compatible joint distribution, and when the compatible joint distribution is unique.

From the theory the authors propose a generative framework called CyGen. CyGen is trained using maximum likelihood based on modelings of the conditional probabilities. Sampling from the marginal distribution modeled by the two conditional distributions is also not straightforward. Luckily this has been investigated before and there are practical MCMC based procedures that converge to the marginal distribution.

The authors also show some experimental results that the proposed CyGen is able to generate more realistic samples compared to VAE or DAE.

**Limitations And Societal Impact:**

 the authors adequately addressed the limitations and potential negative societal impact of their work

**Main Review:**

Problem setting of this paper is interesting. The compatibility and determinacy analysis are very clean. The proposed CyGen algorithm, so far as I know, is novel.

I am a little worried about how useful these cyclic condition settings are. After all, most of the models simply assume some priors instead and they can still generate very realistic-looking samples. Besides the generative settings such as CyGen, is there any other use case for the Cyclic Conditionals?

Another concern is about the practicality of CyGen. Yeah the MCMC based methods are fine but compared to most of the generative models nowadays, it is not as efficient. In the experiment the authors first pretrain the model from VAE, and then use CyGen more like a “finetuning”. I am curious if we run a cold start from CyGen, will we get meaningful results? Will we run into issues such as mode collapse or some types of "bad" local optimum?


**Time Spent Reviewing:**

2 hours

---

> ### Author Response · Authors · 2021-08-10
> **General Response**
>
> Thanks for acknowledging the significance and quality of our theory and the novelty of our method. We also appreciate your valuable feedback, and will include the following discussions in future versions.
>
> * Usefulness for generative modeling.
>
>   Prior-based generative models indeed meet problems in modeling a dense distribution with non-connected support, as shown in the synthetic experiment. This hinders their usage for real-world 3D distributions like the electron density, which is very similar to our synthetic task; see e.g. this [[public link]](https://commons.wikimedia.org/wiki/File:Hydrogen_Density_Plots.png). For such tasks, our CyGen has a clear advantage. For images, CyGen is shown to improve generation quality using the same model structure.
>
> * Other use cases.
>
>   Glad you asked :) Generative modeling is one motivating scenario for considering cyclic conditionals. The consideration is also natural in dual learning (e.g., machine translation, style transfer) where conditional models in both directions are the goal. Our work could then provide a more fundamental perspective and inspire more effective methods. For example, the determinacy theory could verify the sample efficiency over one-direction methods (i.e., discriminative methods), the compatibility loss extends the cycle-consistency loss to allow probabilistic conditionals (especially important in unsupervised dual learning), and the data-fitting loss enables semi-supervised dual learning. We’ll include these outlook discussions.
>
> * Practicality.
>
>   Indeed, MCMC is not as efficient as ancestral sampling, but it is still acceptable ([A1] views the iteration as passing through a generator layer-by-layer) and adopted in trending works [38, A2, A3, A4]. Moreover, data generation is only one utility of generative models, and in many cases people are more interested in the learned representation.
>
> * Pretraining.
>
>   CyGen from a cold start performs as well as from a VAE pretraining in the synthetic experiment (except that the VAE pretraining incorporates the knowledge that the prior is centered and centrosymmetric; see Fig. 11 in appendix). For real-world images, the optimization process of CyGen from a cold start is unstable, possibly because the data distribution roughly concentrates on a low-dimensional space thus is nearly not absolutely continuous, while the CyGen-defined distribution is absolutely continuous (see Appx. C.1). We’ll elaborate on improved techniques to handle this issue in future work. Moreover, the VAE pretraining downweighs the prior (Appx. E.4.2: use $\beta = 0.01$ in Eq. (21)) so it is approximately a DAE, which does not show a reasonable result (Fig. 5). So CyGen is the key to the high-quality results.
>
> [A1] Yang Song, Jascha Sohl-Dickstein, Diederik P. Kingma, Abhishek Kumar, Stefano Ermon, and Ben Poole. "Score-Based Generative Modeling through Stochastic Differential Equations." In International Conference on Learning Representations (2021).
>
> [A2] Yilun Du, and Igor Mordatch. "Implicit Generation and Modeling with Energy Based Models." In Advances in Neural Information Processing Systems 32 (2019): 3608-3618.
>
> [A3] Yang Song, and Stefano Ermon. "Generative Modeling by Estimating Gradients of the Data Distribution." In Advances in Neural Information Processing Systems (2019).
>
> [A4] Ruiqi Gao, Yang Song, Ben Poole, Ying Nian Wu, and Diederik P. Kingma. "Learning Energy-Based Models by Diffusion Recovery Likelihood." In International Conference on Learning Representations. (2021).

---

### Official Review · Reviewer_F3Nz · 2021-07-17

**Rating:** 6
**Confidence:** 2

**Summary:**

Notes:
  -Model the joint p(x,z) by modeling p(x|z) and q(z|x) to form a cycle.
  -Many existing techniques use an uninformative prior p(z).
  -Compatibility and determinacy criterions.
  -Use parameterized density models p(x|z) and q(z|x).
  -Define r(x,z) = log(p(x|z) / q(z|x)) and make the loss the norm of the jacobian, computed over samples from p(x)*q(z|x) (real data / inferred latents).
  -Jacobian is intractable, so instead approximate by taking gradient wrt. x, then multiplying by random vector eta, then taking gradient wrt z.
  -Sample using stochastic gradient langevin dynamics.

**Ethical Concerns:**

No ethical issues, work is purely technical.

**Limitations And Societal Impact:**

Limitation is potential lack of disentanglement, but whether we want this is of course open to discussion.

**Main Review:**

Post-rebuttal: I appreciate the clarifications from the authors and keep my score.

--

This paper proposes a novel criteria for training a joint distribution described by p(x | z) and q(z | x), without the constrain that the z have a simple prior.  The technique clearly works in the sense of producing decent samples and decent latent variables, but results seem unlikely to be state of the art.  At the same time, the analysis and discussion of these cyclical generative models seems like a useful contribution, and gives a more theoretical grounding for this type of algorithm than previous papers in this general area.

Comments:
  -Overall model and objective seems reasonably easy to implement and is cleanly explained in the paper.

  -The experimental evaluation seems rather limited, and the classification accuracies are about the same as GibbsNet and DAE.  The generation quality seems decent, but is not quantitatively evaluated.

  -Is my understanding correct that the langevin dynamics sampling procedure (section 3.3) is only used at test time and does not need to be used at train time?

  -Out of curiosity, if you run a chain of z -> x -> z -> x -> ..., does this chain have good mixing properties in theory?  I.e. will the chain eventually be able to produce samples uniformly from p(x).

**Time Spent Reviewing:**

2

---

> ### Author Response · Authors · 2021-08-10
> **General Response**
>
> Thanks for your valuable feedback.
> * Downstream classification results.
>
>   Classification results of GibbsNet (and BiGAN) in Table 3 are _not_ obtained by the same flow structure but are taken from [33] using a _different_ model structure (Lines 404-408). GibbsNet using the same flow structure did not produce any reasonable result in our trials, due to its involved backpropagation and the consequent unstable training process (Lines 1321-1326 in appendix). We did not choose the working structure of GibbsNet in [33] as the common structure for comparison because both conditional models are Gaussian  so the structure is not appropriate for learning nonlinear representations (Lines 273-281). As for DAE, although it shows a comparable downstream classification performance, it comes at the cost of undesired generation results as shown in Fig. 5, due to its mode-collapsed inference model (Lines 309-311). We’ll better clarify these facts.
>
> * Quantitative evaluation of generation performance.
>
>   The FID scores (lower is better) of the SVHN generation results in Fig. 5 is 157 for DAE, 128 for VAE, and 102 for our CyGen, agreeing with the visual evaluation in Fig. 5.
>
> * The Langevin dynamics sampling procedure.
>
>   Yes indeed, the Langevin dynamics is only used at test time. For training, echoing Line 306, the objective is the compatibility loss Eq. (3) plus the data-fitting loss Eq. (4). Both only involve the expectation under $q_\phi(z|x)$ (note Lines 268-269 for Eq. (3)) which can be estimated using the reparameterization trick as the common technique in VAEs. No expectation under the model-defined data distribution is involved, so there is no need to generate data using the Langevin dynamics at training time.
>
> * Mixing property of Gibbs sampling.
>
>   When the conditionals $p(x|z)$ and $q(z|x)$ satisfy some shared support conditions that guarantee ergodicity, the corresponding Gibbs chain has a unique stationary distribution (Line 69; see also references therein). As for mixing property (convergence rate), there are many results for the Langevin dynamics (e.g., [A1, A2] for exact gradient and [A3, A4] for stochastic gradient; under some conditions, exponential convergence is achieved), but few for Gibbs sampling. In general practice, Langevin dynamics converges faster. Particularly, Table 2 shows that Langevin dynamics performs better and is more robust to incompatibility (Lines 367-371).
>
> [A1] Arnak S. Dalalyan. "Theoretical guarantees for approximate sampling from smooth and log‐concave densities." Journal of the Royal Statistical Society: Series B (Statistical Methodology) 79.3 (2017): 651-676.
>
> [A2] Alain Durmus, and Eric Moulines. "High-dimensional Bayesian inference via the unadjusted Langevin algorithm." Bernoulli 25.4A (2019): 2854-2882.
>
> [A3] Maxim Raginsky, Alexander Rakhlin, and Matus Telgarsky. "Non-convex learning via stochastic gradient langevin dynamics: a nonasymptotic analysis." Conference on Learning Theory. PMLR (2017).
>
> [A4] Arnak S. Dalalyan, and Avetik Karagulyan. "User-friendly guarantees for the Langevin Monte Carlo with inaccurate gradient." Stochastic Processes and their Applications 129.12 (2019): 5278-5311.

---

### Official Review · Reviewer_2j5F · 2021-07-20

**Rating:** 8
**Confidence:** 3

**Summary:**

This paper studies the conditions under which two conditional densities are compatible, with a particular focus on cases relevant to generative modeling, establishing conditions for compatibility and uniqueness. Based on the theoretical results, an objective for regularizing towards compatibility is proposed, implemented, and fit to data.

**Limitations And Societal Impact:**

Yes.

**Main Review:**

In my opinion, this paper has been missing from the VAE literature for ~7 years. It fills an important hole in the literature. Importantly, the paper operationalizes the theory.

That being said, the proofs in the appendix are quite involved and time constraints did not permit me to go through them properly. Nothing obvious seems off, but I didn't check them carefully.

**Time Spent Reviewing:**

2.5

---

> ### Author Response · Authors · 2021-08-10
> **General Response**
>
> Thanks for your acknowledgement! We’ll further try to make the proofs more concise.

---

### Decision · Program_Chairs · 2021-09-27

**Decision:**

Accept (Poster)

**Comment:**

The paper starts with the likelihood and inference model used in variational auto-encoders and studies if and when a joint distribution can be modeled using these two conditionals. The paper is a nice to read and brings to light many of the results on conditionally specified models to the VAE community. I also appreciated a practical implementation with cross-derivatives to encourage the two conditional distributions to be compatible with a joint distribution.  A couple of comments

- I don't think the full Arnold book was cited. It has more results and should be integrated into the narrative

@book{arnold2012conditionally,
  title={Conditionally specified distributions},
  author={Arnold, Barry C and Castillo, Enrique and Alegria, Jose-Maria Sarabia},
  volume={73},
  year={2012},
  publisher={Springer Science \& Business Media}
}

- Edit the paper to frame and highlight the practical implications of the results